

# Edge modes as dynamical frames:
# Charges from post-selection in generally covariant theories

**Sylvain Carrozza**[1,2⋆], **Stefan Eccles**[3†] **and Philipp A. Hoehn**[3,4‡]

**1** Institut de Mathématiques de Bourgogne, UMR 5584 CNRS,
Université de Bourgogne, F-21000 Dijon, France
**2** Institute for Mathematics, Astrophysics and Particle Physics, Radboud University,
Heyendaalseweg 135, 6525 AJ Nijmegen, The Netherlands
**3** Okinawa Institute of Science and Technology Graduate University,
Onna, Okinawa 904 0495, Japan
**4** Department of Physics and Astronomy, University College London,
Gower Street, London, WC1E 6BT, United Kingdom

⋆ sylvain.carrozza@u-bourgogne.fr , † stefan.eccles@oist.jp , ‡ philipp.hoehn@oist.jp

## Abstract

We develop a framework based on the covariant phase space formalism that identifies gravitational edge modes as dynamical reference frames. As such, they enable both the identification of the associated spacetime region and the imposition of boundary conditions in a gauge-invariant manner. While recent proposals considered the finite region in isolation and sought the maximal corner symmetry algebra compatible with that perspective, we here advocate to regard it as a *subregion* embedded in a global spacetime and study the symmetries consistent with such an embedding. This leads to advantages and differences. It clarifies that the frame, although appearing as "new" for the subregion, is built out of the field content of the complement. Given a global variational principle, it also permits us to invoke a systematic post-selection procedure, previously used in gauge theory [1], to produce a consistent dynamics for a subregion with timelike boundary. As in gauge theory, requiring the subregion presymplectic structure to be conserved by the dynamics leads to an essentially unique prescription and unambiguous Hamiltonian charges. Unlike other proposals, this has the advantage that *all* (field-independent) spacetime diffeomorphisms acting on the subregion remain gauge and integrable (as in the global theory), and generate a first-class constraint algebra realizing the Lie algebra of spacetime vector fields. By contrast, diffeomorphisms acting on the frame-dressed spacetime, that we call *relational spacetime*, are in general physical, and those that are parallel to the timelike boundary are integrable. Upon further restriction to relational diffeomorphisms that preserve the boundary conditions (and hence are symmetries), we obtain a subalgebra of conserved corner charges. Physically, they correspond to reorientations of the frame and so to changes in the relation between the subregion and its complement. Finally, we explain how the boundary conditions and conserved presymplectic structure can both be encoded into boundary actions. While our formalism applies to any generally covariant theory, we illustrate it on general relativity, and conclude with a detailed comparison of our findings to earlier works.

# Contents

# 1 Introduction

The development of covariant phase space methods [2–9] that are appropriate for the study of gravitational dynamics in bounded subregions has generated renewed interest in recent years [10–25]. This is in part motivated by the importance of quasi-local charges associated to codimension-two boundaries of spacelike submanifolds – known as *corners* – for classical and quantum gravity. They are, for instance, essential ingredients in the analysis of asymptotic symmetries (see e.g. [20,26–34]), as well as in holographic derivations of Einstein's equations in asymptotically AdS spacetimes [35,36]. More generally, at finite distance, understanding the unitary representations of classical Poisson algebras of corner charges (or deformations thereof) might provide new insights into quantum gravity [32,37–47], at a level of description that neither relies on asymptotic assumptions nor explicitly refers to the infinitesimal structure of spacetime.

An objection one might raise against this program is that the presymplectic structure associated to a corner suffers from ambiguities, which affect the structure of the corner algebra itself. The latter contains a universal part providing a representation of intrinsic diffeomorphisms of the corner, and additional contributions which are non-universal. A classification of those additional contributions was proposed for metric and tetrad formulations of vacuum general relativity in [15], and various strategies to select a particular presymplectic structure (hence, a particular realization of the corner algebra) have been explored in the recent literature [11,13,19,21,23–25]. To our knowledge, the question of how corner ambiguities should be resolved depending on the physical problem at hand is not entirely settled yet, and the present work is a contribution towards its resolution.

As first proposed in an influential paper by Donnelly and Freidel [10] and developed further in [11, 13, 15, 20, 23, 24, 48], a convenient way of analysing the dynamics of a subregion described by Cauchy data on a partial Cauchy slice $\Sigma$ in a gauge or gravitational theory relies on the introduction of non-gauge-invariant extra fields – known as *edge modes*[1] – living at the corner $\partial\Sigma$.[2] Even though they can be reasonably well-motivated by the principle of gauge-invariance, those new degrees of freedom and their transformation properties are strictly-speaking postulated in such constructions. Nonetheless, they can be used to impose gauge-invariance of the presymplectic structure associated to $\Sigma$ under field-dependent gauge transformations. This procedure fixes corner ambiguities to some degree, but as we will see, not completely.

In the present work, we propose an alternative derivation of gravitational edge modes, based on the idea of *post-selection of a global dynamics*, performed in such a way as to induce a consistent dynamics for a subregion. This procedure essentially selects from the space of global solutions those that are compatible with the desired boundary conditions on the interface separating the subregion from its complement. In particular, in contrast to previous works on edge modes that consider the finite region in isolation and, as e.g. in [23, 24, 48], aim for the largest possible corner symmetry algebra from that perspective, we shall treat it as a *sub*system embedded into a larger spacetime and derive the symmetries consistent with such an embedding. This will lead to advantages and crucial differences that we shall explain, and it will clarify the physical interpretations of boundary symmetries and edge modes.

We follow the same strategy as in the related paper [1], which was however limited to the study of gauge theory on a background spacetime manifold. By focusing on the dynamical properties of a subregion $M$ bounded by a time-like boundary $\Gamma$ rather than a single causal

---

[1]There seems to be no general consensus on the terminology in the literature. In many works, like here, one refers to the non-invariant extra fields as "edge modes", while in others one uses this term for the invariant charges on the boundary, built using these extra fields.

[2]$\partial\Sigma$ is the *corner* of the causal domain of $\Sigma$, hence its name.

diamond, this approach is dynamical in nature. Moreover, edge modes are not introduced by hand in this framework, but instead realized as specific non-local functionals of the global fields (with support in both $M$ and its complement $\bar{M}$), which provide *dynamical reference frames* for the relevant gauge group in the same sense in which they appear in the recent literature on quantum reference frames [49–64]. They can in turn be used to construct gauge-invariant observables that capture relational information about $M$ and $\bar{M}$. After post-selection of the dynamics relative to such relational observables and reduction down to $M$, those dynamical frames materialize themselves as independent local fields on the boundary $\Gamma$, which can contribute non-trivially to the regional presymplectic structure. Our proposed realization and interpretation of edge modes as dynamical reference frames, initiated in [1], turns out to be conceptually compatible with that advocated by Gomes, Riello and collaborators [65–71] (traces of this have also appeared in [34,38]). To our knowledge, they were the first to envision edge mode fields – which initially resulted from an abstractly motivated phase space extension – as some kind of dynamical reference frames (or observers in their language), though an explicit link with the recent efforts on quantum reference frames [49–64] was not made. Technically, they implemented the choice of frame through a choice of field space connection, an idea that our incarnation of dynamical frames will indeed connect with. By linking this circle of ideas and extending it into the gravitational realm, we believe the present work makes it all the more compelling.

Given an edge mode frame choice, the next task is to determine the regional presymplectic structure. Our proposal is to fix the ambiguity in the choice of presymplectic form by requiring conservation under evolution of the corner $\partial\Sigma$ along the timelike boundary $\Gamma$, subject to the chosen gauge-invariant boundary conditions. The resulting presymplectic structure is similar to (but different from) earlier proposals [10, 11, 13, 15, 20, 23, 24, 48], as discussed in detail in Sec. 7. Interestingly, it is ambiguity-free given a choice of boundary conditions, and to a large extent even independent of the boundary conditions being imposed (see again Sec. 7 for precise statements). As a result, it can also be employed to define unambiguous corner algebras, which are typically investigated in the absence of boundary conditions (since the latter are not needed in a kinematical set-up).

In generally covariant theories, new conceptual and technical challenges arise from the fact that the gauge group is a group of diffeomorphisms. In particular, dynamical reference frames will enter in the very definition of the subregion and its boundary. As a result, we will find that post-selection is best-implemented in a frame-dressed version of spacetime that we refer to as *relational spacetime*. At the global level, and as observed in [10], diffeomorphisms acting on relational spacetime – that we call *relational diffeomorphisms* – are in general physical, in contrast to gauge diffeomorphisms that act on the original spacetime. The regional presymplectic structure produced by our post-selection procedure has the main advantage of preserving this distinction. More precisely, we find that:

1. as in the global theory, all field-independent spacetime diffeomorphisms are gauge and integrable off-shell, independently of their behavior in the vicinity of the timelike finite boundary, and generate a first-class algebra of constraints that is anti-homomorphic to the Lie algebra of spacetime vector fields;

2. the pre-symplectic potential and form resulting from this construction are fully gauge-invariant on shell, even under field-dependent spacetime diffeomorphisms;

3. on-shell of a set of boundary conditions, relational diffeomorphisms which preserve the invariantly-defined timelike boundary are also integrable but generate non-trivial field-space transformations in general.

To the best of our knowledge, this construction is the only one in the literature that satisfies these properties. These distinguishing features arise, in large part, due to the above insistence

that the bounded region be considered as a subsystem within a well-defined global theory. All aspects of its phase space structure, including its symmetries, must be inherited from, and consistently embedded within, that global theory. Having a genuine Poisson bracket representation of the Lie algebra of gauge diffeomorphisms seems especially beneficial for attempts at constructing a regional quantum theory by seeking unitary representations of this algebra. In this regard, our construction resembles somewhat the extended phase space construction of Isham and Kuchař in canonical geometrodynamics without boundaries [72,73].

Upon further restriction to relational diffeomorphisms that preserve the boundary conditions, one obtains a subalgebra of conserved charges that generate rigid symmetries. We can also recover unambiguous corner algebras from this presymplectic structure, by fixing a particular corner $s$ along the boundary, and restricting our attention to relational vector fields that preserve $s$. In vacuum general relativity with standard choices of boundary conditions (which constrain components of the induced metric and/or the extrinsic curvature tensor of the timelike boundary), the resulting corner algebra provides a representation of $\mathrm{diff}(s)$, and therefore reduces to the universal part of the algebras classified in [15]. Realizing other algebras – such as the algebra of $\mathrm{Diff}(s) \ltimes \mathrm{SL}(2,\mathbb{R})^s$ first described in [10] and investigated further in [42] – is formally possible in our formalism, but to make complete sense, would require the identification of an alternative family of boundary conditions that preserve the well-posedness of the equations of motion. This is a question we have left open for future work.

The paper is organized as follows. In Sec. 2, we introduce the notions of dynamical reference frame fields and relational spacetime, that we rely on to provide a gauge-invariant definition of the subregion of interest and its boundary. These frame fields are constructed from the degrees of freedom of the global theory and will provide an incarnation of gravitational edge modes. After outlining and contrasting the properties of two distinct classes of such frame fields – *material* and *immaterial* frames –, we restrict our attention to the second class. We start Sec. 3 by a review of the covariant phase space formalism, first in spacetime, then in relational spacetime. We then introduce a notion of spacetime covariance relative to the frame, that we refer to as *U-covariance* (where $U$ denotes the frame field), and an ensuing decomposition of field-space objects into $U$-covariant and $U$-non-covariant components. Those concepts play a central role in the rest of the paper, and particularly so in Sec. 4, where we implement the post-selection procedure leading to the definition of a conserved presymplectic structure for the subregion. In Sec. 5, we define integrable charges for arbitrary spacetime diffeomorphisms, as well as for a restricted set of relational diffeomorphisms. The former are found to constitute an algebra of first-class constraints that thus generate gauge-transformations, while the latter are physically non-trivial in general. Relational diffeomorphism charges are not necessarily conserved, but satisfy elementary balance relations that we determine explicitly. Among those, conserved charges form a closed subalgebra under the Poisson bracket induced by our presymplectic structure. In Sec. 6, we explain how to encode the boundary conditions into a regional variational principle involving a bulk and a boundary action that recovers the presymplectic structure of Sec. 4. This regional variational principle follows from the global one upon post-selection on the subset of solutions consistent with the desired boundary conditions on the interface. We illustrate the construction of the boundary action in the context of vacuum general relativity, for three types of boundary conditions, and determine the relational spacetime charges in each case. For instance, for Dirichlet boundary actions, the charges are given by the Brown–York expression [74]. Sec. 7 provides an extensive discussion of our results, and a detailed comparison to the related works [10, 11, 13, 15, 20, 23–25, 48]. It has been designed to be readable by a quick reader, independently from the rest of the manuscript. Finally, we close the paper with a short conclusion in Sec. 8, and, for the reader's convenience, some of our main notations are collected in a glossary at the end of that section.

## 2 Invariant definition of the subregion

### 2.1 Dynamical reference frames

In a generally covariant theory, the notion of spacetime localization is subtle because it is intrinsically relational. This was famously understood by Einstein, after initial struggles with the apparent paradoxes stemming from his "hole argument" [75]. To resolve those paradoxes, spacetime events need to be defined in terms of diffeomorphism-invariant concepts, such as coincidences, rather than points on a representative spacetime manifold $(\mathcal{M}, g)$. In our context, since we are primarily interested in general relativity, this implies that the introduction of a dynamical reference frame is necessary for the very definition of a bounded subregion $M \subset \mathcal{M}$.[3] Dynamical reference frames will therefore play a dual role in our analysis of gravitational edge modes: 1) in a first step, by providing a covariant, and hence physically salient definition of the subregion $M$ and its timelike boundary $\Gamma$; 2) in a second step, by allowing us to decompose the boundary fields into gauge-invariant and gauge-dependent contributions relative to a particular choice of frame. The first item, which we will address in the present section, is the main new feature of gravitational theories as compared to gauge theories in which the geometry of spacetime is non-dynamical [1].

To motivate the general definition of dynamical reference frame we will adopt throughout the paper, let us first outline three concrete implementations of the formalism. The first one is entirely dynamical, and in this sense quite general. The second one, while dynamical too, relies on additional background structure (in the form of a time-like boundary for the global spacetime $\mathcal{M}$) but will prove quite useful for illustrative purposes. The third invokes matter fields to set up a dynamical frame. Nevertheless, formally, all three enjoy the same properties.

Let us first imagine that we are interested in describing the dynamics of a $d$-dimensional Universe in which specific events are known to occur. For definiteness, these events might refer to astrophysical observations of localized phenomena, such as supernovae explosions or black hole mergers. Under suitable conditions (that we will not state explicitly), $d$ such events can be used as references to localize points in some neighborhood $R$ of $\mathcal{M}$, for instance by determining their geodesic distances to any point $p \in R$. We can formalize these ideas by introducing a field-dependent embedding map $E : \{0, \cdots, d-1\} \to \mathcal{M}$, such that $E(A)$ represents the position of the $A$-th event in $\mathcal{M}$. In order for $E$ to describe dynamically defined spacetime events, it must transform appropriately under a diffeomorphism $\varphi : \mathcal{M} \to \mathcal{M}$. Specifically, in order to consistently identify the events $\{E(A)\}$ across different representatives of the same diffeomorphism equivalence class of geometries, they should map into one another under $\varphi$, i.e. $\varphi((\varphi \rhd E)(A)) = E(A)$, where $\rhd$ denotes the action of the diffeomorphism group. This requires the covariance property

$$\varphi \rhd E := \varphi^{-1} \circ E. \tag{1}$$

For any $p \in R$ and $A \in \{0, \cdots, d-1\}$, we can then define

$$U^A(p) := d(p, E(A)), \tag{2}$$

where $d(p, q)$ is the geodesic distance from $p$ to $q$ (for simplicity, we assume that there is a unique geodesic between any two points of $R$). Denoting by $\varphi_* d$ the geodesic distance in the pulled-back metric, it then follows that

$$\varphi \rhd U^A(p) = \varphi_* d(p, (\varphi \rhd E)(A)) = d(\varphi(p), \varphi((\varphi \rhd E)(A))) = U^A(\varphi(p)). \tag{3}$$

---

[3]Besides localization, even bulk microcausality can be defined in a gauge-invariant manner relative to dynamical frame fields [76].

Under suitable conditions, we thus obtain a diffeomorphism $U : R \to r$, $p \to (U^0(p), \ldots, U^d(p))$, where $r$ is an open subset of $\mathbb{R}^d$, which transforms in the following way under a spacetime diffeomorphism:

$$\varphi \rhd U = U \circ \varphi \, . \tag{4}$$

In other words, $U$ can be understood as a *field-dependent local chart* on $R$, defining a *dynamical local coordinate system*. It is defined locally in spacetime, but also in field-space, as is made clear by the construction we have just outlined: $U$ is only well-defined in geometries where the $d$ reference events we implicitly invoked in the construction can themselves be covariantly defined (and, as alluded to already, obey additional conditions). This construction is illustrated in Figure 1.[4]

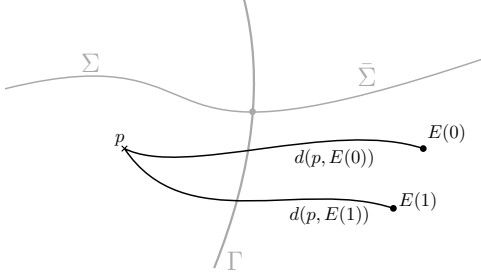

Figure 1: Given a set of reference events $\{E(A), A = 0, \ldots, d\}$, one can introduce a dynamical reference frame $U$ in terms of the geodesic distances $U^A(p) := d(p, E(A))$.

Let us now turn to our second example. In many situations of theoretical or experimental relevance, one is interested in analysing the dynamics of spacetimes with boundaries, subject to specific asymptotic boundary conditions. One may for instance think of asymptotically flat spacetimes, which are especially relevant to the analysis of gravitational waves, or asymptotically AdS geometries which play a prominent role in the context of holography. A number of subtleties (having to do with e.g. soft symmetries or holographic renormalization) arise in the presence of asymptotic boundaries, but given that they are not directly relevant to our exposition we will simply ignore them. For definiteness, let us focus on the case of asymptotically AdS spacetimes, and assume that the time-like boundary $\Gamma_0$ has been regulated in some way so that it is at finite distance from any bulk point. Distinct points on $\Gamma_0$ being physically distinguishable, the gauge group in this context is the subgroup of spacetime diffeomorphisms that vanish on $\Gamma_0$. One way we can then introduce a dynamical coordinate system is the following: 1) we first set up a global coordinate chart $y_0 : \Gamma_0 \to \mathbb{R}^{d-1}$ on the boundary; 2) for any point $p$ in the bulk, we define $z(p)$ as the geodesic distance between $p$ and $\Gamma_0$; 3) if $p$ is sufficiently close to the boundary (and we work locally in field-space), we can assume that there is a unique geodesic of length $z(p)$ connecting $p$ to $\Gamma_0$; 4) calling $q \in \Gamma_0$ the endpoint of this geodesic, we define $y(p) := y_0(q)$. The function $U(p) := (y(p), z(p)) \in \mathbb{R}^d$ then transforms as in (4) under the gauge group, which as already pointed out comprises diffeomorphisms *that vanish on the boundary*. In this sense, the functional $U$ defines a metric dependent — hence dynamical — local coordinate system. This construction is illustrated in Figure 2.

As a third and final example, let us briefly mention dynamical coordinate charts defined by matter fields. Indeed, in experimental or observational contexts, one usually resorts to matter to set up reference systems from which to define coordinates in practice. An idealized setup

---

[4]The construction of geodesic reference frames that we just described, is similar to the strategy employed in [77] to define gauge-invariant observables in three-dimensional quantum gravity on a manifold with boundary. It would be interesting to explore this analogy in more detail.

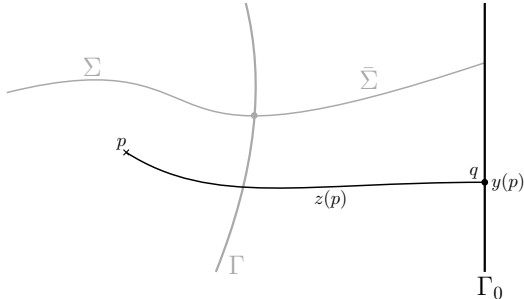

Figure 2: If $\mathcal{M}$ has a time-like boundary $\Gamma_0$, a dynamical reference frame $U(p) := (y(p), z(p))$ can be defined in terms of field-dependent tangential ($y$) and radial ($z$) coordinates.

in which this can be done is given e.g. by Brown-Kuchař dust models [76, 78, 79]. The dust fluid turns out to define a dynamical comoving coordinate system comprised of $d$ scalar fields $U(p) := (T(p), Z_i(p))$ transforming as in (4), where $T$ is the proper time along the dust world lines and the $Z_i$ label the world lines. The dust fields can then be used to deparametrize the theory [79–81].

The three specific constructions we have just described are far from unique. In the first picture, one could for instance change the set of reference events we use to localize other events, or construct the map $U$ in terms of other observables than geodesic distances. Likewise, in the second example, one can set up a dynamical coordinate system in the bulk by shooting in a congruence of geodesics from the boundary at an angle specified by a boundary vector field. The anchor point and proper time or length along the geodesics, for example, then furnish a dynamical coordinate system transforming under diffeomorphisms (that vanish on the boundary) as in Eq. (4); see for instance [76] for a detailed discussion of such a construction. Alternatively, one might decide to coordinatize the bulk in terms of coincidences of families of null rays shot out from the boundary (as in e.g. [82]). Similarly, in the third picture, one could invoke more complicated matter fields, incl. gauge fields to set up a dynamical chart (e.g., see [76] for a general formalism).

Regardless of the specific construction one may want to consider, we will understand any field-dependent local diffeomorphism $U$ transforming as in (4) under the subgroup of (field-independent) gauge diffeomorphisms $\mathrm{Diff}_0(\mathcal{M}) \subset \mathrm{Diff}(\mathcal{M})$[5] as defining a local chart and hence a local dynamical reference frame. We will then say that an atlas of such local charts defines a *dynamical reference frame* for $\mathrm{Diff}_0(\mathcal{M})$; indeed, such a dynamical atlas can be used to parametrize the $\mathrm{Diff}_0(\mathcal{M})$-orbits locally in field space [76]. This constitutes a dynamical frame in the same sense in which quantum reference frames have recently appeared as frames associated with some symmetry group [49–64] and in which edge modes in gauge theories are dynamical frames for the local gauge group [1]. Restricted to a subregion $M \subset \mathcal{M}$ of interest below, we will identify such dynamical frame fields with gravitational edge modes associated with the subregion. Specifically, as these frame fields are constructed from the degrees of freedom of the global theory, the edge modes do not have to be added by hand in contrast to the previous literature. Nevertheless, as we shall see in Sec. 2.4 below, they will appear as "new" degrees of freedom for the subregion.

For simplicity of the exposition, we will assume that the whole spacetime $\mathcal{M}$ can be covered by a single chart, and therefore that a global dynamical reference frame is specified by a

---

[5]Exactly which spacetime diffeomorphisms constitute gauge transformations depends on the setup, including any boundary conditions. For example, as mentioned above, in the presence of an asymptotic boundary typically only those diffeomorphisms are gauge that vanish on that boundary. Note also that the larger set of *field-dependent* gauge diffeomorphisms need not in general form a subgroup of all field-dependent spacetime diffeomorphisms.

diffemorphism $U : \mathcal{M} \to \mathfrak{m}$, from spacetime to a space of physically meaningful parameters $\mathfrak{m}$, transforming according to (4). We will refer to $\mathfrak{m}$ as the *relational spacetime*.[6] Later, we will encounter the pushforward of covariant observables under $U$ from spacetime $\mathcal{M}$ to $\mathfrak{m}$. We will refer to those as *relational observables* as they will describe the original observables relative to the frame in a gauge-invariant manner. As shown in [1, 76], they are covariant versions of the canonical notion of relational observables [49, 56, 79, 83–91]. Note that such a notion of a spacetime global frame is still local in field-space, and it would be interesting to understand how to construct atlases that can in principle cover the whole field-space. However, we will not need this notion here (see [76] for some discussion on this).

Given the nonuniqueness in constructing a dynamical frame field $U$, it is also clear that there does not exist a unique edge mode field for a subregion, but in fact a continuous set of them. In keeping these different choices of edge mode frame fields on an equal footing, it is desirable to be able to relate the different frame-dependent gauge-invariant descriptions. We will not discuss this further in this work. For gauge theories such a framework has been presented in [1], while for generally covariant theories it can be found in [76], giving rise to a notion of dynamical frame covariance (for quantum frame covariance, see [49–64]).

Finally, we will equip $\mathfrak{m}$ with the Lorentzian metric $U^\star g$, where $U^\star$ is the pushforward by $U$.

## 2.2 Subregion localization relative to the frame

Once we have settled on a choice of reference frame $U : \mathcal{M} \to \mathfrak{m}$, it is possible to define physical subregions directly in the space of parameters $\mathfrak{m}$. Indeed, those parameters refer to $\text{Diff}_0(\mathcal{M})$ scalars such as geodesic distances, hence to physical observables that can directly be employed to label spacetime events. We will be interested in a partition of the form $\mathfrak{m} = m \cup \bar{m}$ where $m$ is assumed to have trivial topology, and where $\gamma$, the common boundary shared by $m$ and $\bar{m}$, has the topology of $S^{d-1} \times \mathbb{R}$. The signature of $\gamma$ relative to the dynamical metric $U^\star g$ is dynamical, but we will focus on a domain of field-space where it is everywhere timelike. Similarly, $\sigma$ and $\bar{\sigma}$ will denote everywhere spacelike hypersurfaces with support respectively in $m$ and $\bar{m}$, and such that $\sigma \cup \bar{\sigma}$ constitutes a complete Cauchy slice for the global variational principle.

Finally, we will use capital letters to denote the spacetime counterparts of the previously introduced subregions and hypersurfaces, which are obtained by taking the preimage by $U$: $M := U^{-1}(m)$, $\Gamma = U^{-1}(\gamma)$, $\Sigma := U^{-1}(\sigma)$, etc. Crucially, those quantities are both dynamical and gauge-dependent since they explicitly depend on the dynamical reference frame.

The previous definitions are summarized and illustrated in Figure 3.

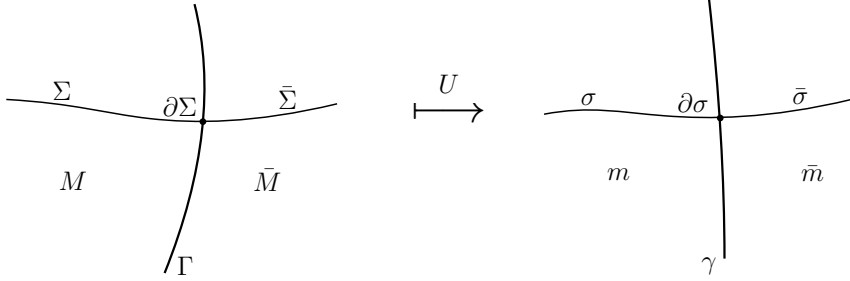

Figure 3: Background subregions and hypersurfaces are first defined in the space of observabes $(\mathfrak{m}, U^\star g)$ (right). Their preimages by $U$ in the spacetime manifold $(\mathcal{M}, g)$ are dynamical (left).

---

[6]Since $\mathfrak{m}$ corresponds to the local values of the frame field $U$, this space is also called the space of local frame orientations in [76].

It is possible to include additional structure in order to capture finer information about how $M$ relates to its complementary region $\bar{M}$. For instance, if we fix a preferred coordinate system $\{U^A\}$ on $\mathfrak{m}$, we can introduce the dynamical *vector frame field*

$$e_\mu^A(p) = \frac{\partial U^A}{\partial x^\mu}(p) \tag{5}$$

on $\mathcal{M}$, where $\{x^\mu\}$ is a local coordinate system in the neighborhood of $p$. The preferred coordinate system $\{U^A\}$ might be operationally well-motivated by e.g. referring to the very definition of the frame in terms of geodesic distances to specific events. Together with the dynamical reference frame $U$, it defines a preferred dynamical basis in the cotangent space $T_p^\star \mathcal{M}$, and by duality in $T_p \mathcal{M}$. This might turn out to be of interest in some contexts.

The vector frame field defined in (5) does not capture independent dynamical information from $U$ since, apart from the explicit dependence on a choice of coordinate, it only depends on $U$ itself. However, it is easy to construct other examples of dynamical vector frame fields, which are dynamically independent from $U$. For instance, in the example illustrated in Fig. 2, one might parallel transport a local frame defined on $\Gamma_0$ along a system of geodesics going into the bulk. Under suitable restrictions, this would define a dynamical vector field $\tilde{e}_\mu^A$, which is in general distinct from $e_\mu^A$. In fact, by being sensitive to the local curvature in $\bar{M}$ (and not just geodesic distances from $\Gamma_0$), $\tilde{e}_\mu^A$ captures independent dynamical information that cannot be inferred from $U$ alone. Hence, if we decide to use a vector frame field such as $\tilde{e}_\mu^A$ to describe the subregion $M$, this may lead to additional edge modes that are not already captured by $U$. If the background frame on $\Gamma_0$ is chosen to be orthonormal, then $\tilde{e}_\mu^A$ defines a dynamical tetrad field.

## 2.3 Embedding field, gauge transformations and symmetries

The initial edge mode construction of Donnelly and Freidel [10], as well as related proposals [11,13,15,20,23,24,48] are formulated in terms of an abstract embedding field $X$, mapping a certain reference spacetime into spacetime. Given the transformation properties of the embedding field, one can consistently make the identification $X := U^{-1}$, and thus reinterpret it as a dynamical reference frame. Doing so allows us to recover the structure of edge modes deriving from the embedding field formalism in a conceptually transparent physical set-up. In particular, it becomes clear that both the embedding field itself and the physical charges that can be subsequently derived from this extended structure can, from the point of view of the global field-space, be interpreted as nonlocal functionals of the fundamental fields. As such, they should not be viewed as new local degrees of freedom that need to be postulated in order to consistently describe the bounded subregion $M$, but rather as composite objects that account for intrinsically relational information between $M$ and $\bar{M}$.

We will explain how this works in detail in the remainder of the paper, but we already emphasize here that in this picture, the distinction between gauge transformations and symmetries put forward in [10] is built-in from the outset. Gauge transformations are associated to the active diffeomorphisms $\text{Diff}_0(\mathcal{M})$ of $(\mathcal{M}, g)$, and do not affect any physical observables. As we have seen in (4), they *act on the frame from the right*. By contrast, diffeomorphisms of the relational spacetime $(\mathfrak{m}, U^\star g)$ directly act on physical observables and they *act from the left* on the frame, $U \mapsto f \circ U$, $f \in \text{Diff}(\mathfrak{m})$. These are what the authors of [10,23,48] refer to as symmetries. Note that in these works, the term symmetry is often used in a loose sense to qualify any transformation on the field-space associated to the subregion $M$ that acts non-trivially on physical observables. In the present paper, we will keep up with the more traditional nomenclature: a symmetry is not only a transformation that affects physical observables, but also a field-space transformation that maps solutions to solutions (and, in particular, preserves

boundary conditions). As will be explained in Sec. 5, under suitable conditions, a subclass of diffeomorphisms of $(\mathfrak{m}, U^\star g)$ will turn out to correspond to actual symmetries of the subregion $M$.

There are, of course, also actions of $\mathrm{Diff}(\mathfrak{m})$ that do not amount to a transformation on field-space and so we cannot unambiguously associate diffeomorphisms of $\mathfrak{m}$ with symmetries. We distinguish:

(i) *"Passive relational diffeomorphisms"*. The transformation $U \mapsto U' = f \circ U$, $f \in \mathrm{Diff}(\mathfrak{m})$, can be induced by a redefinition of what we mean by the frame *without* changing field configuration on $\mathcal{M}$. This amounts to a mere change of coordinates on both field space and spacetime. Indeed, spacetime events are now labeled by the new coordinates $U'^{A'} = f^{A'}(U^A)$. For instance, in our first example for the frame $U$, spacetime events would now be labeled by some smooth invertible functions of the geodesic distances to the events $\{E(A)\}$. If we permit $f$ to depend on the field configuration, this could be achieved, for example, by choosing some suitable distinct set of reference events $\{E'(A')\}$ relative to which we measure the geodesic distance, without changing the field configuration in $\mathcal{M}$. This amounts to a change of *description* of a given field configuration, hence can always be done, and not an action of $\mathrm{Diff}(\mathfrak{m})$ on field space. This action of $\mathrm{Diff}(\mathfrak{m})$ will be generated by Lie derivatives $\mathcal{L}_\rho$ on relational spacetime $\mathfrak{m}$ which act on *all* fields on $\mathfrak{m}$, i.e. incl. any possible background fields.

(ii) *"Active relational diffeomorphisms"* = *frame reorientations*. The transformation $U \mapsto f \circ U$ for the same $f \in \mathrm{Diff}(\mathfrak{m})$, can also correspond to a change of field configuration on $\mathcal{M}$ which 'reorients' the frame $U$ and leaves the remaining degrees of freedom, say $\Phi$, invariant. Accordingly, we call such transformations *frame reorientations*. For instance, in our first example for $U$, this would amount to a change of field configuration that changes the geodesic distance between the set of events $\{E(A)\}$ and the spacetime points we are interested in, but not the values of the other fields $\Phi$ at those points. This action of $\mathrm{Diff}(\mathfrak{m})$ will be generated by a field space Lie derivative $\mathsf{L}_{\check{\rho}}$, where $\check{\rho}$ is the field space vector field corresponding to a vector field $\rho$ on $\mathfrak{m}$. As such, it acts only on dynamical fields. We will see in Sec. 2.4 that not all frame reorientations will map solutions to solutions; those that do are the symmetries.

While the two actions are *a priori* distinct, they agree for covariant quantities, as we will discuss in more detail in Sec. 3.4. However, not all interesting quantities on relational spacetime are covariant; for instance, boundary conditions on $\gamma$ will become background fields. In what follows, we will be primarily interested in the non-trivial case of frame reorientations (case (ii)), assuming they are dynamically permitted, rather than the frame redefinitions (case (i)).

We note that embedding fields have been employed as dynamical frame fields in gravity before the advent of the literature on edge modes, particularly by Isham and Kuchař [72, 73]. While formulated in canonical (ADM) language, their discussion parallels aspects of the one in [10], specifically the distinction of the role assumed by diffeomorphisms acting on the embedding field from the left or right.[7] Their work thus constitutes a canonical version of the extended phase space construction appearing in the covariant setting of the literature on gravitational edge modes.

We conclude this subsection by illustrating the geometric role of $\mathrm{Diff}(\mathfrak{m})$ (in the sense of (ii)) in our three main examples of dynamical reference frames.

---

[7]In particular, their changes of embedding correspond to our frame reorientations. They are symmetries generated by smeared embedding momenta $P(\vec{V})$ which are the canonical charges and realize the algebra of spatial diffeomorphisms.

To begin with, a diffeomorphism $f \in \text{Diff}(\mathfrak{m})$ might fail to leave the partitioning of $\mathfrak{m}$ into $m \cup \bar{m}$ invariant. This occurs whenever $f$ does not leave $\gamma$ invariant: it may correspond to a change of field configuration that affects the geodesic distance between the time-like boundary $\Gamma$ and the reference events $\{E(A)\}$ (in our first example), or the global boundary $\Gamma_0$ (in our second example), or that moves $\Gamma$ relative to the reference dust fluid (in our third example). The action of such a diffeomorphism can be interpreted as a physical change of the subsystem decomposition of $m \cup \bar{m}$ relative to $U$. Hence, from the point of view of $m$, it is intrinsically an open system transformation; therefore, we should not expect it to give rise to integrable charges. Indeed, we will find out that such a transformation is not Hamiltonian with respect to the presymplectic form we will derive in Sec. 4.

By contrast, any diffeomorphism $f$ that leaves $\gamma$ invariant does not affect the invariant definition of the subregion $M$ in terms of the reference frame $U$. In our three examples, this means that the physical location of $\Gamma$, as defined in terms of geodesic distances from dynamical or background reference events, or as defined relative to the dust fluid, is not affected by $f$. In this sense, we can say that $f$ induces a closed system transformation of $M$ relative to $U$. If $f$ is taken to be infinitesimal, we should expect the corresponding vector field $\rho$ to be associated to an integrable charge $Q[\rho]$. Indeed, that is exactly what we will find out in Sec. 5 for the canonical frame-dependent presymplectic structure derived in Sec. 4. In addition, we will also elucidate under which boundary conditions such charges can be associated to proper symmetries of the subregion.

## 2.4 Dynamically (in-)dependent frames: material vs. immaterial

So far we have treated the various realizations of the dynamical frame $U$ on an equal footing as their transformation properties are the same. There is, however, a fundamental difference between the geodesic frames of the first two examples and the dust frame of the third. The former are non-local and immaterial, while the latter constitutes a local matter source of the equations of motion. This implies repercussions for whether the frame can be dynamically independent of the metric or matter degrees of freedom in the subregion $M$ of interest (i.e. can be varied independently on the space of solutions) and this, in turn, leads to consequences for the status of symmetries and charges. In particular, note that in previous works on edge modes in gravity [10, 11, 13, 15, 20, 23, 24, 48] they were (sometimes tacitly) considered as additional local and dynamically independent degrees of freedom in the subregion $M$. We also note that the following differences between material and immaterial frame fields resemble qualitatively those between matter and pure connection-based edge modes in gauge theory [1].

Let us begin with the dust frame. The qualitative argument should hold for any local matter frame. The dynamical dust coordinates $(T, Z^k)$ couple locally to the metric [78] as defined by the Lagrangian form $L_{\text{dust}} = -\frac{1}{2} \epsilon \mu (g^{\alpha\beta} V_\alpha V_\beta + 1)$, where $\epsilon$ is the volume form associated with $g$, $V = -\mathrm{d}T + W_k \mathrm{d}Z^k$ is the dust four-velocity one-form on $\mathfrak{m}$, and $\mu$ the rest-mass density. As a result, the dust coordinates locally source the Einstein equations in $M$. Hence, $U = (T, Z^k)$ cannot be varied independently of the metric in $M$ in an arbitrary manner while remaining on the space of solutions. That is, not all frame reorientations $U \mapsto f \circ U$, $f \in \text{Diff}(\mathfrak{m})$, which amount to a diffeomorphism of the dust coordinates $(T, Z^k)$, can be realized as symmetries (and lead to charges) of the theory. Indeed, as shown in [78], the only diffeomorphisms of $(T, Z^k)$ that correspond to symmetries of the theory are those that preserve the one-form $V$, which constitutes a preferred structure on relational spacetime. As the action of this subset of $\text{Diff}(\mathfrak{m})$ on $(T(p), Z^k(p))$ does not depend on $p \in M$, it corresponds to *global* symmetries of the total action from the point of view of $M$ and gives rise to non-trivial charges.

The situation is quite different for the immaterial geodesic frames of the first two examples for $U$. As they are non-locally constructed from the metric, they cannot be varied independently of $g$ on the global piece $M \cup \bar{M}$ of spacetime. However, when we restrict our attention

to the finite subregion $M$, there *are* circumstances, thanks to its non-local nature, when $U$ becomes dynamically independent from $g$ and any matter degrees of freedom in $M$. Indeed, let us suppose that the geodesics defining $U$ have support in the causal complement of $M$ in $\bar{M}$ so that $U$ is not dynamically determined by the physics inside $M$. For instance, in our first example, we could imagine the reference events $\{E(A)\}$ to all reside in the causal complement of $M$, while in the second example, we may consider the case where all geodesics defining the distance between $\Gamma$ and $\Gamma_0$ pass through the causal complement of $M$. Now we can change the initial data in the causal complement of $M$ without affecting the local physics inside $M$ itself. There will therefore exist global solutions on $M \cup \bar{M}$ which are diffeomorphically inequivalent in $\bar{M}$ and agree in $M$ (up to diffeomorphisms), except that they generally feature distinct values of $U$ in $M$. These distinct global solutions are thus related by symmetries and we can vary $U$ on solutions in *some ways* independently of the other degrees of freedom in $M$.

In fact, owing to the local nature of the Einstein equations, one can glue quite generic pairs of valid initial data surfaces across a common extended interface, in such a way that the initial data remains unaltered but in an arbitrarily small neighborhood of the interface [92–96]. Accordingly, for a fixed $M$, one can choose its causal complement to be sufficiently large by gluing a suitably large initial data surface to an initial data surface of some neighborhood of $M$ without modifying the initial data inside $M$. This should give rise to a set of global solutions for $M \cup \bar{M}$ which in $M$ agree up to diffeomorphisms and the $U$ configuration while creating sufficient "wiggle room" for varying the metric configuration in the causal complement of $M$ such that *all* frame reorientations $U \mapsto f \circ U$, $f \in \mathrm{Diff}(\mathfrak{m})$ can be realized via symmetries of the global theory. Since the regional solutions for $M$ are embedded into the global ones, these frame reorientations will also map regional solutions into one another (though as they may alter the frame-dressed boundary conditions on $\Gamma$, they may simultaneously change the regional theory (see Sec. 6.3).

In conclusion, dynamically independent frames—and thereby edge mode fields in line with the previous literature [10,11,13,15,20,23,24,48]—can typically only arise when we invoke non-local, immaterial constructions like geodesic frames. As in the gauge theory context of [1], these edge mode fields can then be viewed as "internalized" external reference frames for $M$ that originate in its complement. All frame-dressed (or relational) observables associated with "bare" quantities in $M$ then describe in a gauge-invariant manner how $M$ relates to its complement. Most of our following construction is assuming such immaterial edge mode fields, particularly when computing the charges associated with symmetries. We will comment on steps where the construction would differ when using material frames, though we do not pursue that route here explicitly and leave it for future work.

# 3 Covariant phase space formalism

In this section, we will start by recalling background material on the covariant phase space analysis of a generally covariant system, defined by a Lagrangian $d$-form $L$ on $\mathcal{M}$. We will then apply this formalism to the frame-dressed Lagrangian form $U^\star L$, defined on the relational spacetime $\mathfrak{m}$, which we will argue to be the primary object of interest in our approach. The resulting frame-dressed presymplectic current $U^\star[\omega_\chi]$, which has already been extensively investigated in [23,24], will play the role of *bare* presymplectic structure in the rest of our construction. Indeed, as we will explain in Sec. 3.3, there is a sense in which $U^\star[\omega_\chi]$ can be further decomposed into gauge-invariant and gauge-dependent contributions relative to $U$.

## 3.1 Spacetime variational principle

The covariant phase space formalism can be conveniently described in terms of Anderson's variational bicomplex [97]. In this framework, the standard exterior algebra $(\mathrm{d}, \lrcorner, \wedge)$ on $\mathcal{M}$ is supplemented with a field-space exterior algebra $(\delta, \cdot, \wedge)$, together with axioms governing their interplay. In this language, variations of the fundamental fields are described in terms of the field-space exterior derivative $\delta$. A general form field $\alpha$ in the variational bicomplex can be graded by a pair of nonnegative integers $(p, q)$, $p$ representing the spacetime degree and $q$ the field-space degree. In this case, we will say that $\alpha$ is a $(p, q)$-form. Likewise, d is a derivative of degree $(1, 0)$ while $\delta$ is a derivative of degree $(0, 1)$. With this convention, the two exterior derivatives, which both square to 0 ($\mathrm{d}^2 = 0$ and $\delta^2 = 0$), also commute among each other ($[\mathrm{d}, \delta] = 0$).[8] We will recall other relevant properties of the variational bicomplex as we proceed, but encourage the interested reader to consult the appendix of reference [15] for a complete list of axioms compatible with our conventions. Finally, as is common in the literature, we will keep the field-space wedge-product $\wedge$ implicit.

Given a Lagrangian density $L$ – that is, a $(d, 0)$-form – on the spacetime manifold $\mathcal{M}$, one can compute the fundamental variational identity

$$\delta L = E + \mathrm{d}\theta \,, \tag{6}$$

where $E$ is a $(d, 1)$-form encoding the Euler-Lagrange equations of the system, and the $(d-1, 1)$-form $\theta$ is known as the *presymplectic potential*. For instance, vacuum general relativity can be defined by the Einstein-Hilbert Lagrangian $L = R\epsilon$ where $R$ is the Ricci scalar and $\epsilon$ the volume form associated to $g$. Taking the variation of this Lagrangian and using the Leibniz rule to pull-out a bulk contribution proportional to variations of the metric rather than its derivatives, we find (6) to hold with

$$E := E^{\mu\nu}\delta g_{\mu\nu}\epsilon = \left(-R^{\mu\nu} + \frac{1}{2}Rg^{\mu\nu}\right)\delta g_{\mu\nu}\epsilon \quad \text{and} \quad \theta := \left(g^{\alpha\mu}g^{\beta\nu} - g^{\alpha\beta}g^{\mu\nu}\right)\nabla_{\beta}\delta g_{\mu\nu}\epsilon_{\alpha}, \tag{7}$$

where $\epsilon_{\alpha} := \partial_{\alpha}\lrcorner\,\epsilon$. The global action $S = \int_{\mathcal{M}} L$ is then stationary provided that the Einstein equations hold in the bulk and suitable boundary conditions are imposed to make $\theta$ vanish on $\partial\mathcal{M}$.[9]

To any field-independent vector field $\xi$ on $\mathcal{M}$, we associate the field-space vector field $\hat{\xi}$, which provides a field-space representation of the spacetime Lie derivative $\mathcal{L}_{\xi}$. That is, it acts on the metric as $\hat{\xi} \cdot \delta g = \mathcal{L}_{\xi} g$, and similarly for any other dynamical field contributing to $L$. We can also introduce a field-space Lie derivative $\mathsf{L}_{\hat{\xi}}$, a derivative of degree $(0, 0)$ whose action on general forms is specified by Cartan's magic formula $\mathsf{L}_{\hat{\xi}} := \delta \hat{\xi} \cdot + \hat{\xi} \cdot \delta$. It follows that the *anomaly operator* [98]

$$\Delta_{\xi} := \mathsf{L}_{\hat{\xi}} - \mathcal{L}_{\xi} \tag{8}$$

vanishes on any form that is covariant with respect to $\xi$, and in this sense can be used to quantify the degree of covariance of a theory. In the rest of the paper, we will assume the Lagrangian density $L$ to be generally covariant, and the presymplectic potential entering (6) to be anomaly-free, in the sense that:

$$\forall \xi \in \mathfrak{X}(\mathcal{M})\,, \qquad \Delta_{\xi}L = 0 \qquad \text{and} \qquad \Delta_{\xi}\theta = 0\,. \tag{9}$$

---

[8]Different conventions regarding the grading of forms may be found in the literature. Here, we adopt the bigraded convention common in the physics literature. In this convention, $\delta$ and d commute, while they might anticommute in other conventions.

[9]Depending on the desired boundary conditions, one may need to include an additional boundary contribution to the action in order to achieve this. For instance, Dirichlet boundary conditions may be imposed after inclusion of the Gibbons-Hawking-York boundary term.

Those two relations hold in particular for the Einstein-Hilbert Lagrangian and the presymplectic potential provided in (7).

The notion of covariance under field-independent diffeomorphisms can be extended in several ways to the field-dependent context. We will invoke two such notions in this paper, which need to be carefully distinguished. The first notion relies on an extension of the *anomaly operator* (10) encompassing field-dependent diffeomorphisms (see e.g. [19, 98]):

$$\Delta_\xi := \mathsf{L}_{\hat{\xi}} - \mathcal{L}_\xi - \widehat{\delta\xi} \cdot \tag{10}$$

The inclusion of the third term,[10] which vanishes for a field-independent $\xi$ and when acting on field-space 0-forms, ensures that $\Delta_\xi$ acts point-wise in field-space just like its field-independent counterpart. Hence, given our assumptions in (9), we automatically have $\Delta_\xi L = 0$ and $\Delta_\xi \theta = 0$ for any field-dependent $\xi$. As a second option, we can define a notion of covariance with respect to $\xi$ that directly compares the respective actions of $\mathsf{L}_{\hat{\xi}}$ and $\mathcal{L}_\xi$ (that is, without including the correction term $-\widehat{\delta\xi}\cdot$ entering the definition of the anomaly operator). This notion is genuinely more stringent than the field-independent notion of covariance, since covariance under the latter does not imply covariance under the former. To avoid any confusion when discussing field-dependent transformations acting on forms of field-space degree at least one, we will say that:

- $\alpha$ is *non-anomalous with respect to $\xi$* if $\Delta_\xi \alpha = 0$;

- $\alpha$ is *covariant with respect to $\xi$* if $\mathsf{L}_{\hat{\xi}} \alpha = \mathcal{L}_\xi \alpha$.

We will moreover say that $\alpha$ is *anomaly-free* (resp. *generally covariant*) whenever the first (resp. second) statement holds for *any* field-dependent $\xi$. From our assumptions in (9), $L$ is both anomaly-free and generally covariant, while $\theta$ is anomaly-free but not necessarily generally covariant. In particular, the Einstein-Hilbert Lagrangian is both anomaly-free and generally covariant, while its associated $\theta$ in (7) is anomaly-free, but not generally covariant.

The covariance of the Lagrangian translates into:

$$0 = \Delta_\xi L = \hat{\xi} \cdot \delta L - \mathrm{d}\xi \lrcorner L = \hat{\xi} \cdot E + \mathrm{d}\left(\hat{\xi} \cdot \theta - \xi \lrcorner L\right), \tag{11}$$

from which we infer that the *Noether current*

$$J_\xi := \hat{\xi} \cdot \theta - \xi \lrcorner L \tag{12}$$

is conserved on-shell of the equations of motion: $\mathrm{d}J_\xi = -\hat{\xi} \cdot E \approx 0$.[11] A crucial feature of generally covariant theories is that the Noether current is trivially conserved,[12] in the sense that it can be expressed in the form

$$J_\xi = C_\xi + \mathrm{d}q_\xi, \tag{13}$$

---

[10]$\widehat{\delta\xi}$ can be understood as a field-space one-form valued in the space of field-space vector fields – that is, an object of degree $(-1, 1)$ – as follows: its action on dynamical fields is defined via Cartan's magic formula for spacetime forms by $\widehat{\delta\xi} \cdot \delta\Phi = \mathcal{L}_{\delta\xi}\Phi = \mathrm{d}\delta\xi \lrcorner \Phi + \delta\xi \lrcorner \mathrm{d}\Phi$. In this sense, $\widehat{\delta\xi}$ produces a field-space one-form upon contraction with another field-space one-form.

[11]As is standard in the literature, we use the weak equality symbol $\approx$ for relations that hold on-shell of the equations of motion $E = 0$ (which define the solution space $\mathcal{S}$). When a weak equality involves forms of non-vanishing field-space degree, a pullback to $\mathcal{S}$ is implicitly assumed, so that field space variations are horizontal in the space of solutions.

[12]We refer to [99] for an interesting historical and philosophical discussion of this point.

where $C_\xi$ is a constraint that vanishes on-shell ($C_\xi \approx 0$). The quantity $q_\xi$, usually referred to as the *charge aspect*, is furthermore linear in $\xi$ and will play a central role in our construction. For illustrative purposes, let us see how this comes about in vacuum general relativity. Given (7), we can directly compute:

$$\hat{\xi} \cdot E = 2E^{\mu\nu}\nabla_\mu \xi_\nu \epsilon = 2\nabla_\mu(E^{\mu\nu}\xi_\nu)\epsilon - 2\nabla_\mu E^{\mu\nu}\xi_\nu \epsilon = -\mathrm{d}C_\xi - D_\xi, \tag{14}$$

where $C_\xi := -2E^{\mu\nu}\xi_\nu \epsilon_\mu \approx 0$ and $D_\xi := 2\nabla_\mu E^{\mu\nu}\xi_\nu \epsilon$. Furthermore, given that $\xi$ parametrizes a local symmetry, $D_\xi$ must actually vanish identically (which implies the Bianchi identity $\nabla_\mu E^{\mu\nu} = 0$).[13] It follows that $\mathrm{d}J_\xi = \mathrm{d}C_\xi$, where $C_\xi$ is a constraint. There must therefore exist a $(d-2,0)$-form $q_\xi$ such that (13) holds. After a standard computation, one can prove that

$$q_\xi := \epsilon_{\mu\nu}\nabla^\mu \xi^\nu, \tag{16}$$

where $\epsilon_{\mu\nu} := \partial_{\mu\lrcorner}\partial_{\nu\lrcorner}\epsilon$, is a valid choice.

Let us now examine the consequence of the anomaly-freeness of the presymplectic potential postulated in (9). From our definitions, we have:

$$0 = \Delta_\xi \theta = \mathsf{L}_{\hat{\xi}}\theta - \mathcal{L}_\xi \theta - \widehat{\delta\xi} \cdot \theta = \delta\hat{\xi} \cdot \theta + \hat{\xi} \cdot \delta\theta - \mathrm{d}\xi_{\lrcorner}\theta - \xi_{\lrcorner}\mathrm{d}\theta - \widehat{\delta\xi} \cdot \theta \tag{17}$$

$$= \delta J_\xi - J_{\delta\xi} + \hat{\xi} \cdot \delta\theta - \mathrm{d}\xi_{\lrcorner}\theta + \xi_{\lrcorner}E, \tag{18}$$

where (6) and (12) have been invoked in the second line. Introducing the *presymplectic current* $\omega := \delta\theta$ and using the expression (13) of the Noether current, we obtain the standard but central relation

$$-\hat{\xi} \cdot \omega = \delta C_\xi - C_{\delta\xi} + \xi_{\lrcorner}E + \mathrm{d}\left(\delta q_\xi - q_{\delta\xi} - \xi_{\lrcorner}\theta\right) \approx \mathrm{d}\left(\delta q_\xi - q_{\delta\xi} - \xi_{\lrcorner}\theta\right). \tag{19}$$

In vacuum general relativity, the presymplectic current takes the form

$$\omega = P^{\alpha\beta\gamma\lambda\mu\nu}\epsilon_\alpha \delta g_{\beta\gamma}\nabla_\lambda \delta g_{\mu\nu}, \tag{20}$$

with $\epsilon_\alpha := \partial_{\alpha\lrcorner}\epsilon$ and

$$P^{\alpha\beta\gamma\lambda\mu\nu} = \left(g^{\alpha\nu}g^{\lambda\gamma}g^{\beta\mu} - \frac{1}{2}g^{\alpha\beta}g^{\lambda\gamma}g^{\mu\nu} - \frac{1}{2}g^{\alpha\mu}g^{\lambda\nu}g^{\beta\gamma} + \frac{1}{2}g^{\alpha\lambda}g^{\mu\nu}g^{\beta\gamma} - \frac{1}{2}g^{\alpha\lambda}g^{\nu\gamma}g^{\beta\mu}\right).$$

One obtains a *presymplectic form* for the global variational principle by integrating $\omega$ over a complete and *background* Cauchy slice $\Sigma_0$ of $\mathcal{M}$:

$$\Omega(\Sigma_0) := \int_{\Sigma_0} \omega. \tag{21}$$

If one requires $\theta$ to vanish on any global spacetime boundary (as is necessary to obtain a well-defined variational principle),[14] we can make the following observations. First, owing to the fact that $\mathrm{d}\omega \approx 0$, which follows from (6), $\Omega(\Sigma_0)$ is independent of the choice of Cauchy slice on-shell. Moreover, the balance relation (19) leads to the fundamental on-shell identity

$$-\hat{\xi} \cdot \Omega \approx \delta\int_{\partial\Sigma_0} q_\xi - \int_{\partial\Sigma_0} q_{\delta\xi}. \tag{22}$$

---

[13]This can be seen by integrating the relation $\mathrm{d}J_{f\xi} = \mathrm{d}C_{f\xi} + D_{f\xi}$, where $f$ is an arbitrary smooth function with compact support on $\mathcal{M}$. By Stokes' theorem, this results in the identity:

$$0 = \int_{\mathcal{M}} D_{f\xi} = \int_{\mathcal{M}} f D_\xi. \tag{15}$$

This holds for any $f$ if and only if $D_\xi \equiv 0$.

[14]Again, this may require the contribution of a suitable boundary action.

For field-independent vector fields ($\delta\xi = 0$), this tells us that $\hat{\xi}$ is the Hamiltonian vector field generated by the boundary charge $\int_{\partial\Sigma_0} q_\xi$. Equivalently, one says that $\hat{\xi}$ is integrable relative to $\Omega$. If $\xi$ is assumed to converge to zero sufficiently rapidly at the global boundary, the left-hand side of (22) actually vanishes, in which case $\hat{\xi}$ generates a gauge transformation.

## 3.2 Frame-dressed variational principle

The standard construction recalled in the previous subsection needs to be amended to accommodate the fact that the spacetime structures we are primarily interested in – such as the subregion $M$, its time-like boundary $\Gamma$ or the Cauchy slice $\Sigma \cup \bar{\Sigma}$ – are defined relative to the frame $U$, and are therefore field-dependent. This means, for instance, that a natural Ansatz for the action of the subregion $M$ is

$$S_M = \int_M L + \int_\Gamma \bar{\ell} = \int_m U^\star L + \int_\gamma \ell \,, \tag{23}$$

where $\ell$ is some boundary Lagrangian $(d-1,0)$-form intrinsic to $\gamma$ and $\bar{\ell} := U_\star \ell$. Given that both $m$ and $\gamma$ are background structures, this takes the form of a standard variational principle in the space of gauge-invariant observables $\mathfrak{m}$, with bulk Lagrangian $U^\star L$. We will refer to $U^\star L$, a $\text{Diff}(\mathcal{M})$ scalar, as the *dressed* or *invariant Lagrangian*. Once we realize that it is the natural bulk Lagrangian to consider, the construction of a canonical presymplectic structure for the subregion (together with a compatible boundary Lagrangian), can proceed in the same systematic manner as in [1]. The two main inputs of the algorithm outlined in this previous work, which focused on gauge theories defined on non-dynamical spacetime geometries, were indeed a gauge-invariant action principle,[15] together with a dynamical reference frame for the local gauge group of interest. From there, the derivation relied on a canonical frame-dressing of the form fields entering the covariant phase space formalism. We will follow the same strategy here, which will in particular lead to an additional frame-dressing of the presymplectic current derived from the invariant Lagrangian $U^\star L$.

Before elaborating on this second layer of frame-dressing in the next subsection, we introduce the main covariant phase space identities relevant for the analysis of $U^\star L$, which have already been extensively discussed in [23, 24]. Given the physical relevance of diffeomorphisms of $\mathfrak{m}$ (in contrast to spacetime diffeomorphisms, which are gauge), it is convenient to introduce a distinct notation for their associated field-space vector fields: given a vector field $\rho \in \mathfrak{X}(\mathfrak{m})$, we will follow [24] and denote by $\check{\rho}$ the associated field-space vector field. In particular, one has $\check{\rho} \cdot \delta g = 0$ and $\check{\rho} \cdot \delta[U^\star g] = \mathcal{L}_\rho[U^\star g]$.

To analyze variations of dressed forms such as $U^\star L$, it is useful to first understand the commutation relation between $\delta$ and $U^\star$. The latter can be conveniently expressed in terms of the *left-invariant Maurer-Cartan form* associated to the frame $U$:

$$\chi := \delta U^{-1} \circ U \,, \tag{24}$$

whose definition is described in more detail in Appendix A. This is a form of degree $(-1,1)$ on $\mathcal{M}$, or in other words, a $\mathfrak{X}(\mathcal{M})$-valued field-space one-form. From a physical point of view, it is a natural object to consider since it is invariant under a frame reorientation $U \to f \circ U$, where $f$ is a (field-independent) diffeomorphism of $m$. It is important to remember that $\chi$ is not in general closed. Rather, one can prove that

$$\delta\chi + \frac{1}{2}[\chi, \chi] = 0 \,, \tag{25}$$

---

[15]More precisely, the bulk Lagrangian had to be invariant under gauge transformations, up to boundary terms.

where $[\cdot,\cdot]$ denotes the Lie bracket on $\mathfrak{X}(\mathcal{M})$. Furthermore, given $\xi \in \mathfrak{X}(\mathcal{M})$ and $\rho \in \mathfrak{X}(\mathfrak{m})$, one has:

$$\hat{\xi} \cdot \chi = -\xi \qquad \text{and} \qquad \check{\rho} \cdot \chi = U_\star \rho \,. \tag{26}$$

Following [24, 65], it is also convenient to introduce the flat *field-space covariant derivative* canonically associated to $\chi$, which can be defined as:

$$\delta_\chi := \delta + \mathcal{L}_\chi \,, \tag{27}$$

where $\mathcal{L}_\chi$, a derivative of degree $(0,1)$, is defined through Cartan's formula: $\mathcal{L}_\chi := \mathrm{d}\chi_\lrcorner + \chi_\lrcorner\mathrm{d}$.[16] The relation (25) is nothing but a flatness condition for $\delta_\chi$, since it is equivalent to

$$\delta_\chi^2 = 0 \,. \tag{28}$$

With all of this in place, one can finally state the central commutation relation that will be repeatedly used in the rest of the text [10, 24]:

$$\delta(U^\star \alpha) = U^\star(\delta_\chi \alpha) \,. \tag{29}$$

For completeness, let us determine the transformation properties of $\chi$ under spacetime diffeomorphisms, as well as diffeomorphisms on the space of parameters. Given $\varphi \in \mathrm{Diff}(\mathcal{M})$, which we allow to be field-dependent, (4) and (24) imply that:

$$\varphi \rhd \chi = \varphi^{-1} \circ \chi \circ \varphi + \delta\varphi^{-1} \circ \varphi \,. \tag{30}$$

For a field-independent $\varphi$, we recover the transformation $\varphi \rhd \chi = \varphi_\star \chi$ expected of a spacetime vector-field,[17] and more generally, $\chi$ transforms as a field-space vector potential under a field-dependent gauge transformation. This implies that the covariant derivative $\delta_\chi = \delta + \mathcal{L}_\chi$ is indeed covariant, in the sense that:

$$\varphi_\star[(\delta + \mathcal{L}_\chi)\alpha] = (\delta + \mathcal{L}_{\varphi \rhd \chi})(\varphi_\star \alpha) \qquad \Leftrightarrow \qquad \varphi_\star[\delta_\chi \alpha] = (\varphi \rhd \delta_\chi)(\varphi_\star \alpha) \,, \tag{31}$$

for any local field form $\alpha$. In particular, this means that if $\alpha$ is covariant with respect to $\varphi$ (i.e. $\varphi \rhd \alpha = \varphi_\star \alpha$), then

$$\varphi \rhd (\delta_\chi \alpha) = (\varphi \rhd \delta_\chi)(\varphi \rhd \alpha) = \varphi_\star(\delta_\chi \alpha) \,, \tag{32}$$

from which we conclude that $\delta_\chi \alpha$ is also covariant with respect to $\varphi$.

At the infinitesimal level, equation (30) leads to

$$\Delta_\xi \chi = 0 \,, \tag{33}$$

for any field-dependent $\xi$. In other words, $\chi$ is anomaly-free. Likewise, (31) implies the commutation relation

$$[\delta_\chi, \mathsf{L}_{\hat{\xi}} - \mathcal{L}_\xi] = 0 \,. \tag{34}$$

As a result, if $\alpha$ is covariant with respect to $\xi$ (resp. generally covariant), then $\delta_\chi \alpha$ also is. By contrast, we note that $\delta_\chi$ does not in general commute with the anomaly operator $\Delta_\xi$. One can indeed prove the identities

$$[\delta, \Delta_\xi] = \Delta_{\delta\xi} \qquad \text{and} \qquad [\mathcal{L}_\chi, \Delta_\xi] = 0 \,, \tag{35}$$

---

[16]In other words, $\mathcal{L}_\chi$ acts as an anti-derivation with respect to the field-space wedge-product and as a derivation with respect to the spacetime wedge-product.

[17]Our notations are a bit formal here: $\varphi^{-1} \circ \chi$ should not be literally understood as a composition, which does not make mathematical sense, but rather as the composition with the *differential* of $\varphi^{-1}$ (which is the natural action induced by $\varphi^{-1}$ on a vector). With this understanding in mind, $\varphi^{-1} \circ \chi \circ \varphi$ is the pushforward of $\chi$ by $\varphi^{-1}$, or in other words, the pullback by $\varphi$.

from which it follows that

$$[\delta_\chi, \Delta_\xi] = \Delta_{\delta\xi}. \tag{36}$$

Finally, following [47], we remark that $\mathsf{L}_{\hat\xi} - \mathcal{L}_\xi$ can be interpreted as the covariant Lie derivative associated to $\chi$, defined as

$$\mathsf{L}^\chi_{\hat\xi} := [\delta_\chi, \widehat{\xi}\cdot]_+ = \mathsf{L}_{\hat\xi} - \mathcal{L}_\xi. \tag{37}$$

This provides an alternative explanation for the validity of equation (34).

Under a diffeomorphism $f \in \mathrm{Diff}(\mathfrak{m})$ on relational spacetime, $\chi$ transforms instead as:

$$f \rhd \chi = \chi + U^{-1} \circ (\delta f^{-1} \circ f) \circ U = \chi + U_\star[\delta f^{-1} \circ f]. \tag{38}$$

In particular, $\chi$ is invariant under any field-independent $f$, which can be interpreted as a frame reorientation. The general expression will be useful to determine how our formalism behaves under changes of frame.

Using (29), the variation of the invariant Lagrangian can be put in the form

$$\delta U^\star L = U^\star E + \mathrm{d} U^\star[\theta_\chi], \tag{39}$$

where, following [24], we define the *extended presymplectic potential* as

$$\theta_\chi := \theta + \chi \lrcorner L.^{18} \tag{40}$$

The basic variational identity (39) is the analogue of (6) in the frame-dressed setting, and demonstrates that $U^\star[\theta_\chi]$ is the presymplectic potential associated to the invariant Lagrangian $U^\star L$. Invoking (29) again, the resulting presymplectic current on relational spacetime can be expressed as

$$\delta U^\star[\theta_\chi] = U^\star[\omega_\chi], \tag{41}$$

in terms of the spacetime *extended presymplectic current*

$$\omega_\chi := \delta_\chi \theta_\chi. \tag{42}$$

As first proven in [11] and rederived in [23, 24], this current can be expressed as

$$\omega_\chi = \omega - \chi \lrcorner E + \mathrm{d}\left(\chi \lrcorner \theta + \frac{1}{2}\chi \lrcorner \chi \lrcorner L\right), \tag{43}$$

and only differs from $\omega$ by a boundary term on-shell. It was recently proposed by Ciambelli, Leigh and Pai in [23], and Freidel in [24], that

$$\Omega^{\mathrm{CLPF}}(\sigma) := \int_\sigma U^\star[\omega_\chi], \tag{44}$$

which on-shell only depends on the position of the corner $\partial\sigma$, provides a compelling presymplectic structure for the family of partial Cauchy slices associated to a fixed corner (or, equivalently, for the causal diamond associated to $\sigma$). The rationale behind this proposal is that (44) leads to a non-trivial representation of the so-called *extended corner algebra*. Following [48], the latter can be defined as a certain maximal subalgebra of spacetime vector fields associated to the corner $\partial\Sigma$, which, as its name indicates, can be seen as an extension of the so-called *corner algebra* first investigated in [10]. For any element $\xi$ of the extended corner algebra,

---

[18]In [24], this quantity is referred to as the *covariant presymplectic potential*. We adopt a different nomenclature here to avoid confusion with the notion of *U-covariance* we will develop in Sec. 3.3: we will indeed find that $\theta_\chi$ is *not* $U$-covariant in general.

the field-space vector field $\hat{\xi}$ turns out to be Hamiltonian relative to (44), and is furthermore generated by a non-trivial charge. This was seen in [23] as an advantage of the presymplectic structure (44) over earlier proposals: specifically, the presymplectic structure introduced by Donnelly and Freidel in [10] and generalized by Speranza in [11] does attribute integrable charges to spacetime diffeomorphisms, but these turn out to vanish. In [11], they vanish even off-shell, while in [10], they are only integrable for Lagrangians, like the one of vacuum general relativity, that vanish on-shell (see Sec. 7 for details). By contrast, in these latter works, the corner symmetry algebra came from diffeomorphisms in what we call relational spacetime.

As we will see in the following, the symplectic structure resulting from our algorithm is a partial return to the Donnelly–Freidel–Speranza (DFS) choice in the sense that all spacetime diffeomorphisms are once again pure gauge, however we find a nontrivial representation of the constraint algebra for arbitrary generally covariant theories. This is entirely consistent with the physical picture developed in Sec. 2. Indeed, at the global level, *any* vector field $\xi \in \mathfrak{X}(\mathcal{M})$ (that vanishes sufficiently rapidly at the global boundary) is by construction immaterial and therefore corresponds to a gauge direction. Restricting our attention to the subregion dynamics, and therefore to the action of such vector fields on the subregion $M \subset \mathcal{M}$, should not change this physical fact. Hence, whether $\xi$ is part of the extended corner subalgebra or not, it should better lie in the kernel of the regional presymplectic structure. By contrast, physical charges should be attributable to elements of $\mathfrak{X}(\mathfrak{m})$, since those vector fields directly act on physical observables. Indeed, we will find our systematic derivation of the presymplectic structure to be consistent with those expectations. We will furthermore show that the physical version of the extended corner algebra – one defined in terms of vector fields on $\mathfrak{m}$ rather than $\mathcal{M}$ – cannot be represented in full on the regional phase space, as one would physically expect, given that some relational diffeomorphisms amount to open system transformations (cf. Sec. 2.3). We will therefore find the CLPF presymplectic structure (44) to be unsatisfactory in our framework. It might still be relevant in other contexts, but our analysis suggests that this would require going beyond the framework of general covariance (see also the discussion in Sec. 7).

Coming back to the definition of an action of the form (23) for subregion $M$, we find upon variation that:

$$\delta S = \int_M E + \int_\Gamma \left( \theta_\chi + \delta_\chi \bar{\ell} \right) + \int_{\Sigma_2} \theta_\chi - \int_{\Sigma_1} \theta_\chi \,. \tag{45}$$

In this set-up, the Harlow-Wu consistency requirement [14] that any boundary condition on $\Gamma$ must obey is therefore:

$$\left( U^\star[\theta_\chi] + \delta\ell \right)\big|_\gamma \hat{\approx} \mathrm{d}\mathcal{C} \qquad \Leftrightarrow \qquad \left( \theta_\chi + \delta_\chi \bar{\ell} \right)\big|_\Gamma \hat{\approx} \mathrm{d}U_\star \mathcal{C} \,, \tag{46}$$

where $\mathcal{C}$ is some $(d-2, 1)$-form on $\gamma$. Here we introduce a second weak equality, which we will employ throughout this work: the symbol $\hat{\approx}$ entails equality when both the equations of motion and the boundary conditions have been imposed. A pullback to the relevant submanifold of solutions is furthermore implied when this equality involves forms of non-vanishing field-space degree, as is the case in equation (46). We will derive suitable $\ell$ and $\mathcal{C}$ for a wide class of boundary conditions in Sec. 6.

### 3.3 Canonical decomposition of a form relative to the frame

In analogy with our definition of the invariant Lagrangian $U^\star L$, given any tensor field $T$ on $\mathcal{M}$, we can define its invariant component relative to $U$ by the pushforward to $\mathfrak{m}$

$$T_{\mathrm{inv}} := U^\star T \,. \tag{47}$$

We would now like to extend this prescription to non-covariant objects, including arbitrary field-space forms. Consider a functional of the form $\alpha[g, \Psi]$, where $\Psi$ collectively denotes dynamical tensor fields other than the metric. Given a diffeomorphism $\varphi : \mathcal{M}' \to \mathcal{M}$, we can define the gauge-transformed functional on $\mathcal{M}'$ by:

$$\varphi \rhd (\alpha[g, \Psi]) := \alpha[\varphi_\star g, \varphi_\star \Psi] . \tag{48}$$

With this convention, $\rhd$ defines a left-action of field-independent diffeomorphisms: given two field-independent diffeomorphisms $\varphi_1 : \mathcal{M}' \to \mathcal{M}$ and $\varphi_2 : \mathcal{M}'' \to \mathcal{M}'$, we find that

$$(\varphi_1 \circ \varphi_2) \rhd (\alpha[g, \Psi]) = \varphi_2 \rhd (\varphi_1 \rhd (\alpha[g, \Psi])) . \tag{49}$$

In particular, whenever $\alpha[g, \Psi]$ is a tensorial expression, we recover $\varphi \rhd (\alpha[g, \Psi]) = \varphi_\star (\alpha[g, \Psi])$. Taking $\mathcal{M}' = \mathfrak{m}$ and $\varphi = U^{-1}$, we can then define the *invariant component of* $\alpha[g, \Psi]$ as

$$(\alpha[g, \Psi])_{\text{inv}} := U^{-1} \rhd (\alpha[g, \Psi]) = \alpha[U^\star g, U^\star \Psi] . \tag{50}$$

This name is justified since $\alpha_{\text{inv}}$, now an object on relational spacetime $\mathfrak{m}$, is invariant under a (field-dependent) spacetime diffeomorphism $\varphi : \mathcal{M} \to \mathcal{M}$:

$$\varphi \rhd (\alpha[g, \Psi])_{\text{inv}} = \alpha[U^\star \varphi^\star \varphi_\star g, U^\star \varphi^\star \varphi_\star \Psi] = \alpha[U^\star g, U^\star \Psi] = (\alpha[g, \Psi])_{\text{inv}} . \tag{51}$$

It is convenient at this stage to extend the definition of $U^\star$ and $U_\star$ to non-covariant objects by means of the standard coordinate expression. We can then introduce a canonical decomposition of $U^\star[\alpha]$ into *invariant and gauge components relative to* $U$:

$$U^\star[\alpha] = \alpha_{\text{inv}} + \alpha_{\text{gauge}} , \qquad \alpha_{\text{inv}} = U^{-1} \rhd \alpha , \qquad \alpha_{\text{gauge}} := U^\star[\alpha] - U^{-1} \rhd \alpha , \tag{52}$$

where, from now on, we lighten our notations by omitting explicit dependences in $(g, \Psi)$. The gauge component $\alpha_{\text{gauge}}$ transforms under gauge transformations in the same way as $U^\star[\alpha]$, hence its name. We will also find convenient to pull this decomposition back to $\mathcal{M}$. To this effect, we introduce the projector

$$\mathcal{P}_U \alpha := U_\star (U^{-1} \rhd \alpha) , \qquad \mathcal{P}_U^2 = \mathcal{P}_U . \tag{53}$$

With such definitions at our disposal, we can canonically decompose $\alpha$ into *covariant and non-covariant components relative to* $U$

$$\alpha = \alpha_U^{\text{c}} + \alpha_U^{\text{nc}} , \qquad \text{with} \qquad \alpha_U^{\text{c}} := \mathcal{P}_U \alpha , \qquad \alpha_U^{\text{nc}} := (1 - \mathcal{P}_U) \alpha , \tag{54}$$

which is nothing but the pullback of (52), given that $U^\star[\alpha_U^{\text{c}}] = \alpha_{\text{inv}}$ and $U^\star[\alpha_U^{\text{nc}}] = \alpha_{\text{gauge}}$. We will call any field-space form which lies in the image of $\mathcal{P}_U$ (or, equivalently, in the kernel of $1 - \mathcal{P}_U$) a *U-covariant* object. By construction, any $U$-covariant quantity $\alpha_U^{\text{c}}$ is also *covariant*, in the sense that $\varphi \rhd [\alpha_U^{\text{c}}] = \varphi_\star [\alpha_U^{\text{c}}]$ for any (field-dependent) spacetime diffeomorphism $\varphi$. At the infinitesimal level, this implies that

$$(\mathsf{L}_{\hat{\xi}} - \mathcal{L}_\xi) \alpha_U^{\text{c}} = 0 , \tag{55}$$

for any spacetime (and possibly field-dependent) vector field $\xi$, which indeed corresponds to the notion of general covariance we defined in Sec. 3.1. However, the converse is not true: not all covariant field-space expressions lie in the kernel of $1 - \mathcal{P}_U$. It is then clear that the

non-covariant component $\alpha_U^{\mathrm{nc}}$ vanishes when $\alpha$ is a tensor, in which case we recover definition (47), and otherwise measures the failure of $\alpha$ to transform as a spacetime tensor.[19]

The abstract construction we just outlined is general enough to apply to field-space scalars that do not transform covariantly, as well as to variations of tensorial quantities that nonetheless fail to be covariant due to the nontrivial commutator (29). The second situation is the most relevant one for our purpose, but for completeness, the covariant dressing of Christoffel symbols is described as an example of the first situation in Appendix B.1. The basic consequences of (50) that will prove most useful in our construction are:

$$(\alpha\beta)_{\mathrm{inv}} = \alpha_{\mathrm{inv}}\beta_{\mathrm{inv}}, \qquad (\delta\alpha)_{\mathrm{inv}} = \delta\alpha_{\mathrm{inv}}, \tag{56}$$

where $\alpha$ and $\beta$ are field-space forms of arbitrary degrees. The second relation is equivalent to $(\delta\alpha)_U^{\mathrm{c}} = \delta_\chi \alpha_U^{\mathrm{c}}$, and will guarantee consistency relations such as $\delta\theta_{\mathrm{inv}} = (\delta\theta)_{\mathrm{inv}} = \omega_{\mathrm{inv}}$.

One basic example of a non-tensorial quantity entering the covariant phase space formalism is the field-space derivative of the metric tensor $\delta g$. By (29) and (50), we find that:

$$(\delta g)_{\mathrm{inv}} = U^\star[\delta_\chi g] \neq U^\star[\delta g], \qquad (\delta g)_U^{\mathrm{c}} = \delta_\chi g \qquad \text{and} \qquad (\delta g)_U^{\mathrm{nc}} = -\mathcal{L}_\chi g. \tag{57}$$

The reason why $(\delta g)_U^{\mathrm{c}} \neq \delta g$ is that, even though $\delta g$ transforms as a tensor under *field-independent* diffeomorphisms of $\mathcal{M}$, it is not covariant under *field-dependent* diffeomorphisms. However, its covariant component $\delta_\chi g$ is. This observation applies to any field space one-form $\delta T$, where $T$ is a tensor: one then has $(\delta T)_U^{\mathrm{c}} = \delta_\chi T$ and $(\delta T)_U^{\mathrm{nc}} = -\mathcal{L}_\chi T$. Given (56), it is then straightforward to extend this discussion to arbitrary field-space forms involving tensors and their variations.

A field-space form of primary interest is the extended presymplectic potential $\theta_\chi$, whose invariant and covariant components relative to $U$ are determined to be

$$(\theta_\chi)_{\mathrm{inv}} = \theta_{\mathrm{inv}} = U^\star[\theta + \hat{\chi}\cdot\theta], \qquad (\theta_\chi)_U^{\mathrm{c}} = \theta_U^{\mathrm{c}} = \theta + \hat{\chi}\cdot\theta, \tag{58}$$

where $\hat{\chi}$ is the field-space vector field associated to $\mathcal{L}_\chi$ (and $\hat{\chi}\cdot$ thus defines a derivative of degree $(0,1)$). To see this, we first notice that $(\chi)_{\mathrm{inv}} = \delta(U^{-1} \rhd U)^{-1} \circ (U^{-1} \rhd U) = 0$ and therefore $(\theta_\chi)_{\mathrm{inv}} = \theta_{\mathrm{inv}}$. We can then express the potential in local Darboux coordinates as $\theta = \pi_a \delta\phi^a$, where both $\pi_a$ and $\phi^a$ are covariant tensors, to find that:

$$\theta_U^{\mathrm{c}} = \pi_a \delta_\chi \phi^a = \theta + \pi_a \mathcal{L}_\chi \phi^a = \theta + \hat{\chi}\cdot\theta. \tag{59}$$

As for the non-covariant component of $\theta_\chi$, it turns out to be equal to (minus) the Noether current $J_\chi := \hat{\chi}\cdot\theta - \chi \lrcorner L$ associated to the variational vector field $\chi$. In summary, we have obtained the following decomposition of $\theta_\chi$ relative to the frame:[20]

$$\boxed{\theta_\chi = (\theta_\chi)_U^{\mathrm{c}} + (\theta_\chi)_U^{\mathrm{nc}}, \quad \text{with} \quad (\theta_\chi)_U^{\mathrm{c}} = \theta_U^{\mathrm{c}} = \theta + \hat{\chi}\cdot\theta \quad \text{and} \quad (\theta_\chi)_U^{\mathrm{nc}} = -J_\chi.} \tag{60}$$

---

[19]It is perhaps illuminating to illustrate the previous definitions in the non-gravitational context. Following [1], the invariant part of a $\mathfrak{g}$-valued connection one-form $A$ relative to a $G$-valued reference frame $U$ is $A_{\mathrm{inv}} = U^{-1} \rhd A = U^{-1}AU - \mathrm{d}U^{-1}U$. The quantity $A_U^{\mathrm{c}} := UA_{\mathrm{inv}}U^{-1} = A - U\mathrm{d}U^{-1}$ is then covariant, in the sense that a gauge transformation acts on it by the adjoint $g \rhd A_U^{\mathrm{c}} := gA_U^{\mathrm{c}}g^{-1}$. The analogue of $U^\star$ in this context is therefore the adjoint action $A \mapsto U^{-1}AU$. It is also clear that the quantity $A_U^{\mathrm{nc}} := A - A_U^{\mathrm{c}} = U\mathrm{d}U^{-1}$ transforms in the same way as $A$ under a gauge transformation, and is in this sense non-covariant: $g \rhd A_U^{\mathrm{nc}} = gA_U^{\mathrm{nc}}g^{-1} + g\mathrm{d}g^{-1}$. We finally note that [1] relied on a different (but however consistent) definition of $A_{\mathrm{gauge}}$, namely $A_{\mathrm{gauge}} = A - A_{\mathrm{inv}}$ instead of $A_{\mathrm{gauge}} = U^{-1}AU - A_{\mathrm{inv}}$. The reason we depart from this definition here is that it is not very natural in the active treatment of the diffeomorphism gauge symmetry we adopted in the present work.

[20]Note that, while the standard and extended potentials have the same covariant component ($\theta_U^{\mathrm{c}} = (\theta_\chi)_U^{\mathrm{c}}$), their non-covariant parts differ: $\theta_U^{\mathrm{nc}} \neq (\theta_\chi)_U^{\mathrm{nc}}$.

We also note that the fundamental variational identity (39) can be reexpressed in terms of frame-covariant quantities as:

$$(\delta L)_U^{\mathrm{c}} = E_U^{\mathrm{c}} + \mathrm{d}\theta_U^{\mathrm{c}}, \quad (\delta L)_U^{\mathrm{c}} = \delta_\chi L, \quad E_U^{\mathrm{c}} = E + \hat{\chi} \cdot E, \quad \theta_U^{\mathrm{c}} = \theta + \hat{\chi} \cdot \theta. \tag{61}$$

Given that $E_U^{\mathrm{c}} \approx 0$, $\theta_U^{\mathrm{c}}$ is a legitimate choice of presymplectic potential, but is better behaved than $\theta_\chi$ in the sense that $\varphi \rhd \theta_U^{\mathrm{c}} = \varphi_\star \theta_U^{\mathrm{c}}$, even for a field-dependent diffeomorphism $\varphi$. As an aside, we note for later use that, because $\chi_U^{\mathrm{c}} = \chi + \hat{\chi} \cdot \chi$ must identically vanish, we must also have $\hat{\chi} \cdot \chi = -\chi$.

Turning to the presymplectic current $\omega_\chi$, we find by applying $\delta_\chi$ to (60) that

$$\omega_\chi = \delta_\chi(\theta_\chi)_U^{\mathrm{c}} + \delta_\chi(\theta_\chi)_U^{\mathrm{nc}} = (\delta\theta_\chi)_U^{\mathrm{c}} - \delta_\chi J_\chi. \tag{62}$$

In addition, we also know from $U^{-1} \rhd \chi = 0$ that

$$(\omega_\chi)_U^{\mathrm{c}} = (\delta\theta_\chi)_U^{\mathrm{c}} = (\delta\theta)_U^{\mathrm{c}} = \omega_U^{\mathrm{c}}. \tag{63}$$

Altogether, we therefore obtain the following decomposition of $\omega_\chi$ relative to $U$:

$$\boxed{\omega_\chi = (\omega_\chi)_U^{\mathrm{c}} + (\omega_\chi)_U^{\mathrm{nc}}, \qquad \text{with} \qquad (\omega_\chi)_U^{\mathrm{c}} = \omega_U^{\mathrm{c}} \qquad \text{and} \qquad (\omega_\chi)_U^{\mathrm{nc}} = -\delta_\chi J_\chi.} \tag{64}$$

By invoking local Darboux coordinates, one can also show that, for a two-form like $\omega$, one has the general formula:

$$\omega_U^{\mathrm{c}} = \omega + \frac{3}{2}\hat{\chi} \cdot \omega + \frac{1}{2}\hat{\chi} \cdot \hat{\chi} \cdot \omega. \tag{65}$$

See Appendix B.2 for a proof.

We conclude this section by noting that, for any $(p,q)$-form $\alpha$, and any (possibly field-dependent) spacetime vector field $\xi$:

$$\boxed{\hat{\xi} \cdot \alpha_U^{\mathrm{c}} = 0} \tag{66}$$

In other words, we have the operator identity $\hat{\xi} \cdot \mathcal{P}_U = 0$. This observation will play an important role in Sec. 5. It is trivially true when $\alpha$ is a field-space scalar ($q = 0$). For exact one-forms such as $\delta g$, it is a direct consequence of (27) and (26):

$$\hat{\xi} \cdot (\delta g)_U^{\mathrm{c}} = \hat{\xi} \cdot \delta_\chi g = \hat{\xi} \cdot (\delta + \mathcal{L}_\chi)g = \mathcal{L}_\xi g - \mathcal{L}_\xi g = 0. \tag{67}$$

More generally, when $q = 1$, we can use the general expression in (59) to explicitly check that:

$$\hat{\xi} \cdot \alpha_U^{\mathrm{c}} = \hat{\xi} \cdot (\alpha + \hat{\chi} \cdot \alpha) = \hat{\xi} \cdot \alpha - \hat{\xi} \cdot \alpha = 0, \tag{68}$$

where we have used $[\hat{\xi}\cdot, \hat{\chi}\cdot] = -\hat{\xi}\cdot$. Similarly, when $q = 2$, we can explicitly check the validity of (66) by means of (65); see Appendix B.2. Finally, by virtue of (56), relation (66) must in fact hold for any value of $q$.

Together with (55), the identity (66) implies that

$$\Delta_\xi \alpha_U^{\mathrm{c}} = 0 \tag{69}$$

for any field-dependent vector-field $\xi$. In words, the $U$-covariant component of a form is not only generally covariant (as we saw in (55)) but also anomaly-free.

### 3.4 (Non-)covariance in relational spacetime: material vs. immaterial frames

Let us briefly consider the covariance properties of frame-dressed observables under diffeomorphisms of relational spacetime $\mathfrak{m}$. As a special case, we will then consider the relational covariance of the frame-dressed bulk Lagrangian $U^\star L$ and what this entails for the charges of the theory.

Suppose $\alpha$ is some field-space $(p,q)$-form on $M$ and $\rho$ some (possibly field-dependent) vector field on $\mathfrak{m}$. Then, using (10) and (29), we have

$$\Delta_\rho U^\star \alpha = \check{\rho} \cdot U^\star[\delta + \mathcal{L}_\chi]\alpha + \delta\check{\rho} \cdot U^\star \alpha - U^\star \mathcal{L}_{U_\star \rho}\alpha - \widetilde{\delta\rho} \cdot U^\star \alpha. \tag{70}$$

Now assume $\alpha$ is independent of the frame $U$ and its variations, so that it is oblivious to what happens in relational spacetime $\mathfrak{m}$. In this case, we find

$$\check{\rho} \cdot \alpha = 0, \qquad \check{\rho} \cdot \delta\alpha = 0, \qquad \widetilde{\delta\rho} \cdot \alpha = 0. \tag{71}$$

As a consequence, we also have $\check{\rho} \cdot U^\star \mathcal{L}_\chi \alpha = U^\star \mathcal{L}_{U_\star \rho}\alpha$ and, altogether, those identities allow to infer from (70) that, for all field-dependent $\rho \in \mathfrak{X}(\mathfrak{m})$,

$$\Delta_\rho U^\star \alpha = 0. \tag{72}$$

Hence, the relational observable $U^\star \alpha$ is anomaly-free in relational spacetime $\mathfrak{m}$.

Specifically, if the bulk spacetime Lagrangian $L$ is independent of the frame $U$, as is the case for the immaterial geodesic frames (cf. Sec. 2.4), we find that the dressed Lagrangian of the variational principle in $\mathfrak{m}$ is relationally generally covariant in the sense that $\Delta_\rho U^\star L = 0$ for all $\rho \in \text{Diff}(\mathfrak{m})$. This means that any field-space vector field $\check{\rho}$ corresponding to a relational diffeomorphism $\rho$ should be tangential to the space of solutions. However, this does *not* mean that diffeomorphisms of relational spacetime $\mathfrak{m}$ are gauge and neither that all of them correspond to Hamiltonian charges, both of which depend on the regional presymplectic structure. Indeed, after constructing a dynamically preserved regional presymplectic structure in the next section, we will see in Sec. 5 that a class of the relational diffeomorphisms correspond to non-trivial Hamiltonian charges, while others constitute non-degenerate, yet non-integrable directions and, depending on the boundary conditions, another set may in principle turn into gauge directions (even though the latter situation will not occur in the examples discussed in Sec. 6.3). This is consistent with the fact that the map $U : \mathcal{M} \to \mathfrak{m}$ defined by the frame constitutes a deparametrization of the spacetime theory.

By contrast, if $\alpha$ depends on the frame $U$, we can at best hope to find (72) for special relational diffeomorphisms $\rho$. For example, in Sec. 2.4, we have seen that if we set $\alpha = L_{\text{dust}}$, where $L_{\text{dust}}$ is the Lagrangian associated with the dust frame, we find the conditions in (71) satisfied—and hence $\Delta_\rho U^\star L_{\text{dust}} = 0$—*only* for diffeomorphisms of $\mathfrak{m}$ that preserve the dust four-velocity $V$ [78]. For other relational diffeomorphisms we have $\Delta_\rho U^\star L_{\text{dust}} \neq 0$, so that in this matter frame case the variational principle on $\mathfrak{m}$ is not fully relationally generally covariant. In particular, only the relational diffeomorphisms preserving $V$ correspond to field-space vector fields $\check{\rho}$ tangential to the space of solutions.

## 4 Regional presymplectic structure relative to the frame

We now turn to the problem of identifying the full presymplectic structure for a subregion theory, relative to a choice of frame. To achieve this, we implement the process of "splitting post-selection" outlined in [1], here adapted to gravitational theories where the gauge group is general diffeomorphisms. The goal of the post-selection process is to generate consistent

and self-contained subregion theories via restriction from a more global theory in a systematic manner. Decomposition of the global theory into theories on complementary subregions results in a whole family of subregion theories, each appropriate to different physical conditions at the interface (which enter the subregion theory as boundary conditions).

We will illustrate the post-selection procedure in two stages, here at the level of symplectic structure, and later at the level of actions (in Sec. 6). In the former case, it is the global symplectic structure itself that is factorized, on shell of both the equations of motion and a particular choice of boundary condition. Under some minor assumptions, the subregion symplectic structure is uniquely inherited from that of the global theory. In particular, we assume that the subregion bulk symplectic structure is locally identical to that of the global theory, so that it takes the same form as worked out previously ($\theta_\chi$ in equation (40) and $\omega_\chi$ in equation (43), as well as the associated decompositions (60) and (64)). It only remains to identify possible boundary contributions. To proceed, our primary guiding principle is that the total subregion symplectic structure should be conserved (i.e. that it is independent of any choice of Cauchy slice), whenever gauge-invariant boundary conditions are imposed. This strongly restricts the boundary contribution, and in fact has the additional implication that the resulting symplectic form is invariant under field-dependent gauge transformations. This is an illuminating feature which is particularly clear in this context: the extension of the standard gauge-invariance condition to include field-dependent gauge transformations can be motivated not only as an aesthetically pleasing feature, but a requirement owing to the conservation of symplectic structure under gauge-invariant boundary conditions.

In Sec. 6, we will return to the idea of splitting post-selection as a means of generating subregion actions. Analogous to the process just described for symplectic structure, the action of the global theory is decomposed into pieces associated with a subregion and its complement. Assuming a local equivalence between bulk actions of the global and subregion theories, the decomposition is obvious up to the inclusion of boundary terms. These are chosen to achieve two goals: to dynamically impose the boundary conditions and to enforce the conservation of symplectic structure just described. The former may be achieved through the inclusion of Lagrange multiplier terms on the boundary $\Gamma$. To meet the latter, we will apply criteria ascertained in this section from consideration of symplectic structure alone. This is one reason for considering the post-selection process in two stages; it allows us to conceptually disentangle these distinct roles for the boundary action.

As described in Sec. 2.2, we take the subregion of interest, which we call $m$, to be invariantly defined on relational spacetime $\mathfrak{m}$. The region and its bounding surfaces are illustrated in figure 4. We take it to have the topology of a cylinder. It has time-like boundary $\gamma$, and is bounded to the past and future by partial Cauchy surfaces $\sigma_1$ and $\sigma_2$, respectively. These surfaces can be arbitrarily extended into the complement region, forming complete Cauchy surfaces $\sigma_1 \cup \bar{\sigma}_1$ and $\sigma_2 \cup \bar{\sigma}_2$. We refer to the complement region between $\bar{\sigma}_1$ and $\bar{\sigma}_2$ as $\bar{m}$, and to the union $\mathfrak{m} = m \cup \bar{m}$ as the "global" relational spacetime (excluding, as a matter of terminology, any region not between these Cauchy surfaces). Each of these relational spacetime subregions/submanifolds may be pulled back by $U$ to corresponding subregions/submanifolds on spacetime $\mathcal{M} = U^{-1}(\mathfrak{m})$. We denote these spacetime manifolds with the corresponding upper-case letter:

$$M = U^{-1}(m), \ \bar{M} = U^{-1}(\bar{m}), \ \Gamma = U^{-1}(\gamma), \ \Sigma_i = U^{-1}(\sigma_i), \ \bar{\Sigma}_i = U^{-1}(\bar{\sigma}_i) \ \text{for} \ i = 1, 2. \quad (73)$$

To proceed we first assume that the global theory is already well-defined, in the sense that a global symplectic form $\Omega_{\mathfrak{m}}$ is known, and if necessary suitable boundary conditions have been imposed on some more distant, possibly asymptotic, boundary.

Let the space of solutions to the global theory be denoted by $\mathcal{S}$. We consider a foliation of $\mathcal{S}$ such that each leaf corresponds to a distinct choice of boundary conditions on $\gamma$. To

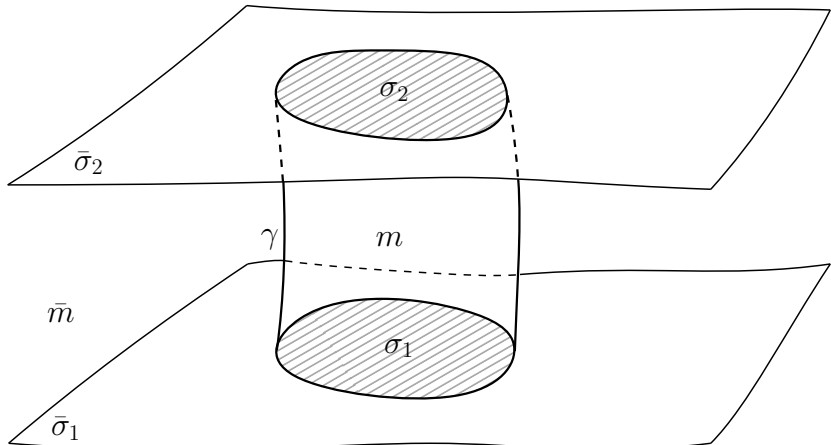

Figure 4: A subregion $m$ on relational spacetime, is delineated by boundaries labelled as shown.

clearly distinguish gauge conditions from physical boundary conditions, we require the latter to be gauge-invariant. In the context of diffeomorphisms, this is in large part a statement about the specification of the boundary itself. The timelike boundary $\Gamma$ dividing $M$ from $\bar{M}$ on spacetime is, by construction, related to the invariant submanifold $\gamma$ in relational spacetime via $\gamma = U(\Gamma)$. A gauge-invariant boundary condition is then ultimately a restriction on a $U$-covariant combination of fundamental fields and their derivatives, pushed forward to $\mathfrak{m}$ and pulled back to $\gamma$. We denote such an invariant combination as $x^a$ (collectively $x$), and assume boundary conditions of the form $x^a = x_0^a$ (collectively $x = x_0$). This can equivalently be written in terms of spacetime fields on $\Gamma$ by simply defining $X = U_\star(x)$ and $X_0 = U_\star(x_0)$, and imposing $X = X_0$. A key difference is that while $x_0$ is taken to be a background structure in the variation, such that $\delta x_0 = 0$, the corresponding condition imposed on $X_0$ is that $\delta_\chi X_0 = 0$. The foliation of the global solution space can then be written

$$\mathcal{S} = \bigsqcup_{x_0} \mathcal{S}_{x_0}, \quad \mathcal{S}_{x_0} = \{\Phi \in S | x = x_0\}. \tag{74}$$

Of course, the boundary condition $x = x_0$ (or $X = X_0$) must also be suitable to the dynamical problem on the subregion: it must be restrictive enough for a well-posed dynamical problem, but not so restrictive that it does not allow the free specification of initial data on an interior Cauchy slice.

We will restrict the physical symplectic flux through $\gamma$ by requiring that $(\omega_\chi)_{\text{inv}}$, after pulling back to $\gamma$, is of the form

$$(\omega_\chi)_{\text{inv}}\big|_\gamma \hat{\approx} \mathrm{d}\delta\beta_{\text{inv}}. \tag{75}$$

for some invariant $(d-2,1)$ form $\beta_{\text{inv}}$ on $\gamma$. Remember that the notation $\hat{\approx}$ indicates that equality holds when both equations of motion and boundary conditions have been imposed, i.e. within the restricted space $\mathcal{S}_{x_0}$ of solutions. In particular, the field-space two-forms appearing in (75) are pulled back to $\mathcal{S}_{x_0}$. This can equivalently be written on spacetime, as a restriction on the $U$-covariant part of the symplectic current $(\omega_\chi)_U^{\mathrm{c}}$ on $\Gamma$:

$$(\omega_\chi)_U^{\mathrm{c}}\big|_\Gamma \hat{\approx} \mathrm{d}\delta_\chi \beta_U^{\mathrm{c}}. \tag{76}$$

The condition (75) may appear as a generalization of that considered in [1], where it was required that the analogous quantity in various gauge theories vanishes on $\Gamma$, forcing the physical (gauge-invariant) part of the symplectic flux through the boundary to vanish. Condition (75) is analogous but allows for a nonvanishing contribution of a form which could

be understood as a shift in the symplectic potential by an exact, $U$-covariant term. As a result, allowing for a non-vanishing flux as appearing on the right-hand side of (75) has no physical implications; it is only a matter of convenience.

With this condition in place, we proceed to seek the appropriate symplectic structure $\Omega_m$ of the subregion $m$ via decomposition of the global presymplectic structure $\Omega_{\mathfrak{m}}$ of $\mathfrak{m} = m \cup \bar{m}$. We first assume that these are locally identical to $\Omega_{\mathfrak{m}}$ in the bulk of $m$ and $\bar{m}$, and that the decomposition is additive. This fixes $\Omega_m$ and $\Omega_{\bar{m}}$ up to possible equal and opposite boundary terms on $\partial\sigma$. Expressed in terms of integrals on $\mathfrak{m}$, the decomposition is

$$\Omega_{\mathfrak{m}} \mathrel{\hat{\approx}} \Omega_m + \Omega_{\bar{m}}\,, \qquad \text{with} \qquad \begin{cases} \Omega_m \mathrel{\hat{\approx}} \displaystyle\int_\sigma U^\star[\omega_\chi] + \int_{\partial\sigma} \omega_\partial \\[1em] \Omega_{\bar{m}} \mathrel{\hat{\approx}} \displaystyle\int_{\bar{\sigma}} U^\star[\omega_\chi] - \int_{\partial\sigma} \omega_\partial + \Omega_{\partial(\sigma\cup\bar{\sigma})} \end{cases}. \tag{77}$$

Here, $\omega_\partial$ is a $(d-2,2)$-form on $\gamma$, possibly comprised of both bulk and boundary fields. We have allowed for the possibility of boundary contributions to the global symplectic form through $\Omega_{\partial(\sigma\cup\bar{\sigma})}$, though the details of this structure are hereafter not important. We also suppose that $\omega_\partial$ can be taken to originate from a boundary term in the presymplectic potential of $m$, that is $\omega_\partial = \delta\theta_\partial$ for some $(d-2,1)$-form $\theta_\partial$. With this definition at hand, we can introduce the following off-shell forms:

$$\Theta(\sigma) := \int_\sigma U^\star[\theta_\chi] + \int_{\partial\sigma} \theta_\partial\,, \qquad \Omega(\sigma) := \delta\Theta(\sigma)\,, \tag{78}$$

such that

$$\Omega_m := \Omega(\sigma)\big|_{\mathcal{S}_{x_0}}\,. \tag{79}$$

The requirement that $\Omega_m$ is independent of the choice of Cauchy slice $\sigma$ strongly restricts the form of $\omega_\partial$, and thereby $\theta_\partial$.[21] In particular, using the fact that $d\omega_\chi \approx 0$, this condition entails that

$$U^\star[\omega_\chi]\big|_\gamma \mathrel{\hat{\approx}} -d\omega_\partial\,. \tag{80}$$

The $U$-covariant part of the symplectic flux is already restricted in equation (76). And as we have seen in equation (64), for the covariant Lagrangians we are considering, $(\omega_\chi)_U^{\mathrm{nc}} = -\delta_\chi J_\chi \approx -d\delta_\chi q_\chi$. We can then rewrite (80) as

$$d\delta U^\star[q_\chi|_\Gamma - \beta_U^{\mathrm{c}}] \mathrel{\hat{\approx}} d\omega_\partial\,. \tag{81}$$

Inserting $\omega_\partial = \delta\theta_\partial$ into equation (81) this condition can be solved by defining $\theta_\partial$ as

$$\boxed{\theta_\partial = U^\star[q_\chi|_\Gamma - \beta_U^{\mathrm{c}}] + \delta\ell_\partial} \tag{82}$$

for some $(d-2,0)$ form $\ell_\partial$ on $\gamma$, and

$$\boxed{\omega_\partial = \delta U^\star[q_\chi|_\Gamma - \beta_U^{\mathrm{c}}]}\,. \tag{83}$$

In writing these expressions we have ignored possible spacetime-exact terms, as they will not contribute to the symplectic structure upon integration over $\partial\sigma$. Equation (83) guarantees the conservation of the subregion symplectic structure, in the sense that

$$\Omega(\sigma) := \int_\sigma U^\star[\omega_\chi] + \int_{\partial\sigma} \omega_\partial \mathrel{\hat{\approx}} \Omega_m \tag{84}$$

---

[21]In the present section we are primarily interested in defining the on-shell quantity $\Omega_m$, but we will here define $\theta_\partial$ and $\omega_\partial$ off-shell, completing the definition of $\Theta(\sigma)$ and $\Omega(\sigma)$ as off-shell forms. The on-shell restriction of $\Omega(\sigma)$ then defines $\Omega_m$, which is required to be independent of the choice of slice $\sigma$, whereas the on-shell restriction of $\Theta(\sigma)$ will still depend on this choice.

is independent, fully on-shell, of the choice of $\sigma = U(\Sigma)$ in these integrals. In the remainder of this text, we will denote the resulting on-shell presymplectic structure for subregion $m$ by $\Omega_m$ and reserve the notation $\Omega(\sigma)$ for off-shell computations. We also note that the final expression of the presymplectic form (84) is independent of any specific choice of boundary condition (see Sec. 7.3 for a discussion of this point). Investigating how to turn the subregion into a closed subsystem turned out to be a powerful guiding principle, but as we will illustrate below, the resulting presymplectic structure is in a sense universal. In particular, it is equally relevant to the analysis of a fixed corner $\partial\sigma$ in the absence of any boundary condition: in this context, it can be used to correctly discriminate between gauge and physical transformations acting on the corner, as was first observed in [10] (using a slightly different presympletic structure) and will be described again in Sec. 5.

## 5 Gauge transformations and symmetries

We can now determine the subalgebras of $\mathfrak{X}(\mathcal{M})$ and $\mathfrak{X}(\mathfrak{m})$ that can be represented by Hamiltonian vector fields relative to the regional presymplectic form $\Omega(\sigma) \,\hat{\approx}\, \Omega_m$, as defined in (84), and compute the respective Poisson algebras generated by those Hamiltonian vector fields. Consistently with our general expectations, we will find out that: 1) any spacetime diffeomorphism is integrable and also null with respect to $\Omega(\sigma)$ on-shell of the bulk equations of motion, and therefore generates a gauge transformation; 2) only a subalgebra of $\mathfrak{X}(\mathfrak{m})$ gives rise to integrable charges, and those are non-trivial in general. Furthermore, once boundary conditions are taken into consideration, we will find that both the integrable character of an element of $\mathfrak{X}(\mathfrak{m})$ and its physical interpretation (e.g. as a gauge transformation or a non-trivial symmetry transformation) may explicitly depend on the choice of boundary conditions.

### 5.1 Algebra of spacetime diffeomorphisms

To determine the Hamiltonian charges associated to spacetime diffeomorphisms and their algebra, we are interested in computing $-\hat{\xi} \cdot \Omega(\sigma)$ *off-shell*, for a *field-independent* $\xi$. In particular, we need to pick up a preferred $\sigma$ in order to run this computation, since $\Omega(\sigma)$ is not conserved off-shell. Using (64) and (66), we immediately obtain

$$-\hat{\xi} \cdot \omega_\chi = -\hat{\xi} \cdot (\omega_\chi)^{\text{nc}}_U = \delta_\chi(C_\xi + \mathrm{d}q_\xi) \approx \mathrm{d}(\delta_\chi q_\xi). \tag{85}$$

As a result, in terms of the CPLF presymplectic two-from (equation (44)), we find that

$$-\hat{\xi} \cdot \Omega^{\text{CPLF}}(\sigma) = \delta\tilde{H}_\xi(\sigma), \quad \text{where} \quad \tilde{H}_\xi(\sigma) := \int_\sigma U^\star[C_\xi + \mathrm{d}q_\xi] \approx \int_{\partial\sigma} U^\star[q_\xi] = \int_{\partial\Sigma} q_\xi. \tag{86}$$

Hence, $\Omega^{\text{CPLF}}(\sigma)$ assigns a non-trivial boundary charge to any $\xi$. Including the boundary contribution $\int_{\partial\sigma} \omega_\partial$ into the presymplectic structure preserves integrability, but turns any $\hat{\xi}$ into a gauge direction. To see this, we first compute:

$$-\hat{\xi} \cdot \delta_\chi(q_\chi - \beta^c_U) = \delta_\chi\hat{\xi} \cdot (q_\chi - \beta^c_U) - \Delta_\xi(q_\chi - \beta^c_U) = -\delta_\chi(q_\xi + \hat{\xi} \cdot \beta^c_U) = -\delta_\chi q_\xi, \tag{87}$$

where we have used $[\hat{\xi}\cdot, \delta_\chi]_+ = \mathsf{L}_{\hat{\xi}} - \mathcal{L}_\xi = \Delta_\xi$, $\hat{\xi} \cdot q_\chi = -q_\xi$ (which follows from the fact that $q_\chi$ is a field-space one-form linear in $\chi$), the fact that $q_\chi$ and $\beta^c_U$ are anomaly-free,[22] as well

---

[22] The anomaly-freeness of $q_\chi$ follows from the fact that $\chi$ is itself anomaly-free, as stated in equation (33). The anomaly-freeness of $\beta^c_U$ follows from its $U$-covariance, and the fact that $\Gamma$ is defined through its relation to the invariant boundary $\gamma$ on $\mathfrak{m}$, via $\gamma = U(\Gamma)$ – in other words, it is crucial that $\Gamma$ is itself a covariantly defined submanifold. See Appendix C for a discussion of this point.

as $\hat{\xi} \cdot \beta_U^c = 0$. It follows that

$$-\hat{\xi} \cdot \Omega(\sigma) = \delta H_\xi(\sigma), \qquad \text{where} \qquad H_\xi(\sigma) = \int_\sigma U^\star[C_\xi] \approx 0. \tag{88}$$

As anticipated, a field-independent $\xi$ is integrable with respect to $\Omega(\sigma)$, irrespective of how it behaves close to the boundary $\gamma$, and the resulting diffeomorphism charge $H_\xi(\sigma)$ vanishes on-shell of the bulk equations of motion. Note that the $U$-covariance of $\beta_U^c$ played a crucial role leading to this result. If this were instead some arbitrary field-space one-form $\beta$, $H_\xi$ would have received a correction of the form $\int_{\partial\sigma} U^\star[\hat{\xi} \cdot \beta]$, which in general does not vanish on-shell.

As a side remark, we note that for a *field-dependent* $\xi$, we find the more general relation

$$-\hat{\xi} \cdot \Omega(\sigma) = \delta \left( \int_\sigma U^\star[C_\xi] \right) - \int_\sigma U^\star[C_{\delta\xi}] \approx 0, \tag{89}$$

which in turn implies that $\Omega(\sigma)$ is invariant under $\hat{\xi}$ on-shell:

$$\mathsf{L}_{\hat{\xi}}\Omega(\sigma) = \delta(\hat{\xi} \cdot \Omega(\sigma)) = \delta \left( \int_\sigma U^\star[C_{\delta\xi}] \right) \approx 0. \tag{90}$$

From the point of view of the original edge mode derivation of [10], this invariance property should be understood as a primary postulate *defining* $\Omega(\sigma)$. By contrast, it is conceptually secondary in our approach, but nonetheless a necessary consequence of our construction.

Let us finally determine the off-shell algebra of spacetime diffeomorphism charges induced by $\Omega(\sigma)$. We define the kinematical Poisson bracket associated to the Cauchy slice $\sigma$ in the usual way:

$$\{H_{\xi_1}(\sigma), H_{\xi_2}(\sigma)\} := \hat{\xi}_1 \cdot \delta H_{\xi_2}(\sigma) = -\hat{\xi}_1 \cdot \hat{\xi}_2 \cdot \Omega(\sigma). \tag{91}$$

To determine this algebra, we first remark that $\Delta_{\xi_1} C_{\xi_2} = C_{-\mathcal{L}_{\xi_1}\xi_2} = -C_{[\xi_1,\xi_2]}$, which directly follows from the fact that $C_\xi$ depends linearly on the background vector field $\xi$, while every other field entering its definition is covariant. We can then compute:

$$\begin{aligned}
\{H_{\xi_1}(\sigma), H_{\xi_2}(\sigma)\} &= \hat{\xi}_1 \cdot \delta \int_\sigma U^\star[C_{\xi_2}] = \hat{\xi}_1 \cdot \int_\sigma U^\star[\delta_\chi C_{\xi_2}] \\
&= \hat{\xi}_1 \cdot \int_\Sigma \left( \delta + \mathcal{L}_\chi \right) C_{\xi_2} = \int_\Sigma \left( \hat{\xi}_1 \cdot \delta - \mathcal{L}_{\xi_1} \right) C_{\xi_2} \\
&= \int_\Sigma \left( \mathsf{L}_{\hat{\xi}_1} - \mathcal{L}_{\xi_1} \right) C_{\xi_2} = \int_\Sigma \Delta_{\xi_1} C_{\xi_2} = -\int_\Sigma C_{[\xi_1,\xi_2]} \\
&= -H_{[\xi_1,\xi_2]}(\sigma).
\end{aligned} \tag{92}$$

In other words, the spacetime diffeomorphism charges form the first-class constraint algebra:

$$\{H_{\xi_1}(\sigma), H_{\xi_2}(\sigma)\} = -H_{[\xi_1,\xi_2]}(\sigma), \tag{93}$$

which in particular is anti-homomorphic to the algebra of spacetime vector fields. The latter is thus properly represented on the covariant phase space in terms of the constraints of the theory and without central extension. In this regard, our construction differs from previous ones [10, 11, 23, 24, 48], as we discuss in more detail in Sec. 7.1.

We thus see that the inclusion of the embedding fields into the regional phase space leads in our presymplectic structure to a Poisson bracket representation of the Lie algebra of gauge diffeomorphisms regardless of how they act near the boundary. In gauge-fixed constructions without edge modes this is not possible as diffeomorphisms with support on the boundary are

no longer gauge and only diffeomorphisms which vanish there can be properly represented. This resembles somewhat the situation in canonical geometrodynamics where only the action of diffeomorphisms that preserve the spacelike nature of the Cauchy slice is defined on the standard ADM phase space [3, 72, 73]. Of those, only the spatial diffeomorphisms (i.e. the ones tangential to the slice) come with a proper Poisson bracket representation, while the full set of spacelike preserving diffeomorphisms gives rise to the Dirac hypersurface deformation algebroid. As Isham and Kuchař showed in [73], if one extends the standard ADM phase space by the embedding fields that describe the Cauchy slices, one obtains a proper Poisson bracket representation of the *full* Lie algebra of spacetime diffeomorphisms also in canonical geometrodynamics (in the absence of boundaries). Our construction gives rise to essentially the same feature in the covariant phase space description of spacetime subregions.

## 5.2 Boundary symmetries: integrability and charges

Let us now turn to the analysis of frame reorientations and their charges. In this discussion, we shall assume the frame $U$ to correspond to a non-local immaterial frame (as the geodesic frames in Sec. 2). This will entitle us to assume all frame-dressed quantities below to be anomaly-free on relational spacetime, see Secs. 2.4 and 3.4, which will simplify the analysis substantially.[23] We will also restrict our attention to *field-independent* diffeomorphisms on relational spacetime, for which the notions of anomaly-freeness and covariance coincide.

Given a field-independent vector field $\rho \in \mathfrak{X}(\mathfrak{m})$, we start by computing the contraction of $\check{\rho}$ with the symplectic form:

$$-\check{\rho} \cdot \Omega(\sigma) = -\check{\rho} \cdot \left( \int_\sigma U^\star[\delta_\chi \theta_\chi] + \int_{\partial\sigma} U^\star[\delta_\chi(q_\chi - \beta_U^c)] \right) \tag{94}$$

$$= -\check{\rho} \cdot \delta \left( \int_\sigma U^\star[\theta_\chi] + \int_{\partial\sigma} U^\star[q_\chi - \beta_U^c] \right) \tag{95}$$

$$= \left( -\mathsf{L}_{\check{\rho}} + \delta\check{\rho}\cdot \right) \left( \int_\sigma U^\star[\theta_\chi] + \int_{\partial\sigma} U^\star[q_\chi - \beta_U^c] \right) \tag{96}$$

$$= \left( -\mathcal{L}_\rho + \delta\check{\rho}\cdot \right) \left( \int_\sigma U^\star[\theta_\chi] + \int_{\partial\sigma} U^\star[q_\chi - \beta_U^c] \right) + \int_{\partial\sigma} \Delta_\rho U^\star[\beta_U^c] \tag{97}$$

$$= \delta \left( \int_\sigma \rho\lrcorner U^\star L + \int_{\partial\sigma} U^\star[q_{U_\star\rho} - \widehat{U_\star\rho}\cdot\beta] \right) - \int_\sigma \rho\lrcorner dU^\star[\theta_\chi] \tag{98}$$

$$\quad - \int_{\partial\sigma} \left( \rho\lrcorner U^\star[\theta_\chi] + \rho\lrcorner dU^\star[q_\chi - \beta_U^c] + d\rho\lrcorner U^\star[q_\chi - \beta_U^c] - \Delta_\rho U^\star[\beta_U^c] \right)$$

$$= \delta \left( \int_{\partial\sigma} U^\star[q_{U_\star\rho} - \widehat{U_\star\rho}\cdot\beta] \right) \tag{99}$$

$$\quad + \int_\sigma \rho\lrcorner U^\star E + \int_{\partial\sigma} \left( \rho\lrcorner \left( U^\star[C_\chi] - \theta_{\text{inv}} + dU^\star[\beta_U^c] \right) + \Delta_\rho U^\star[\beta_U^c] \right).$$

In moving to equation (97), we have used that the integrals on the right are anomaly-free with respect to diffeomorphisms of $\mathfrak{m}$, with the possible exception of $U^\star[\beta_U^c]$ (which we will come back to shortly). This assumption holds for the geodesic, but not necessarily for the material frames (cf. Sec. 3.4); the following steps would thus have to be modified for the dust frame (or restricted to those $\rho \in \mathfrak{X}(\mathfrak{m})$ that preserve the dust four-velocity $V$). In moving to (98), we have made use of $\check{\rho} \cdot U^\star[\theta_\chi] = U^\star[\check{\rho} \cdot (\theta + \chi\lrcorner L)] = U^\star[(U_\star\rho)\lrcorner L)] = \rho\lrcorner U^\star L$ and

---

[23]As an exception, note that thus far $U^\star[\beta_U^c]$ is only defined on the timelike boundary $\gamma$ through equation (75). To compute its Lie derivative in any direction not tangential to $\gamma$ requires some prescription for extending this quantity off the boundary. We leave open the possibility of such an extension here, though for most of the ensuing discussion we will be interested in the case that $\check{\rho}$ is tangential to $\gamma$.

$\check{\rho} \cdot U^\star[(q_\chi - \beta_U^c)] = U^\star[q_{U_\star \rho} - \widehat{U_\star \rho} \cdot \beta]$.[24] In moving to equation (99), we have assumed that $\partial\sigma$ has no boundary to discard the exact term in the boundary integrand, used (60) to infer that $\theta_{\text{inv}} = U^\star[\theta_\chi] + U^\star[C_\chi] + dU^\star[q_\chi]$, and $\rho \lrcorner dU^\star[\theta_\chi] = \rho \lrcorner (\delta U^\star L - U^\star E) = \delta(\rho \lrcorner U^\star L) - \rho \lrcorner U^\star E$. All-in-all, we have obtained the fundamental off-shell relation

$$\boxed{-\check{\rho} \cdot \Omega(\sigma) = \delta Q_\rho(\partial\sigma) + \mathcal{F}_\rho(\sigma)\,,} \tag{100}$$

where the *charge* $Q_\rho$ and *flux* $\mathcal{F}_\rho$ are defined as

$$Q_\rho(\partial\sigma) := \int_{\partial\sigma} U^\star[q_{U_\star \rho} - \widehat{U_\star \rho} \cdot \beta] = \int_{\partial\sigma} \left(U^\star[q_{U_\star \rho}] - \check{\rho} \cdot \beta_{\text{inv}}\right), \tag{101}$$

$$\mathcal{F}_\rho(\sigma) := \int_\sigma \rho \lrcorner U^\star E + \int_{\partial\sigma} \rho \lrcorner \left(U^\star[C_\chi] - \theta_{\text{inv}} + dU^\star[\beta_U^c]\right) + \int_{\partial\sigma} \Delta_\rho U^\star[\beta_U^c] \tag{102}$$

$$\approx -\int_{\partial\sigma} \rho \lrcorner (\theta_{\text{inv}} - d\beta_{\text{inv}}) + \int_{\partial\sigma} \Delta_\rho \beta_{\text{inv}}\,. \tag{103}$$

The presence of a non-exact flux term $\mathcal{F}_\rho$, even on-shell, means that additional restrictions need to be imposed on $\rho$ and/or field-space in order to render $\check{\rho}$ integrable. This is a key difference with local gauge theories such as Yang-Mills theory, for which any analogue of the symmetry generator $\check{\rho}$ is integrable off-shell (see, for instance, Sec. 7.4 of [1]). The reason behind this difference is however clear: given that, in gravitational systems, the definition of the subregion $M$ is frame-dependent, certain transformations of the frame will not leave it invariant; such transformations cannot be understood as closed subsystem transformations, and therefore cannot be Hamiltonian with respect to $\Omega(\sigma)$.

By construction, the contribution of $\Delta_\rho \beta_{\text{inv}}$ to the flux in (103) must vanish identically for any $\rho$ that is tangential to $\gamma$. The case of non-tangential vector fields may be interesting to consider, but it would require a prescription to extend $\beta_{\text{inv}}$ off of $\gamma$. A natural option may be to require $\Delta_\rho \beta_{\text{inv}} = 0$ for transverse $\rho$. However, from the next paragraph onwards, we will always restrict our attention to vector fields leaving $\gamma$ invariant. Even for such vector fields, one might still worry that $\beta_{\text{inv}}$ depends not only on $\chi$ and the pullback of fields to the boundary, but also on the boundary geometry. For the case of timelike boundaries, this geometric structure can be expressed in terms of the normal vector to the surface $\gamma$, $n^a$. It can be shown (see Sec. 6.2) that $\Delta_\rho n^a = 0$ for any tangential $\rho$, and thus $\beta_{\text{inv}}$ itself is anomaly-free for such vector fields. The contribution of $\Delta_\rho \beta_{\text{inv}}$ can thus be dropped unambiguously (see Appendix C for some related discussion).

From (100), one can pursue several (standard) strategies to turn $\check{\rho}$ into an Hamiltonian vector field. As a first option, the flux term can be made field-space exact if one imposes suitable boundary conditions. In our framework, boundary equations of motion on $\gamma$ will always be such that

$$(\theta_{\text{inv}} + \delta\ell)\big|_\gamma \underset{\mathcal{S}_{x_0}}{=} d\beta_{\text{inv}} \tag{104}$$

(with $\beta_{\text{inv}} = U^\star[\beta_U^c]$) for some boundary Lagrangian form $\ell$ on $\gamma$, where the symbol $\underset{\mathcal{S}_{x_0}}{=}$ indicates that the equality only holds for field configurations that lie on $\mathcal{S}_{x_0}$. Said differently, $\mathcal{S}_{x_0}$ is *defined* as the locus of field configurations in $\mathcal{S}$ where $(\theta_{\text{inv}} + \delta\ell)\big|_\gamma = d\beta_{\text{inv}}$ holds. This is a stronger statement than the weak equality $(\theta_{\text{inv}} + \delta\ell)\big|_\gamma \hat{\approx} d\beta_{\text{inv}}$, as the latter also involves

---

[24] We have used $U^\star[\widehat{U_\star \rho} \cdot \beta] = \check{\rho} \cdot \beta_{\text{inv}} = \check{\rho} \cdot U^\star \beta_U^c$, which can be checked by writing $\beta = \beta_a \delta\phi^a$ in components:

$$U^\star[\widehat{U_\star \rho} \cdot \beta] = U^\star[\beta_a \mathcal{L}_{U_\star \rho} \phi^a] = U^\star[\beta_a]\mathcal{L}_\rho U^\star[\phi^a] = \check{\rho} \cdot (U^\star[\beta_a]\delta U^\star[\phi^a]) = \check{\rho} \cdot U^\star[\beta_a \delta_\chi \phi^a] = \check{\rho} \cdot \beta_{\text{inv}}\,.$$

a pullback to $\mathcal{S}_{x_0}$.[25] We defer a more in-depth discussion of this point to Sec. 6, but as a first consistency check, note that this condition does imply $\omega_{\text{inv}} = \delta\theta_{\text{inv}} \,\hat{\approx}\, \mathrm{d}\delta\beta_{\text{inv}}$ on $\gamma$, which coincides with the assumption made in (75). Moreover, remembering that $\theta_{\text{inv}} = U^{\star}[\theta_U^{\mathrm{c}}] = U^{\star}[\theta_\chi] - U^{\star}[(\theta_\chi)_{\text{gauge}}]$ and $(\theta_\chi)_{\text{gauge}} = -J_\chi \approx -\mathrm{d}q_\chi$, we can reexpress this condition in the same form as derived in (46):

$$\left(U^{\star}[\theta_\chi] + \delta\ell\right)\big|_\gamma \,\hat{\approx}\, \mathrm{d}U^{\star}[\beta_U^{\mathrm{c}} - q_\chi]. \tag{105}$$

In particular, we can identify $\mathcal{C}$ from (46) as $\mathcal{C} := U^{\star}[\beta_U^{\mathrm{c}} - q_\chi] = \beta_{\text{inv}} - U^{\star}[q_\chi]$. If, in addition to the boundary condition (104) (which involves a pullback of $\theta_{\text{inv}}$ to $\gamma$), we assume $\rho$ to leave $\gamma$ invariant, then our boundary equations of motion imply that $\rho \lrcorner (\theta_{\text{inv}} - \mathrm{d}\beta_{\text{inv}}) \underset{\mathcal{S}_{x_0}}{=} -\delta(\rho\lrcorner\ell)$ on $\partial\sigma$, and therefore $\mathcal{F}_\rho(\sigma) \underset{\mathcal{S}_{x_0}}{=} \delta\left(\int_{\partial\sigma}\rho\lrcorner\ell\right)$. Under these conditions, $\check{\rho}$ becomes integrable in the sense that

$$-\check{\rho}\cdot\Omega(\sigma) \underset{\mathcal{S}_{x_0}}{=} \delta Q_\rho^{\mathrm{H}}(\partial\sigma), \tag{106}$$

with *Hamiltonian charge* defined as:

$$Q_\rho^{\mathrm{H}}(s) = Q_\rho(s) + \int_s \rho\lrcorner\ell = \int_s \left(U^{\star}[q_{U_{\star}\rho}] - \check{\rho}\cdot\beta_{\text{inv}} + \rho\lrcorner\ell\right), \tag{107}$$

where $s$ is any corner contained in $\gamma$ (that is, a spacelike cross-section of $\gamma$ with the topology of $S^{d-2}$). As a consistency check, we verify that $\mathcal{L}_\rho\Omega(\sigma) = \mathsf{L}_{\check{\rho}}\Omega(\sigma) = \delta(\check{\rho}\cdot\Omega(\sigma)) \,\hat{\approx}\, 0$: this should be the case, since $\mathcal{L}_\rho\Omega(\sigma) \,\hat{\approx}\, 0$ is the infinitesimal version of the conservation equation we required $\Omega(\sigma) \,\hat{\approx}\, \Omega_m$ to verify on-shell in Sec. 4.[26]

As a second option leading to integrable charges, one can obtain a vanishing flux without recourse to boundary conditions if we impose stronger conditions on $\rho$ itself. For example, the flux $\mathcal{F}_\rho(\sigma)$ in (100) vanishes off-shell if one requires $\rho$ to leave the Cauchy slice $\sigma$ invariant. However, this type of integrability depends explicitly on a kinematical choice of Cauchy slice $\sigma$. We are led to a more interesting situation if we only fix the boundary of this Cauchy slice (i.e. the corner) $s = \partial\sigma$ while working on-shell of the bulk equations of motion $E \approx 0$. Then, $\Omega(\sigma') \approx \Omega(\sigma)$ for any Cauchy slice $\sigma'$ such that $\partial\sigma = \partial\sigma'$. We can therefore view $\Omega(\sigma)$ as a function of $s$ which we denote $\Omega(s)$ by slight abuse of notation. In this set-up, any vector field $\rho$ leaving the corner $\partial\sigma$ invariant is integrable relative to $\Omega(s)$, independently of any boundary condition. Furthermore, the associated Hamiltonian charge is $Q_\rho^{\mathrm{H}}(s) = Q_\rho(s)$, which is also consistent with (107).

In summary, we have found two particularly interesting situations in which (100) leads to integrable frame reorientations:

1. Boundary conditions of the form (104) are being imposed: then, any $\rho$ leaving $\gamma$ invariant is integrable *fully on-shell*, and gives rise to boundary charges $Q_\rho^{\mathrm{H}}(s)$ (107) relative to $\Omega(s)$, where $s$ is a corner contained in $\gamma$. In addition, the conservation equation (84) ensures that $\delta Q_\rho^{\mathrm{H}}(s) \,\hat{\approx}\, \delta Q_\rho^{\mathrm{H}}(s')$ whenever $\check{\rho}$ leaves the boundary conditions invariant (or in other words, whenever $\check{\rho}$ is parallel to the leaf $\mathcal{S}_{x_0}$), in which case we can directly write $-\check{\rho}\cdot\Omega_m = \delta Q_\rho^{\mathrm{H}}$.

---

[25]This is similar to the way the solution space $\mathcal{S}$ is defined, namely as the locus of field configurations in $\mathcal{F}$ where $E := E_a\delta\Phi^a = 0$ holds. This entails that $E_a = 0$ on $\mathcal{S}$, which is indeed a stronger statement than the pulled-back equation $E \approx 0$.

[26]Note that this conservation statement can only be made after pullback to $\mathcal{S}_{x_0}$, as is already clear from the derivation in Sec. 4. Indeed, we could have $-\check{\rho}\cdot\Omega(\sigma) = \delta Q_\rho^{\mathrm{H}}(\partial\sigma) + \Upsilon_a\delta\Phi^a$, where $\Upsilon_a \underset{\mathcal{S}_{x_0}}{=} 0$ but $\delta\Upsilon_a \underset{\mathcal{S}_{x_0}}{\neq} 0$. It is then clear that $\delta(\check{\rho}\cdot\Omega(\sigma)) = \delta\Upsilon_a\delta\Phi^a \underset{\mathcal{S}_{x_0}}{\neq} 0$ in general. However, pulling back the forms to $\mathcal{S}_{x_0}$, we do recover $\delta(\check{\rho}\cdot\Omega(\sigma)) \,\hat{\approx}\, 0$ from $\delta\Upsilon_a \,\hat{\approx}\, 0$.

2. Bulk equations of motion are assumed, but no boundary condition is being imposed: then, any $\rho$ leaving a fixed corner $s$ invariant is integrable relative to $\Omega(s)$, and gives rise to boundary charges $Q_\rho^{\mathrm{H}}(s) = Q_\rho(s)$ (107).

The second situation has received a lot of attention since the pioneering work of Donnelly and Freidel [10], under the umbrella name of *local holography* (see e.g. [20] and references therein).

Before concluding this subsection, let us reexpress our findings in terms of the CLPF presymplectic structure. To start with, (100) yields:

$$-\breve{\rho} \cdot \Omega^{\mathrm{CLPF}}(\sigma) = \int_\sigma \rho \lrcorner U^\star[E] - \int_{\partial\sigma} \rho \lrcorner U^\star[\theta_\chi]. \tag{108}$$

The right-hand side only contains a flux contribution, which means that, on-shell, any generator from the corner symmetry algebra is null with respect to $\Omega^{\mathrm{CLPF}}$. In our construction, this conclusion is unsatisfactory since those are not in general gauge transformations. If one instead tries to define charges on-shell of physical boundary conditions of the form (104), one is led to

$$-\breve{\rho} \cdot \Omega^{\mathrm{CLPF}}(\sigma) \underset{\mathscr{S}_{x_0}}{=} \delta\left(\int_{\partial\sigma} \rho \lrcorner \ell\right) + \int_{\partial\sigma} \rho \lrcorner \mathrm{d}U^\star[q_\chi - \beta_U^{\mathrm{c}}]. \tag{109}$$

We now pick up a non-trivial exact contribution, but the non-exact second term does not vanish unless we impose further restrictions. Moreover, it is not clear which type of extra condition may prove useful for that purpose: asking $\rho$ to be parallel to $\gamma$ is not sufficient to cancel the flux in general, and asking it to leave the corner $\partial\sigma$ invariant leads us back to trivial charges, as we have just discussed. We observe once more that, in our construction, the CLPF form is not the appropriate one to consider. As we have already seen, it is not conserved in the presence of boundary conditions (since conservation requires the inclusion of the boundary contribution in equation (82)). It also attributes non-vanishing charges to diffeomorphisms on $\mathcal{M}$ (which are gauge directions). We have now identified a third defect: $\Omega^{\mathrm{CLPF}}$ does not associate Hamiltonian charges to frame-reorientations that leave $\gamma$ invariant.

### 5.3 Conservation equations

The boundary charges $Q_\rho^{\mathrm{H}}(s)$ are not in general conserved, meaning that their numerical values depend on the choice of corner $s$ on which they are evaluated. To see this, consider two corners $s_1$ and $s_2$ included in $\gamma$, and let us assume for definiteness that the latter is in the future of the former (this assumption can be relaxed). We can then call $\gamma_{s_1 s_2}$ the portion of the boundary $\gamma$ that is delimited by those two corners, as illustrated in Fig. 5. Integrating $\mathrm{d}(U^\star[q_{U_\star\rho}] - \breve{\rho} \cdot \beta_{\mathrm{inv}} + \rho \lrcorner \ell)$ on $\gamma_{s_1 s_2}$ and invoking Stokes' theorem, we then obtain:

$$Q_\rho^{\mathrm{H}}(s_2) - Q_\rho^{\mathrm{H}}(s_1) = \int_{\gamma_{s_1 s_2}} \mathrm{d}(U^\star[q_{U_\star\rho}] - \breve{\rho} \cdot \beta_{\mathrm{inv}} + \rho \lrcorner \ell)$$

$$= \int_{\gamma_{s_1 s_2}} \breve{\rho} \cdot (\theta_{\mathrm{inv}} + \delta\ell - \mathrm{d}\beta_{\mathrm{inv}}) - U^\star[C_{U_\star\rho}] - \Delta_\rho\ell, \tag{110}$$

where we have used the off-shell identity $\mathrm{d}U^\star[q_{U_\star\rho}] = U^\star[J_{U_\star\rho} - C_{U_\star\rho}] = U^\star[\widehat{U_\star\rho} \cdot \theta - (U_\star\rho)\lrcorner L - C_{U_\star\rho}] = \breve{\rho} \cdot \theta_{\mathrm{inv}} - \rho\lrcorner U^\star L - U^\star[C_{U_\star\rho}]$ and the fact that $\rho$ is parallel to $\gamma$. The second term in (110) vanishes on-shell of the equations of motion, while the first vanishes on-shell of the boundary conditions by (104). In other words, in the post-selected theory, the change in the charge along $\gamma_{s_1 s_2}$ is determined by the anomaly of $\ell$, that is, its failure to be covariant under $\rho$:[27]

---

[27]Remember that in this section $\rho$ is assumed to be field-independent. Hence $\Delta_\rho = \mathsf{L}_{\breve{\rho}} - \mathcal{L}_\rho$, which means that covariance and anomaly capture the same concept.

$$Q_\rho^{\mathrm{H}}(s_2) - Q_\rho^{\mathrm{H}}(s_1) \underset{\mathcal{S}_{x_0}}{=} -\int_{\gamma_{s_1 s_2}} \Delta_\rho \ell \,. \tag{111}$$

In particular, $Q_\rho^{\mathrm{H}}$ defines a *conserved charge* whenever the boundary Lagrangian $\ell$ is covariant with respect to $\rho$. For the boundary Lagrangians constructed in Sec. 6, any such $\rho$ will be a symmetry of the background fields defining the boundary conditions (such as, for instance, a Killing field of the induced metric on $\gamma$). Hence, conserved charges are associated to rigid symmetries, as one might expect.

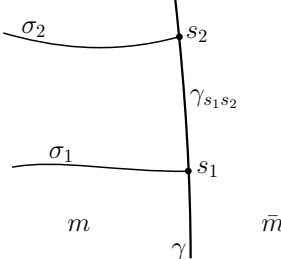

Figure 5: The boundary charges computed on $s_1$ and $s_2$ only differ by an integral over the portion of the time-like boundary $\gamma_{s_1 s_2}$ they delimit, as specified by equation (111).

We can also deduce an infinitesimal version of the balance equation (111). If we take $s_2$ to be an infinitesimal deformation of $s_1 = s$ by the vector field $\rho_2$, and define $\rho_1 = \rho$, (110) implies:

$$\mathcal{L}_{\rho_2} Q_{\rho_1}^{\mathrm{H}}(s) = \int_s \rho_{2\lrcorner} \left( \breve{\rho}_1 \cdot (\theta_{\mathrm{inv}} + \delta \ell - \mathrm{d}\beta_{\mathrm{inv}}) - U^\star[C_{U_\star \rho_1}] - \Delta_{\rho_1} \ell \right) \underset{\mathcal{S}_{x_0}}{=} -\int_s \rho_{2\lrcorner} \Delta_{\rho_1} \ell \,. \tag{112}$$

Taking $\ell = 0$, we finally obtain an off-shell expression for $\mathcal{L}_{\rho_2} Q_{\rho_1}$ which will prove useful below:

$$\mathcal{L}_{\rho_2} Q_{\rho_1}(s) = \int_s \rho_{2\lrcorner} \left( \breve{\rho}_1 \cdot (\theta_{\mathrm{inv}} - \mathrm{d}\beta_{\mathrm{inv}}) - U^\star[C_{U_\star \rho_1}] \right) . \tag{113}$$

## 5.4 Boundary symmetry algebras

Let us turn to an analysis of the Poisson algebra generated by the previously defined charges. We start by computing the quantity

$$-\breve{\rho}_1 \cdot \breve{\rho}_2 \cdot \Omega(\sigma) = \breve{\rho}_1 \cdot \left( \delta Q_{\rho_2}(\partial\sigma) + \mathcal{F}_{\rho_2}(\sigma) \right) \tag{114}$$

fully off-shell, and will then discuss appropriate conditions under which a Poisson bracket for boundary charges can be extracted from it. The first term in (114) can be evaluated in the following way:

$$\begin{aligned}
\breve{\rho}_1 \cdot \delta Q_{\rho_2}(\partial\sigma) = \mathsf{L}_{\breve{\rho}_1} Q_{\rho_2}(\partial\sigma) &= \mathcal{L}_{\rho_1} Q_{\rho_2}(\partial\sigma) - Q_{\mathcal{L}_{\rho_1}\rho_2}(\partial\sigma) \\
&= \mathcal{L}_{\rho_1} Q_{\rho_2}(\partial\sigma) - Q_{[\rho_1,\rho_2]}(\partial\sigma) .
\end{aligned} \tag{115}$$

In the second equality, we noticed that $Q_{\rho_2}$ is linear in the background vector field $\rho_2$, and its only other dependence on $m$ is through $U$, which transforms covariantly. We can then turn to the flux contribution:

$$\begin{aligned}
\breve{\rho}_1 \cdot \mathcal{F}_{\rho_2}(\sigma) = \breve{\rho}_1 \cdot \left( \int_\sigma \rho_{2\lrcorner} U^\star E + \int_{\partial\sigma} \rho_{2\lrcorner} \left( U^\star[C_\chi] - \theta_{\mathrm{inv}} + \mathrm{d}\beta_{\mathrm{inv}} \right) \right) \\
= \int_{\partial\sigma} \rho_{2\lrcorner} \left( U^\star[C_{U_\star \rho_1}] - \breve{\rho}_1 \cdot (\theta_{\mathrm{inv}} - \mathrm{d}\beta_{\mathrm{inv}}) \right) = -\mathcal{L}_{\rho_2} Q_{\rho_1}(\partial\sigma),
\end{aligned} \tag{116}$$

where in the second line, we have used (113) together with the fact that $\check{\rho}_1 \cdot U^\star E = 0$ (since $E$ only involves variations of spacetime fields). All in all, we conclude that

$$\boxed{-\check{\rho}_1 \cdot \check{\rho}_2 \cdot \Omega(\sigma) = -Q_{[\rho_1,\rho_2]}(\partial\sigma) + \mathcal{L}_{\rho_1}Q_{\rho_2}(\partial\sigma) - \mathcal{L}_{\rho_2}Q_{\rho_1}(\partial\sigma)}, \qquad (117)$$

or, if we express the last two contributions as in (113):

$$- \check{\rho}_1 \cdot \check{\rho}_2 \cdot \Omega(\sigma) = -Q_{[\rho_1,\rho_2]}(\partial\sigma) \qquad (118)$$
$$+ \int_{\partial\sigma} \left( \rho_{1\lrcorner} \left( \check{\rho}_2 \cdot (\theta_{\text{inv}} - \mathrm{d}\beta_{\text{inv}}) - U^\star[C_{U_\star\rho_2}] \right) - \rho_{2\lrcorner} \left( \check{\rho}_1 \cdot (\theta_{\text{inv}} - \mathrm{d}\beta_{\text{inv}}) - U^\star[C_{U_\star\rho_1}] \right) \right).$$

Note that this result is valid completely off-shell.

Let us now discuss two ways in which (117) allows to define a corner symmetry algebra: first at the off-shell level, then on-shell of the bulk equations of motion. What we need is to impose conditions such that the vector fields $\rho_1$ and $\rho_2$ are both integrable relative to $\Omega(\sigma)$. Given that $\Omega(\sigma)$ is field-space closed, we are then guaranteed that the bracket $\{Q_{\rho_1}(\partial\sigma), Q_{\rho_2}(\partial\sigma)\}_\sigma := -\check{\rho}_1 \cdot \check{\rho}_2 \cdot \Omega(\sigma)$ verifies the Jacobi identity, and thus defines a Poisson bracket.

As we have discussed in the previous subsection, the functionals $\{Q_\rho(s)\}$ do not generically constitute Hamiltonian charges of integrable field-space vector fields off-shell. Indeed, remember that at the off-shell level we need to commit to a particular Cauchy slice $\sigma$. If we do so, then the charges $\{Q_\rho(\partial\sigma)\}$ are off-shell integrable with respect to $\Omega(\sigma)$ for any $\rho$ *leaving* $\sigma$ *invariant*. This subclass of vector fields generates the off-shell algebra:

$$\{Q_{\rho_1}(\partial\sigma), Q_{\rho_2}(\partial\sigma)\}_\sigma := -\check{\rho}_1 \cdot \check{\rho}_2 \cdot \Omega(\sigma) = -Q_{[\rho_1,\rho_2]}(\partial\sigma), \qquad (119)$$

where the notation $\{\cdot, \cdot\}_\sigma$ emphasizes that we have defined a purely kinematical notion that depends on a choice of Cauchy slice $\sigma$. Note from (118) that the $\mathcal{L}_{\rho_i}Q_{\rho_j}(\partial\sigma)$ contributions vanish because the integrable vector fields $\rho_i$ leave the corner $\partial\sigma$ invariant.

If we instead decide to work on-shell (but still without imposing boundary conditions), the charges $\{Q_\rho(\partial\sigma)\}$ are integrable for any $\rho$ leaving the corner $s \equiv \partial\sigma$ invariant. Furthermore, given that $\Omega(\sigma) \approx \Omega(\sigma')$ whenever $\partial\sigma = s = \partial\sigma'$, the choice of Cauchy slice in the bulk is irrelevant provided that the corner $s$ is kept fixed. We thus obtain a closed corner algebra indexed by arbitrary vector fields *leaving the corner s invariant*:

$$\{Q_{\rho_1}(s), Q_{\rho_2}(s)\}_s := -\check{\rho}_1 \cdot \check{\rho}_2 \cdot \Omega(\sigma) = -Q_{[\rho_1,\rho_2]}(s), \qquad (120)$$

where the notation $\{\cdot, \cdot\}_s$ emphasizes that we have now obtained a Poisson bracket that only depends on a choice of corner $s$. Upon quotienting by gauge directions (that is, vector fields $\rho$ such that $\delta Q_\rho(s) \approx 0$), this defines a *corner algebra* as initially investigated in [10] in the context of vacuum general relativity. It has a universal part, that generates a non-trivial representation of $\text{Diff}(s)$, and depending on the choice of $\beta$ in (104), it may also include additional factors [15].[28] As we will discuss in Sec. 6.3, the original proposal of [10] is recovered from the standard presymplectic potential (7) of general relativity with $\beta = 0$. The resulting corner algebra is then isomorphic to the Lie algebra of $\text{Diff}(s) \ltimes \text{SL}(2, \mathbb{R})^s$: in addition to intrinsic surface diffeomorphisms of the corner which are encoded in the first factor, vector fields whose normal component to $s$ vanish on $s$ but not in a first order neighborhood of $s$ – the so-called *surface boosts* – also contribute.

---

[28]In [15], the possible structures of the corner algebra are actually classified in terms of a choice of Lagrangian form. This is made possible by a specific choice of presymplectic potential that ties the ambiguity encoded in the choice of $\beta$ to a choice of Lagrangian. We do not take this step in the present work, and therefore directly discuss the ambiguity in terms of $\beta$.

Let us now repeat this construction for the Hamiltonian charges $\{Q_\rho^{\mathrm{H}}(s)\}$ in the presence of a boundary condition ensuring that (104) holds. To ensure integrability, there is no question that we need to work fully on-shell in this case. Given that $\ell$ is also a potential source of non-covariance, we have (note the extra term in the first line as compared to (115)):

$$\check{\rho}_1 \cdot \delta Q_{\rho_2}^{\mathrm{H}}(s) = \mathsf{L}_{\check{\rho}_1} Q_{\rho_2}^{\mathrm{H}}(s) = \mathcal{L}_{\rho_1} Q_{\rho_2}^{\mathrm{H}}(s) - Q_{[\rho_1,\rho_2]}^{\mathrm{H}}(s) + \int_s \rho_2 \lrcorner \Delta_{\rho_1} \ell \qquad (121)$$

$$\underset{\mathcal{S}_{x_0}}{=} -Q_{[\rho_1,\rho_2]}^{\mathrm{H}}(s) + \int_s \rho_2 \lrcorner \Delta_{\rho_1} \ell - \int_s \rho_1 \lrcorner \Delta_{\rho_2} \ell, \qquad (122)$$

where we invoked (112) in arriving at the last line. This leads to the following definition of the bracket in the presence of boundary conditions:

$$\boxed{\{Q_{\rho_1}^{\mathrm{H}}(s), Q_{\rho_2}^{\mathrm{H}}(s)\}_\gamma := -Q_{[\rho_1,\rho_2]}^{\mathrm{H}}(s) - K_{\rho_1,\rho_2}(s), \text{ with } K_{\rho_1,\rho_2}(s) := \int_s \left(\rho_1 \lrcorner \Delta_{\rho_2} \ell - \rho_2 \lrcorner \Delta_{\rho_1} \ell\right).} \qquad (123)$$

The notation $\{\cdot,\cdot\}_\gamma$ indicates that this bracket explicitly depends on a choice of boundary conditions on the time-like boundary $\gamma$, and that the vector fields $\rho_1$ and $\rho_2$ are required to leave $\gamma$ invariant. If, in addition, one assumes $\check{\rho}_1$ and $\check{\rho}_2$ to be parallel to $\mathcal{S}_{x_0}$, the bracket is also independent of the choice of corner $s \subset \gamma$ on which it is evaluated. The quantity $K_{\rho_1,\rho_2}(s)$ contributing to the right-hand side of (123) identically vanishes for vector fields $\rho_1$ and $\rho_2$ that are parallel to $s$. It is therefore clear that, for each corner $s \subset \gamma$, the associated corner algebra (120) is a subalgebra of the codimension-1 boundary algebra generated by (123).

The structure of the second term in (123) is consistent with the results of [19], which we recover here in their frame-dressed version. In that work, an expression of the form (123) was proposed at the fully off-shell level, for what is known in the literature as the *Barnich–Troessaert bracket* [30]. In the present construction, $\{\cdot,\cdot\}_\gamma$ is more straightforwardly understood as the Poisson bracket associated to $\Omega(s)$ in $\mathcal{S}$, and as such, it is guaranteed to obey Jacobi's identity.

The physical interpretation of a charge $Q_{\rho_1}^{\mathrm{H}}(s)$ is in general dependent on the specific boundary conditions being considered, or equivalently, on the choice of boundary Lagrangian $\ell$. Following the nomenclature of [1], it may in principle generate one of three types of transformations:

- a *boundary symmetry*, that leaves the boundary conditions invariant and can be recorded by a non-trivial variation $\delta Q_\rho^{\mathrm{H}}(s) \not\approx 0$;

- a *boundary gauge transformation*, that leaves the boundary conditions invariant but leads to a trivial variation of the charge $\delta Q_\rho^{\mathrm{H}}(s) \approx 0$;

- a *meta-symmetry*, that does not leave the boundary conditions invariant, but still defines a symplectomorphism (to a solution space associated to different values of the background fields that define the boundary conditions).

We will illustrate this classification in the specific context of vacuum general relativity in Sec. 6.3 below. However, we can already show that infinitesimal boundary symmetries and boundary gauge transformations must generate a closed subalgebra under the bracket $\{\cdot,\cdot\}_\gamma$. To exclude meta-symmetries, we assume in the rest of this subsection that any vector field on $m$ *preserves both $\gamma$ and the boundary conditions*, unless otherwise specified. For any such $\rho_i$, with $1 \leq i \leq 4$, we can show that:

$$\boxed{\{Q_{\rho_3}^{\mathrm{H}}(s), K_{\rho_1,\rho_2}(s)\}_\gamma = 0, \qquad \{K_{\rho_1,\rho_2}(s), K_{\rho_3,\rho_4}(s)\}_\gamma = 0.} \qquad (124)$$

As a result, we do obtain a closed algebra of charges, which consists in a representation of the algebra of vector fields preserving $\gamma$ and $\mathcal{S}_{x_0}$, or a central extension thereof. In order to prove (124), we observe the following: given that $\delta\ell \,\hat{\approx}\, -\theta_{\text{inv}} - \mathrm{d}\beta_{\text{inv}}$ and both $\theta_{\text{inv}}$ and $\beta_{\text{inv}}$ are covariant with respect to diffeomorphisms on $\mathfrak{m}$, it follows that $\Delta_\rho\delta\ell \,\hat{\approx}\, 0$ whenever $\check{\rho}$ is parallel to $\mathcal{S}_{x_0}$. That is, even if the boundary Lagrangian $\ell$ may fail to be covariant, its variation is always covariant once pulled-back to $\mathcal{S}_{x_0}$. Since we are only considering field-independent vector fields $\rho$, we can equivalently say that the pullback of $\delta(\Delta_\rho\ell)$ to $\mathcal{S}_{x_0}$ vanishes. Given (123), this in turn implies that

$$\delta K_{\rho_1,\rho_2}(s) \,\hat{\approx}\, 0\,. \tag{125}$$

By definition, the bracket $\{Q_{\rho_3}^{\text{H}}(s), K_{\rho_1,\rho_2}(s)\}_\gamma$ is obtained by contracting the previous expression with $\check{\rho}_3$, which therefore vanishes. More generally, we conclude from (125) that $K_{\rho_1,\rho_2}(s)$ is a central element in the algebra of gauge-invariant observables on $\mathcal{S}_{x_0}$ equipped with $\{\cdot,\cdot\}_\gamma$, which proves (124).

Consider a second corner $s' \subset \gamma$. Given that $\Omega(\sigma)$ is independent of $\sigma$ (and hence $s = \partial\sigma$) fully on-shell, we always have $\delta Q_\rho^{\text{H}}(s) \,\hat{\approx}\, \delta Q_\rho^{\text{H}}(s')$, even though $Q_\rho^{\text{H}}(s) \neq Q_\rho^{\text{H}}(s')$ in general. Hence the algebra (123) is completely independent from the choice of corner, meaning in particular that:

$$\{Q_{\rho_1}^{\text{H}}(s), Q_{\rho_2}^{\text{H}}(s)\}_\gamma = \{Q_{\rho_1}^{\text{H}}(s'), Q_{\rho_2}^{\text{H}}(s')\}_\gamma\,, \tag{126}$$

which is of course consistent with the conservation of the presymplectic structure once pulled-back to $\mathcal{S}_{x_0}$. However, the values of the central charges themselves may *a priori* depend on the corner on which they are evaluated. More specifically, combining (123) and (111) with the previous equation leads to the following balance relation for the central charges:

$$K_{\rho_1,\rho_2}(s') - K_{\rho_1,\rho_2}(s) = \int_{\gamma_{ss'}} \Delta_{[\rho_1,\rho_2]}\ell\,. \tag{127}$$

In its infinitesimal version, it yields

$$\mathcal{L}_{\rho_3} K_{\rho_1,\rho_2}(s) = \int_s \rho_3 \lrcorner \Delta_{[\rho_1,\rho_2]}\ell = -\int_s \rho_3 \lrcorner [\Delta_{\rho_1}, \Delta_{\rho_2}]\ell\,, \tag{128}$$

where $\rho_3$ is assumed to preserve $\gamma$, but not necessarily the boundary conditions. By contrast, we emphasize again that $\check{\rho}_1$ and $\check{\rho}_2$ must preserve $\mathcal{S}_{x_0}$ in order for these balance relations to hold, which implies that $\rho_1$ and $\rho_2$ are rigid symmetries.

We could in principle extend the algebra defined by (123) and (124) to also represent vector fields $\rho$ that are parallel to $\gamma$, but that fail to preserve the boundary conditions (that is, such that $\check{\rho}$ has a transverse component to $\mathcal{S}_{x_0}$).[29] This would require computing the *Dirac bracket* [100, 101] induced by the boundary conditions, which can typically be viewed as a set of second-class constraints defining the constraint hypersurface $\mathcal{S}_{x_0} \subset \mathcal{S}$. The Dirac bracket does coincide with $\{\cdot,\cdot\}_\gamma$ in the subalgebra of vector fields that preserve the boundary conditions, as considered in the previous paragraph, but we leave its complete construction for future work.[30]

---

[29]Following the nomenclature introduced in [1], such vector fields generate *meta-symmetries*.

[30]We refer the interested reader to [102] for an explicit comparison of the Barnich–Troessaert and Dirac brackets in asymptotically flat spacetimes.

# 6 Boundary actions

## 6.1 General construction

We now turn to the problem of identifying suitable actions for the subregion $m = U(M)$. As with the case of the symplectic structure in Sec. 4, we do this via the process of splitting post-selection developed in [1], to which we refer for further details on some of the steps. Beginning with a global action on $m \cup \bar{m}$, we seek a consistent factorization such that the variational problems are well-defined separately on both $m$ and its complement, and when combined, are equivalent to the global problem. In the case of the symplectic structure, we considered a restricted space $\mathcal{S}_{x_0}$ of solutions already satisfying a choice of gauge-invariant boundary condition $x = x_0$ on the timelike interface $\gamma = U(\Gamma)$. Here we will impose these boundary conditions dynamically. An integrable condition such as $\delta x = 0$ may be imposed at the level of the action as a holonomic constraint through the use of Lagrange multiplier terms in the global action.

To proceed with the factorization, we assume that the bulk action for each subregion is locally identical to the global action. This fixes the subregion actions up to possible additional boundary terms on $\gamma$. By including equal and opposite boundary contributions in the subregion actions for $M$ and $\bar{M}$, the sum of the subregion actions remains equivalent to the global action. Demanding that the total subregion action defines a consistent variational problem in its own right leads to a set of possible boundary terms appropriate to a wide class of different boundary conditions. From the perspective of the restricted variational problem on subregion $m$, the additional boundary term fills a role that would have been played by the complement region, effectively retaining some information about how the subregion theory is embedded within the global theory.

Recall that $m = U(M)$, $\bar{m} = U(\bar{M})$ provide an invariant specification of complementary subregions under the reference frame $U$, thought of as a field-dependent map from spacetime to an invariant parameter space we call relational spacetime. The same map also provides an invariant specification of subregion boundaries, including the time-like interface $\gamma = U(\Gamma)$, and future and past Cauchy surfaces $\sigma_i = U(\Sigma_i)$ and $\bar{\sigma}_i = U(\bar{\Sigma}_i)$ for $i \in (1, 2)$ (see figure 4). We first suppose that we are given some action for the global theory, and express it as

$$S_{m \cup \bar{m}} = \int_{m \cup \bar{m}} U^\star[L] = S_m + S_{\bar{m}}. \tag{129}$$

We assume that this action poses a well-defined variational problem for the global theory, possibly necessitating the inclusion of boundary terms for the complement region, which we leave implicit in $S_{\bar{m}}$. To further impose gauge-invariant boundary conditions at the interface between the two regions, we employ Lagrange multipliers. As discussed in Sec. 4, rather than fixing fields on $\Gamma \subset M$, we fix invariant data on $U(\Gamma) = \gamma$. We express the boundary condition as

$$x^i = x_0^i, \tag{130}$$

where the $x^i$ (collectively $x$) are functionals comprised of the invariant part of fundamental fields pushed forward to $\mathfrak{m}$ and pulled back to $\gamma$, and $x_0$ is a corresponding background configuration. Such boundary conditions may be imposed by adding

$$S_{\gamma,\text{bc}} = \int_\gamma \lambda_i (x^i - x_0^i) \tag{131}$$

to the global action, where the $\lambda_i$ are Lagrange multiplier densities on $\gamma$. Defining $\Lambda_i := U_\star \lambda_i$ and $X^i := U_\star x^i$, this can equivalently be written in terms of spacetime fields on $\Gamma$, with the understanding that the condition (130) will not impose $\delta X^i = 0$ but rather $\delta_\chi X^i = 0$. In

other words, the variation of $S_{\gamma,\text{bc}}$ can be written as either (henceforth leaving the sum over $i$ implicit)

$$\delta S_{\gamma,\text{bc}} = \int_\gamma \left( \delta\lambda(x-x_0) + \lambda\delta x \right), \tag{132}$$

or

$$\delta S_{\gamma,\text{bc}} = \int_\gamma U^\star[\delta_\chi\Lambda(X-X_0) + \Lambda\delta_\chi X], \tag{133}$$

and this action will be stationary when the boundary conditions are satisfied. Note that there is no separate boundary condition imposed on $\chi$ alone. After adding this term, we denote the resulting global action as

$$S_{\text{tot}} = S_{m\cup\bar{m}} + S_{\gamma,\text{bc}}. \tag{134}$$

Next, to split this global variational problem into subregion variational problems, we consider additional boundary terms on $\gamma$ ($S_m$ and $S_{\bar{m}}$ in $S_{\text{tot}}$ are only uniquely defined up to these local integrals over $\gamma$), defining

$$\tilde{S}_m = S_m + \int_\gamma \ell_{\text{corr}}, \qquad \tilde{S}_{\bar{m}} = S_{\bar{m}} - \int_\gamma \ell_{\text{corr}}, \tag{135}$$

where $\ell_{\text{corr}}$ is a $(d-1,0)$ form on $\gamma$, the details of which are to be determined.[31] The primary requirement for the consistency of the subregion variational problem is that

$$\delta\tilde{S}_m \,\hat{\approx}\, \int_{\sigma_2-\sigma_1} U^\star[\theta_\chi] + \int_{\partial\sigma_2-\partial\sigma_1} U^\star[\mathcal{C}], \tag{136}$$

for some $(d-2,1)$ form $\mathcal{C}$. This condition amounts to the statement that for field configurations satisfying both the bulk equations of motion on $M$ and the boundary condition, we must have stationarity up to terms on the past and future Cauchy slices $\sigma_1$ and $\sigma_2$ (and their corners $\partial\sigma_2, \partial\sigma_1$). This is achieved if

$$U^\star[\theta_\chi] + \delta\ell_{\text{corr}} \,\underset{\gamma}{\hat{\approx}}\, -\text{d}\mathcal{C}, \tag{137}$$

where the equality sign $\underset{\gamma}{\hat{\approx}}$ indicates that terms defined in the bulk are pulled back to $\gamma$, and both equations of motion and boundary conditions have also been imposed. This condition imposes that the variation of the added boundary term $\ell_{\text{corr}}$ must combine on shell with $U^\star[\theta_\chi]$ from the bulk variation to give a total derivative term on $\gamma$. The minus sign in equation (137) results from reaching $\partial\sigma$ surfaces via Stokes theorem on $\gamma$ as opposed to $\sigma$. When the boundary conditions are not imposed, a sufficient condition to satisfy (137) is that

$$\delta\ell_{\text{corr}} \,\underset{\gamma}{\approx}\, -U^\star[\theta_\chi] - \text{d}\mathcal{C} + y\delta x. \tag{138}$$

This can be thought of as the key consistency condition on the choice of $\ell_{\text{corr}}$. This will require choosing a decomposition of $U^\star[\theta_\chi]\big|_\gamma$, and a corresponding identification of suitable $\mathcal{C}, x$, and $y$, as we will see in examples shortly. Once $\ell_{\text{corr}}$ has been identified, we can proceed to impose the boundary conditions at the level of the subregion action via the same Lagrange multiplier term as in the global problem:

$$\tilde{S}_{m\cup\gamma} = S_m + \int_\gamma \left( \ell_{\text{corr}} + \lambda(x-x_0) \right). \tag{139}$$

---

[31]We choose the convention that the orientation of the surface $\gamma$ is always that induced by the subregion $M$, with outward pointing normal from this region.

The variational problem for $x^i$ simply tells us that $\lambda_i = -y_i$, where the $y_i$ are the densities (implicitly summed) appearing in equation (138). This can be reinserted into the action to give a variational problem for the remaining degrees of freedom. Thus, our final boundary action for the subregion problem on $\mathfrak{m}$ will be

$$S_\gamma := \int_\gamma \ell = \int_\gamma \left( \ell_{\mathrm{corr}} - y(x - x_0) \right). \tag{140}$$

It remains to find a satisfactory $\ell_{\mathrm{corr}}$. Thus far we have required that it satisfy equation (138). Of course a complete choice will depend on the theory at hand and the context under consideration, but we can say a few more things generally. We have shown in Sec. 3.3 that for covariant Lagrangians, the extended symplectic potential can always be decomposed as $\theta_\chi = (\theta_\chi)^{\mathrm{c}}_U + (\theta_\chi)^{\mathrm{nc}}_U$ where $(\theta_\chi)^{\mathrm{nc}}_U = -J_\chi \approx -dq_\chi$. The condition (138) becomes

$$\delta \ell_{\mathrm{corr}} \underset{\gamma}{\approx} -\theta_{\mathrm{inv}} + \mathrm{d}(U^\star[q_\chi] - \mathcal{C}) + y\delta x \,. \tag{141}$$

To proceed, we assume a decomposition of $\theta_{\mathrm{inv}}$ on $\gamma$ of the form

$$\theta_{\mathrm{inv}} \underset{\gamma}{\approx} -\delta \ell_{\mathrm{inv}} + \mathrm{d}\beta_{\mathrm{inv}} + \pi_{\mathrm{inv}}\delta\phi_{\mathrm{inv}}\,, \tag{142}$$

for some $\ell_{\mathrm{inv}}$ and $\beta_{\mathrm{inv}}$, where $\pi_{\mathrm{inv}}\delta\phi_{\mathrm{inv}}$ constitutes a sum (left implicit) over conjugate pairs of degrees of freedom on which the boundary conditions will by imposed. Note that this decomposition is compatible with the condition (75) considered in Sec. 4 for the symplectic structure. Under such a decomposition, equation (141) becomes

$$\delta \ell_{\mathrm{corr}} \underset{\gamma}{\approx} \delta \ell_{\mathrm{inv}} + (y\delta x - \pi_{\mathrm{inv}}\delta\phi_{\mathrm{inv}}) - \mathrm{d}(\mathcal{C} + \beta_{\mathrm{inv}} - U^\star[q_\chi])\,. \tag{143}$$

At this point it is clear that we can take

$$\mathcal{C} := U^\star[q_\chi] - \beta_{\mathrm{inv}}\,. \tag{144}$$

It is also clear that if we wish to impose Dirichlet boundary conditions on fields $\phi_{\mathrm{inv}}$, then we may choose

$$\begin{aligned} x = \phi_{\mathrm{inv}}\,, \quad y = \pi_{\mathrm{inv}}\,, \\ \ell_{\mathrm{corr}} = \ell_{\mathrm{inv}}\,. \end{aligned} \tag{145}$$

Alternatively, we could hold fixed the conjugate variables $\pi_{\mathrm{inv}}$ on $\gamma$ by choosing

$$\begin{aligned} x = \pi_{\mathrm{inv}}\,, \quad y = -\phi_{\mathrm{inv}}\,, \\ \ell_{\mathrm{corr}} = \ell_{\mathrm{inv}} - \pi_{\mathrm{inv}}\phi_{\mathrm{inv}}\,. \end{aligned} \tag{146}$$

More generally, we may choose $x$ and $y$ to be nontrivial combinations of $\pi_{\mathrm{inv}}$ and $\phi_{\mathrm{inv}}$ which render the right hand side of (143) field-space exact, thus identifying an appropriate boundary term $\ell_{\mathrm{corr}}$. For example, in the gauge theories explored in [1], actions appropriate to a variety of mixed (Robin type) boundary conditions are easy to identify in this format. In the case of general relativity, the options are more limited,[32] but we explore some alternative choices in the next section.

---

[32] For example, as we will see below, in general relativity we can have $\phi_{\mathrm{inv},\mu\nu} = h_{\mu\nu}$ and $\pi_{\mathrm{inv}}^{\mu\nu} = P^{\mu\nu}$, where $h_{\mu\nu}$ is the induced metric on $\gamma$ and $P^{\mu\nu}$ is related to its extrinsic curvature. If one wanted to define mixed boundary conditions in terms of linear combinations $x = a\,\phi_{\mathrm{inv}} + b\,\pi_{\mathrm{inv}}$, where we have left the field indices implicit, the coefficients would have to be tensors such that the index structure of the two terms agrees. When the theory on $\mathfrak{m}$ is relationally generally covariant and there are thus no background structures available, as is the case with the immaterial geodesic frames assumed here (see Sec. 3.4), the tensors $a, b$ would have to be field-dependent. This complicates the analysis considerably as one will be dealing with field-dependent canonical transformations that may only be valid in infinitesimal neighborhoods on field space. By contrast, the case of dust frames may offer an advantage here: the dust four-velocity $V$ represents a preferred structure on relational spacetime that could possibly be used as a background field to define field-independent coefficients $a, b$ and canonical transformations. However, we shall not explore this direction here further.

## 6.2 Einstein-Hilbert gravity

We illustrate the procedure of post-selection of subregion actions for the case of Einstein-Hilbert (EH) gravity, where the sole fundamental field is the metric. We must first achieve a decomposition of $\theta_{\mathrm{inv}}$ of the form in equation (142), which we repeat here for convenience:

$$\theta_{\mathrm{inv}} \underset{\gamma}{\approx} -\delta \ell_{\mathrm{inv}} + \mathrm{d}\beta_{\mathrm{inv}} + \pi_{\mathrm{inv}} \delta \phi_{\mathrm{inv}}.$$

Because the bulk Lagrangian satisfies the property $U^\star L_{\mathrm{EH}}[\bar{g}] = L_{\mathrm{EH}}[g]$, where $\bar{g}$ is the metric on spacetime and $g = U^\star \bar{g}$ is the metric on relational spacetime, $\theta_{\mathrm{inv}}^{\mathrm{EH}}$ takes the standard form for the Einstein-Hilbert Lagrangian (with $16\pi G = 1$):

$$\theta_{\mathrm{inv}}^{\mathrm{EH}} = g^{\alpha\beta} g^{\mu\nu} \left( \nabla_\mu \delta g_{\beta\nu} - \nabla_\beta \delta g_{\mu\nu} \right) \epsilon_\alpha, \tag{147}$$

but now this form and the fields comprising it are defined on relational spacetime $\mathfrak{m}$, and so crucially they do not transform under spacetime diffeomorphisms: $\hat{\xi} \cdot \theta_{\mathrm{inv}}^{\mathrm{EH}} = 0$. Upon pulling back to a timelike boundary $\gamma$, this can be written

$$\theta_{\mathrm{inv}}^{\mathrm{EH}}\big|_\gamma = -\delta \left( 2K\epsilon_\gamma \right) - D_\mu \left( h^{\mu\nu} n^\alpha \delta g_{\nu\alpha} \right) \epsilon_\gamma + \left( Kh^{\mu\nu} - K^{\mu\nu} \right) \epsilon_\gamma \delta h_{\mu\nu}. \tag{148}$$

This notation is fairly standard, but let us here define the various terms appearing in this expression. First, suppose that the hypersurface $\gamma$ is specified as a level set of some scalar function $f$ on relational spacetime. The extension of $f$ off $\gamma$ should be smooth, and we take it to increase away from the subregion $m$, but it is otherwise arbitrary. The unit-normalized vector field $n_\mu = \frac{\nabla_\mu f}{\sqrt{\nabla_\alpha f \nabla^\alpha f}}$ is orthogonal to $\gamma$ (and all other level sets of $f$), and is used to define the tensors $h_{\mu\nu} = g_{\mu\nu} - n_\mu n_\nu$ and $K_{\mu\nu} = h_\mu^{\ \alpha} \nabla_\alpha n_\nu$ in the vicinity of $\gamma$ where $f$ is defined. Evaluated on $\gamma$, the former is the projection tensor onto $\gamma$, and the latter is the extrinsic curvature tensor of $\gamma$, with trace $K = h^{\mu\nu} K_{\mu\nu} = g^{\mu\nu} K_{\mu\nu}$. As before, $\epsilon_\gamma$ denotes the induced volume form on the hypersurface $\gamma$, which is related to the full volume form as $\epsilon = n \wedge \epsilon_\gamma$. Lastly, the derivative $D_\mu = h_\mu^{\ \nu} \nabla_\nu$ projects the covariant derivative tangential to the hypersurface $\gamma$. When acting on tensors with only tangential components (as is the case in (148)) it acts as the covariant derivative intrinsic to the hypersurface. This allows us to write the last term in (142) as a total derivative term, in terms of the exterior derivative intrinsic to $\gamma$:

$$D_\mu \left( h^{\mu\nu} n^\alpha \delta g_{\nu\alpha} \right) \epsilon_\gamma = \mathrm{d}(V \lrcorner \epsilon_\gamma), \tag{149}$$

where the vector field $V$ is the field space one-form and an element of the tangent space of $\gamma$, with components in $d$-dimensional relational spacetime given by $V^\mu := h^{\mu\nu} n^\alpha \delta g_{\nu\alpha}$.

As a side remark, we observe that the normal $n$ is non-anomalous with respect to any diffeomorphism $\rho$ that is tangential to $\gamma$. To see this, consider a second vector field $\rho'$, also tangential to $\gamma$, and compute

$$0 = \Delta_{\rho'}(\rho^a n_a) = [\rho, \rho']^a n_a + \rho'^a \Delta_\rho n_a = \rho'^a \Delta_\rho n_a. \tag{150}$$

We have used that $\delta \rho^a = \delta \rho'^a = 0$ in the second equality, and the fact that $[\rho, \rho']$ is parallel to $\gamma$ in the third. Given that the last equality must hold for any tangential $\rho'$, we conclude that $\Delta_\rho n_a = w_\rho n_a$ for some function $w_\rho$. In addition, we can infer from the normalization condition $n^a n_a = 1$ that $n^a \Delta_\rho n_a = 0$. Hence we must have $w_\rho = 0$, which proves that $\Delta_\rho n_a = 0$. Note that this argument would not exclude a non-vanishing $w_\rho$ if we had taken $\gamma$ to be null. This is consistent with the fact that the normal to a null boundary is known to introduce anomalies in certain situations; see for instance [19].

The field-space exact term in (148) may be recognized as (minus) the field space exterior derivative of a standard Gibbons-Hawking-York (GHY) term

$$\ell_{\text{GHY}} := 2K\epsilon_\gamma .\tag{151}$$

Defining

$$P^{\mu\nu} := (Kh^{\mu\nu} - K^{\mu\nu})\epsilon_\gamma ,\tag{152}$$

we may rewrite (148) compactly as

$$\theta_{\text{inv}}^{\text{EH}}\big|_\gamma = -\delta\ell_{\text{GHY}} - \mathrm{d}(V_{\lrcorner}\epsilon_\gamma) + P^{\mu\nu}\delta h_{\mu\nu} .\tag{153}$$

As a brief aside, note that in our interpretation the invariant part $\theta_{\text{inv}}^{\text{EH}}$ can be thought of as arising from the $U$-covariant part $(\theta_\chi)_U^{\text{c}}$ of the extended symplectic potential on spacetime, pushed forward to relational spacetime $\mathfrak{m}$. From the results of Sec. 3.3, $(\theta_\chi)_U^{\text{c}}$ is related to the bare (non-extended, non-dressed) symplectic potential $\theta^{\text{EH}}$ on spacetime simply via the replacement $\delta \to \delta_\chi$. We then have

$$(\theta_\chi^{\text{EH}})_U^{\text{c}}\Big|_\Gamma = -\delta_\chi\bar{\ell}_{\text{GHY}} - \mathrm{d}(\bar{V}_{\chi\lrcorner}\bar{\epsilon}_\Gamma) ,\tag{154}$$

where the overbar indicates the spacetime fields as opposed to relational spacetime fields, e.g. $\bar{g} = U_\star[g]$ and $\bar{V}_\chi^\mu = \bar{h}^{\mu\nu}\bar{n}^\alpha\delta_\chi\bar{g}_{\nu\alpha}$. The only subtle point in this procedure is that the function $\bar{f}$ on spacetime, whose level sets define $\Gamma$, is not itself a background function on $M$. Rather it is $\gamma = U(\Gamma)$ which is defined by a background function $f$ on relational spacetime, and $\bar{f} = U_\star[f]$ may be taken to be the pullback of this function to $M$. Therefore in principle, the bare potential $\theta^{\text{EH}}\big|_\Gamma$ on spacetime would involve some nonstandard terms arising from the variation of $\bar{f}$, but these terms do not appear in $\theta_U^{\text{c}}\big|_\Gamma$ due to the replacement $\delta \to \delta_\chi$, because $\delta f = U^\star[\delta_\chi\bar{f}] = 0$. We can therefore always ignore such terms in (154).

Returning to the problem of identifying subregion boundary actions, we compare (153) to the decomposition (142), identifying

$$\begin{aligned}
\pi_{\text{inv}}\delta\phi_{\text{inv}} &= P^{\mu\nu}\delta h_{\mu\nu} , \\
\ell_{\text{inv}} &= \ell_{\text{GHY}} , \qquad \text{(Einstein-Hilbert)} \\
\beta_{\text{inv}} &= -V_{\lrcorner}\epsilon_\gamma .
\end{aligned}\tag{155}$$

The condition (143) on the boundary contribution $\ell_{\text{corr}}$ then gives

$$\delta\ell_{\text{corr}} \underset{\gamma}{\approx} \delta\ell_{\text{GHY}} + (y\delta x - P^{\mu\nu}\delta h_{\mu\nu}) - \mathrm{d}(\mathcal{C} - V_{\lrcorner}\epsilon_\gamma - U^\star[q_\chi]) .\tag{156}$$

We can immediately take

$$\mathcal{C} := V_{\lrcorner}\epsilon_\gamma + U^\star[q_\chi] ,\tag{157}$$

and

$$\delta\ell_{\text{corr}} \underset{\gamma}{\approx} \delta\ell_{\text{GHY}} + (y\delta x - P^{\mu\nu}\delta h_{\mu\nu}) .\tag{158}$$

To solve the remaining condition, we seek choices of $x, y$ that render the right-hand side field-space exact. Of course we have the Dirichlet and Neumann cases (see equations (145) and (146)). For the former, we fix the induced metric on $\gamma$ to a background boundary configuration $h_{\mu\nu}^{(0)}$ yielding the action (140) in the form

$$S_\gamma^{\text{Dirichlet}} = \int_\gamma \left(\ell_{\text{GHY}} - P^{\mu\nu}(h_{\mu\nu} - h_{\mu\nu}^{(0)})\right) .\tag{159}$$

For the Neumann case, we can fix $P^{\mu\nu} = P^{\mu\nu}_{(0)}$ on $\gamma$ utilizing

$$S^{\text{Neumann}}_\gamma = \int_\gamma \left( \ell_{\text{GHY}} - h_{\mu\nu} P^{\mu\nu}_{(0)} \right). \tag{160}$$

The choice of $P^{\mu\nu}_{(0)}$ cannot be made arbitrarily, but must solve the necessary constraint equations on $\gamma$.[33] In general this is a nontrivial condition. Another choice [103,104], which may actually be preferable from the point of view of the well-posedness of the boundary value problem [105,106], is to hold fixed the conformal metric on $\gamma$, together with the extrinsic curvature $K$. For these conditions, we write the conformal metric as

$$\tilde{h}_{\mu\nu} = h^{-\frac{1}{d-1}} h_{\mu\nu}, \tag{161}$$

where $h$ is the metric determinant and $d-1$ is the dimension of $\gamma$ in a $d$-dimensional spacetime. Then defining

$$\begin{aligned}
\tilde{P}^{\mu\nu}\tilde{\epsilon} &:= h^{\frac{1}{d-1}} \left( P^{\mu\nu} - (d-1)^{-1} h^{\mu\nu} P \right), \\
P &:= h_{\mu\nu} P^{\mu\nu} = (d-2) K \epsilon_\gamma, \\
\tilde{K} &:= \frac{2(d-2)}{(d-1)} K, \\
\tilde{\epsilon} &:= \frac{\epsilon_\gamma}{\sqrt{h}},
\end{aligned} \tag{162}$$

we rewrite

$$P^{\mu\nu}\delta h_{\mu\nu} = \left( \tilde{P}^{\mu\nu}\delta\tilde{h}_{\mu\nu} + \tilde{K}\delta\sqrt{h} \right)\tilde{\epsilon}, \tag{163}$$

and equation (158) becomes

$$\delta\ell_{\text{corr}} \underset{\gamma}{\approx} \delta\ell_{\text{GHY}} + y\delta x - \left( \tilde{P}^{\mu\nu}\delta\tilde{h}_{\mu\nu} + \tilde{K}\delta\sqrt{h} \right)\tilde{\epsilon}. \tag{164}$$

Thus far the index structure and summation on $y\delta x$ has been left implicit. Here we break it into two parts:

$$y\delta x = \left( \tilde{y}^{\mu\nu}\delta\tilde{x}_{\mu\nu} + \tilde{y}\delta\tilde{x} \right)\tilde{\epsilon}, \tag{165}$$

and choose immediately to hold the conformal metric fixed:

$$\tilde{x}_{\mu\nu} = \tilde{h}_{\mu\nu}, \qquad \tilde{y}^{\mu\nu} = \tilde{P}^{\mu\nu}. \tag{166}$$

Finally, equation (158) becomes

$$\delta\ell_{\text{corr}} \underset{\gamma}{\approx} \delta\ell_{\text{GHY}} + \left( \tilde{y}\delta\tilde{x} - \tilde{K}\delta\sqrt{h} \right)\tilde{\epsilon}. \tag{167}$$

There remains a final choice to fix either the volume factor or its conjugate variable $\tilde{K}$. In fact, as a slight generalization from the usual case, we can choose to fix a linear combination of these. Let

$$\begin{aligned}
\tilde{x} &= A\sqrt{h} + B\tilde{K}, \\
\tilde{y} &= C\sqrt{h} + D\tilde{K},
\end{aligned} \tag{168}$$

---

[33]Namely, in general, one must ensure that the momentum constraints $D_\mu \left( K h^{\mu\nu} - K^{\mu\nu} \right) = 0$ and the scalar constraint $^3R + K^2 - K_{\mu\nu}K^{\mu\nu} = 0$, where $^3R$ is the Ricci scalar curvature of the induced metric on $\gamma$, can be satisfied. In the Neumann case, the momentum constraints directly translate into the conservation equations $D_\mu P^{\mu\nu}_{(0)} = 0$.

with background scalar functions $A, B, C, D$ satisfying

$$AD - BC = 1. \tag{169}$$

Under this condition, we have

$$\left( \tilde{y} \delta \tilde{x} - \tilde{K} \delta \sqrt{h} \right) \tilde{\epsilon} = \frac{1}{2} \delta \left( \tilde{y} \tilde{x} - \tilde{K} \sqrt{h} \right) \tilde{\epsilon}. \tag{170}$$

In combination with the GHY term in (167) this gives

$$\ell_{\text{corr}} = \left( \left( 2 - \frac{d-2}{d-1} \right) K \sqrt{h} + \frac{1}{2} \tilde{y} \tilde{x} \right) \tilde{\epsilon}. \tag{171}$$

In conjunction with the Lagrange multiplier terms imposing the boundary conditions, the full boundary action is then (see equation (140))

$$
\begin{aligned}
S_{\gamma,\text{mixed}}^{\text{GHY}} &= \int_\gamma \left[ \left( \left( 2 - \frac{d-2}{d-1} \right) K \sqrt{h} + \frac{1}{2} \tilde{y} \tilde{x} \right) - \tilde{P}^{\mu\nu} (\tilde{h}_{\mu\nu} - \tilde{h}_{\mu\nu}^{(0)}) - \tilde{y} \left( \tilde{x} - \tilde{x}^{(0)} \right) \right] \tilde{\epsilon} \\
&= \int_\gamma \left[ \left( 2 - \frac{d-2}{d-1} \right) K \sqrt{h} - \frac{1}{2} \tilde{y} \tilde{x} + \tilde{P}^{\mu\nu} \tilde{h}_{\mu\nu}^{(0)} + \tilde{y} \tilde{x}^{(0)} \right] \tilde{\epsilon},
\end{aligned}
\tag{172}
$$

where we have remarked that $\tilde{P}^{\mu\nu}$ is traceless in going to the second line.

## 6.3 Interplay between boundary symmetries and boundary conditions

We have now all the information we need to discuss how the algebras of charges defined in Sec. 5.4 are realized in vacuum general relativity. Given a corner $s$, we introduce the future-pointing unit normal $k$ on $\gamma$, such that $\epsilon_\gamma = k \wedge \epsilon_s$, where $\epsilon_s$ is the induced volume form on $s$. In particular, we have $k_\mu n^\mu = 0$. This can be used to evaluate the pullback of $(\epsilon_\gamma)_\mu := \partial_\mu \lrcorner \epsilon_\gamma$ and $\epsilon_{\mu\nu} := \partial_\mu \lrcorner \partial_\nu \lrcorner \epsilon$ to $s$ as:

$$(\epsilon_\gamma)_\mu \big|_s = k_\mu \epsilon_s, \qquad \epsilon_{\mu\nu} \big|_\gamma = (k_\mu n_\nu - k_\nu n_\mu) \epsilon_s. \tag{173}$$

For any vector field $\rho$ leaving $\gamma$ invariant, we find using (16), (101) and (155) that

$$
\begin{aligned}
Q_\rho(s) &= \int_s \left( U^\star [q_{U_\star \rho}] - \check{\rho} \cdot \beta_{\text{inv}} \right) \\
&= \int_s \left( \epsilon_{\mu\nu} \nabla^\mu \rho^\nu - (\check{\rho} \cdot V) \epsilon_\gamma \right) = \int_s \left( \epsilon_{\mu\nu} \nabla^\mu \rho^\nu + (\epsilon_\gamma)_\mu n_\nu (\nabla^\mu \rho^\nu + \nabla^\nu \rho^\mu) \right),
\end{aligned}
\tag{174}
$$

which, together with (173), evaluates to

$$Q_\rho(s) = 2 \int_s k_\mu n_\nu \nabla^\mu \rho^\nu \epsilon_s = -2 \int_s \rho^\nu k_\mu \nabla^\mu n_\nu \epsilon_s = -2 \int_s k_\mu \rho_\nu K^{\mu\nu} \epsilon_s. \tag{175}$$

Those charges generate a corner algebra (120), indexed by vector fields $\rho$ that are parallel to $s$. Since the expression (175) does not involve the derivatives of $\rho$ in directions normal to $s$, it is found to be isomorphic to the algebra of vector fields intrinsic to $s$, namely $\mathfrak{X}(s)$ [15]. In particular, the surface boosts of [10] are not represented in this charge algebra. Hence, we conclude that, with our presymplectic structure, they do not play any role when standard boundary conditions are being imposed on $\gamma$, by which we mean: boundary conditions that annihilate the presymplectic form on $\gamma$, such that $\delta P^{\mu\nu} \delta h_{\mu\nu} \overset{\approx}{\approx} 0$. We could recover a non-trivial representation of $\text{Diff}(s) \ltimes \text{SL}(2, \mathbb{R})^s$ by ensuring that $Q_\rho(s)$ is given by the Komar expression $\int_s U^\star [q_{U_\star \rho}]$, which corresponds to choosing an alternate decomposition

of $\theta_{\text{inv}}\big|_\gamma = -\delta\ell'_{\text{inv}} + \mathrm{d}\beta'_{\text{inv}} + \pi'_{\text{inv}}\delta\phi'_{\text{inv}}$ such that $\beta'_{\text{inv}} = 0$. In view of relation (104), this would in turn require to impose boundary conditions that make $\delta\theta^{\text{EH}}_{\text{inv}}$ to vanish on $\gamma$. It would be interesting to investigate whether physically interesting boundary conditions of this type can be identified in some contexts while preserving the (formal) well-posedness of the variational problem.

Let us now discuss the additional terms contributing to the charges (107), starting with the cases of **Dirichlet** and **Neumann** boundary conditions, as respectively prescribed by the boundary actions (159) and (160). Including the contribution of $\ell_{\text{GHY}}$ which contributes to (107) in both cases, we find that

$$Q_\rho(s) + \int_s \rho\lrcorner\ell_{\text{GHY}} = 2\int_s k_\mu\rho_\nu T^{\mu\nu}\epsilon_s\,, \qquad T^{\mu\nu} := Kh^{\mu\nu} - K^{\mu\nu}\,. \tag{176}$$

We recognize the expression of the *Brown-York charge* [74],[34] expressed as an integral of the *Brown–York stress tensor* $T^{\mu\nu}$, which we recover here on relational spacetime. In the Dirichlet case, the Hamiltonian charge $Q^{\text{H}}_\rho$ reduces on-shell to the Brown–York expression:

$$Q^{\text{H}}_\rho(s) = 2\int_s k_\mu\rho_\nu\left(T^{\mu\nu} - \frac{1}{2}h^{\mu\nu}(T - T^{\alpha\beta}h^{(0)}_{\alpha\beta})\right)\epsilon_s \approx 2\int_s k_\mu\rho_\nu T^{\mu\nu}\epsilon_s\,. \tag{177}$$

The quantity $K_{\rho_1,\rho_2}(s)$ can also be expressed in terms of the Brown–York stress tensor as

$$K_{\rho_1,\rho_2}(s) = \int_s\left((\rho_2\lrcorner P^{\mu\nu})\mathcal{L}_{\rho_1}h^{(0)}_{\mu\nu} - (\rho_1\lrcorner P^{\mu\nu})\mathcal{L}_{\rho_2}h^{(0)}_{\mu\nu}\right) \tag{178}$$

$$\approx \int_s\left[\left(D^\mu\rho_1^\nu + D^\nu\rho_1^\mu\right)\rho_2^\alpha - \left(D^\mu\rho_2^\nu + D^\nu\rho_2^\mu\right)\rho_1^\alpha\right]T_{\mu\nu}k_\alpha\epsilon_s\,. \tag{179}$$

Specializing to $\rho_1$ and $\rho_2$ that preserve the boundary conditions, and are therefore Killing fields of the background metric $h^{(0)}_{\mu\nu}$, we conclude that the central charges defined in (123) identically vanish.

To discuss Neumann boundary conditions, it is convenient to introduce the densitized Brown–York stress tensor $\tilde{T}^{\mu\nu} := \sqrt{h}T^{\mu\nu}$, in terms of which we can reexpress

$$P^{\mu\nu} \approx \tilde{P}^{\mu\nu}_{(0)} \qquad \Leftrightarrow \qquad \tilde{T}^{\mu\nu} \approx \tilde{T}^{\mu\nu}_{(0)}\,, \tag{180}$$

where $\tilde{T}^{\mu\nu}_{(0)}$ is a background field. The Hamiltonian charges then take the form

$$Q^{\text{H}}_\rho(s) = 2\int_s\left(\rho^\alpha h_{\alpha\nu}\tilde{T}^{\mu\nu} - \frac{1}{2}\rho^\mu h_{\alpha\beta}\tilde{T}^{\alpha\beta}_{(0)}\right)\frac{1}{\sqrt{h}}k_\mu\epsilon_s \approx 2\int_s k_\mu\rho_\nu\left(T^{\mu\nu} - \frac{1}{2}h^{\mu\nu}T\right)\epsilon_s\,, \tag{181}$$

while

$$K_{\rho_1,\rho_2}(s) = \int_s\left(\rho_1\lrcorner\mathcal{L}_{\rho_2}P^{\mu\nu}_{(0)} - \rho_2\lrcorner\mathcal{L}_{\rho_1}P^{\mu\nu}_{(0)}\right)\,. \tag{182}$$

Restricting to vector fields $\rho_1$ and $\rho_2$ that preserve the boundary condition in the last equation, we again find identically vanishing central charges.

In the two examples just described, the Hamiltonian charges generate a representation of the algebra of symmetries of the boundary background fields, without central extension. Moreover, given that the charges computed in (177) and (181) are non-trivial,[35] they are generators of *boundary symmetries* in the nomenclature of [1]. On the other hand, any vector field

---

[34]Up to a factor of $16\pi G$ which was fixed to 1. Note that it is common to find the convention $8\pi G = 1$ in the literature, in which case there is no factor 2 in the expression of the charge.

[35]By "non-trivial", we mean that $\delta Q^{\text{H}}_\rho(s) \not\approx 0$ in both cases. Indeed, we find that $\delta Q^{\text{H}}_\rho(s) \approx 2\int_s k_\mu\rho_\nu\delta T^{\mu\nu}\epsilon_s$ in the Dirichlet example, and $\delta Q^{\text{H}}_\rho(s) \approx 2\int_s\left(\rho^\alpha\delta h_{\alpha\nu}\tilde{T}^{\mu\nu}_{(0)} - \frac{1}{2}\rho^\mu\delta h_{\alpha\beta}\tilde{T}^{\alpha\beta}_{(0)}\right)\frac{1}{\sqrt{h}}k_\mu\epsilon_s$ for Neumann boundary condition. To obtain the second formula, we observed that $\delta\rho^\mu = 0$ and $\delta\left(\frac{1}{\sqrt{h}}k_\mu\epsilon_s\right) = 0$.

$\rho_1$ that is not a symmetry of the background fields, generates a *meta-symmetry*: that is, a symplectomorphism from $\mathcal{S}_{x_0}$ to another leaf in the foliation (74), where both leaves are equipped with the presymplectic structure induced by $\Omega(s)$ on $\mathcal{S}$.[36] Given such a transformation, we have in general $K_{\rho_1,\rho_2}(s) \hat{\approx}\kern-0.8em/\ \ 0$, even if $\rho_2$ itself leaves the boundary conditions invariant. For instance, taking $\rho_1$ to be conformal Killing (but not Killing) with respect to $h^{(0)}_{\mu\nu}$ in (178), and $\rho_2$ to be Killing, we find that $K_{\rho_1,\rho_2}(s) = \frac{2(d-2)}{d-1}\int_s \rho_2^\nu k_\nu D_\mu \rho_1^\mu K \epsilon_s$ is directly sourced by the extrinsic curvature $K$, and does not identically vanish unless $\rho_2$ is parallel to $s$.

Finally, for the mixed boundary conditions defined by the boundary action (172), we find that

$$Q^{\mathrm{H}}_\rho(s) \hat{\approx} -2\int_s k_\mu \rho_\nu T^{\mu\nu}\epsilon_s + \int_s \left[-\frac{1}{2}\tilde{K}\sqrt{h} + \frac{1}{2}\tilde{y}\tilde{x}\right]\frac{1}{\sqrt{h}}\rho^\mu k_\mu \epsilon_s \tag{183}$$

$$= -2\int_s \tilde{T}^{\mu\nu}h_{\alpha\nu}\frac{1}{\sqrt{h}}k_\mu\rho^\alpha\epsilon_s + \int_s\left[-\frac{1}{2}\tilde{K}\sqrt{h} + \frac{1}{2}\tilde{y}\tilde{x}\right]\frac{1}{\sqrt{h}}\rho^\mu k_\mu\epsilon_s\,, \tag{184}$$

and

$$K_{\rho_1,\rho_2}(s) = \int_s\left[-\left(\tilde{P}^{\mu\nu}\mathcal{L}_{\rho_2}\tilde{h}^{(0)}_{\mu\nu} + \tilde{y}\mathcal{L}_{\rho_2}\tilde{x}^{(0)}\right)\rho_1^\alpha + \left(\tilde{P}^{\mu\nu}\mathcal{L}_{\rho_1}\tilde{h}^{(0)}_{\mu\nu} + \tilde{y}\mathcal{L}_{\rho_1}\tilde{x}^{(0)}\right)\rho_2^\alpha\right]\frac{1}{\sqrt{h}}k_\alpha\epsilon_s\,. \tag{185}$$

When $A = D = 1$ and $B = C = 0$, we recover Dirichlet boundary conditions and expressions consistent with (177) and (178). As a second example, we can take $A = D = 0$ and $B = -D = 1$ in equation (168), which leads to $\tilde{x} = \tilde{K}$ and $\tilde{y} = -\sqrt{h}$. In this polarization, the conformal metric $\tilde{h}_{\mu\nu}$ and the extrinsic scalar curvature $K$ are both held fixed on $\gamma$, and $Q^{\mathrm{H}}_\rho(s)$ is sourced by the traceless part of the Brown-York stress-tensor:

$$Q^{\mathrm{H}}_\rho(s) \hat{\approx} -2\int_s k_\mu\rho_\nu\left(T^{\mu\nu} - \frac{1}{d-1}h^{\mu\nu}T\right)\epsilon_s\,. \tag{186}$$

In this example, boundary symmetries are generated by conformal vector fields on $\gamma$ that also preserve $K$.

Before closing this section, we note that *boundary gauge symmetries* do not seem to naturally occur in general relativity. This is to be contrasted with gauge theories, where they do [1]. For instance, in Yang-Mills theory, it suffices to impose a Neumann boundary condition[37] to trivialize the charge. Any non-trivial symmetry of this boundary condition must then generate a gauge transformation on $\mathcal{S}_{x_0}$. We have not found a similarly simple mechanism in general relativity, but we cannot exclude that such accidental gauge symmetries might play a role in more complicated situations.

## 7 Discussion

We have implemented the strategy of post-selection outlined in [1] to derive symplectic structure and actions appropriate to subregions, viewed as subsystems within a global diffeomorphism invariant theory. The use of a dynamical frame field $U$, both to define the subregion itself, and to parse physically meaningful observables, played a central role in this analysis. At this point we pause to relate our choice of symplectic structure to various alternatives which have appeared in other recent literature.

---

[36]It is crucial here that $\Omega(s)$ only depends on the foliation by $x$, not on the background field entering the boundary condition $x \hat{\approx} x_0$.

[37]That is, to fix the pullback of (the edge mode dressing and hence invariant form of) $\star F$ to the boundary, where $F$ is the curvature of the connection.

We first recall some pertinent facts, which motivate an extended phase space. Starting from some anomaly-free Lagrangian $L$, satisfying the relation $\delta L = E + \mathrm{d}\theta$ on spacetime,[38] any vector field $\xi$ generates an infinitesimal diffeomorphism with associated current $J_\xi$, which is exact on shell (cf. (11-13)):

$$J_\xi := \hat{\xi} \cdot \theta - \xi \lrcorner L = C_\xi + \mathrm{d}q_\xi \approx \mathrm{d}q_\xi \, .$$

The contraction of field-independent $\hat{\xi}$ with the standard (pre)symplectic[39] form $\Omega_\Sigma = \int_\Sigma \delta\theta$ in the absence of any embedding fields or extended phase space, results in

$$-\hat{\xi} \cdot \Omega_\Sigma = \int_\Sigma \left( \delta J_\xi + \xi \lrcorner E \right) - \int_{\partial\Sigma} \xi \lrcorner \theta \approx \int_{\partial\Sigma} \left( \delta q_\xi - \xi \lrcorner \theta \right) \, . \tag{187}$$

For $\xi$ that vanishes sufficiently rapidly near the boundary $\partial\Sigma$, the right-hand side vanishes on shell, and these transformations represent pure gauge transformations. By contrast, when $q_\xi$ is nonvanishing on $\partial\Sigma$ it provides a nontrivial symmetry charge (though elsewhere in this work we reserve the term "symmetry" for transformations on relational spacetime). However if the right-hand side of (187) fails to be field-space exact, the transformation does not correspond to a Hamiltonian generator. The non-exactness is controlled entirely by the third term, which is nonzero if $\theta$ fails to vanish at the boundary and $\xi$ has nonvanishing normal component to $\partial\Sigma$. This non-integrability may be unsurprising, as such transformations shift the bounding surface and render the system open. Nevertheless these transformations can be canonically incorporated into the subregion's algebra of symmetries and/or constraints by considering an extended phase space which includes embedding fields.

## 7.1 Alternative symplectic potentials

Donnelly and Freidel [10] considered an extended symplectic structure with the inclusion of embedding fields which locate the subregion and boundary in some reference frame. Many other works have since explored related choices of symplectic structure, utilizing embedding fields to extend the phase space [11, 13, 20, 23, 24, 48]. We will refer collectively to these extended symplectic structures with a subscript "ext". Given a choice of extended symplectic potential $\theta_{\mathrm{ext}}$, one obtains a corresponding extended symplectic current $\omega_{\mathrm{ext}}$, extended potential form $\Theta_{\mathrm{ext}}(\sigma)$, and extended symplectic form $\Omega_{\mathrm{ext}}(\sigma)$ through the relations

$$\omega_{\mathrm{ext}} = \delta_\chi \theta_{\mathrm{ext}} \, , \tag{188}$$

$$\Theta_{\mathrm{ext}}(\sigma) = \int_\sigma U^\star[\theta_{\mathrm{ext}}] = \int_\Sigma \theta_{\mathrm{ext}} \, , \tag{189}$$

$$\Omega_{\mathrm{ext}}(\sigma) = \delta\Theta_{\mathrm{ext}}(\sigma) = \int_\sigma U^\star[\omega_{\mathrm{ext}}] = \int_\Sigma \omega_{\mathrm{ext}} \, , \tag{190}$$

where $\delta_\chi := \delta + \mathcal{L}_\chi$ is the extended field-space covariant derivative with flat connection ($\delta_\chi^2 = 0$) and $\chi$ is the Maurer-Cartan form $\chi := \delta U^{-1} \circ U$. We have here made explicit that the resulting forms might be considered as corresponding to either the relational Cauchy surface $\sigma$, or to the corresponding spacetime surface $\Sigma$ because of the equivalence of integrating (for instance) $\omega_{\mathrm{ext}}$ on $\Sigma$, versus $U^\star[\omega_{\mathrm{ext}}]$ on $\sigma$. The crucial point is that in the variational principle,

---

[38]Ambiguities of this procedure will be discussed in Sec. 7.3.

[39]To obtain the true symplectic form on the phase space from the pre-symplectic form, one must project out degenerate directions of the latter. The identification of such degenerate directions is a primary task in the ensuing analysis, but for brevity we will usually neglect the prefix "pre" except where directly relevant.

a field-space variation passes freely over $\int_\sigma$, but not $\int_\Sigma$. For this reason we label the forms by the corresponding relational manifolds, treating these as primary.[40]

The original work of Donnelly and Freidel [10] was in the context of vacuum general relativity with vanishing cosmological constant. Using diffeomorphism invariance as a guiding principle, they arrive at an additional boundary term in their extended symplectic potential, which in our notation we write as

$$\theta^{\text{DF}} := \theta + \mathrm{d}q_\chi \,.^{[41]} \tag{191}$$

Speranza [11] generalized this to other diffeomorphism invariant theories in a manner more amenable to the case that the on-shell bulk Lagrangian is nonvanishing. The potential chosen there was

$$\theta^{\text{DFS}} := \theta + \hat{\chi} \cdot \theta \,, \tag{192}$$

and the corresponding symplectic form gives

$$\hat{\xi} \cdot \Omega^{\text{DFS}}(\sigma) = 0 \,. \tag{193}$$

Therefore one finds that *all* diffeomorphisms, even field-dependent, correspond to gauge symmetries. In fact, the charges vanish *identically* and are thus not (non-trivially) proportional to the constraints, in contrast to our (88) and (93). A new family of nontrivial symmetries is available for these choices, however, by considering transformations acting not on fundamental fields but only on the embedding map $U$. In our terminology, this comprises the action of diffeomorphisms on relational spacetime $\mathfrak{m}$, as opposed to those on spacetime $\mathcal{M}$. As pointed out in [25], the resulting charges and integrability conditions are equivalent to those of (187), as can be seen by making a particular gauge choice for the embedding field. So in fact the phase space has only been extended by an enlarged gauge symmetry.

An alternative choice of symplectic structure for the extended phase space was suggested by Ciambelli, Leigh, and Pai [23], and Freidel [24]:[42]

$$\theta^{\text{CLPF}} = \theta + \chi \lrcorner L \,, \tag{194}$$

leading to

$$-\hat{\xi} \cdot \Omega^{\text{CLPF}}(\sigma) = \delta \int_\sigma U^\star \left[ J_\xi \right] - \int_\sigma U^\star \left[ J_{\delta\xi} \right] \tag{195}$$

$$\approx \delta \int_{\partial\sigma} U^\star \left[ q_\xi \right] - \int_{\partial\sigma} U^\star \left[ q_{\delta\xi} \right] \tag{196}$$

Interestingly, all field-independent diffeomorphisms now correspond to integrable charges, regardless of their behavior near the boundary.

---

[40]We follow the nomenclature introduced in Sec. 2, that any entity on spacetime $\mathcal{M}$ may be pushed forward to "relational spacetime" $\mathfrak{m}$ via the map $U : \mathcal{M} \to \mathfrak{m}$. We consider $U$ as a type of dressing that moves gauge covariant objects to an invariant space. Note, however, that many other works establish a similar map in the opposite direction (more properly "embedding" submanifolds within spacetime). Since the map is assumed invertible, it is of course a matter of convention. We also denote the pushforward by $U^\star$ and the pullback $U_\star$. If the opposite convention is chosen on both matters, the result is that formulas look the same, but the words applied differ. Of course the conceptual content is unchanged.

[41]Since the relation $\delta L = E + \mathrm{d}\theta$ leaves some ambiguity in the choice of $\theta$, here and elsewhere in this section we only insist that $\theta$ denotes a "bare" potential, i.e. that it has no dependence on $\chi$. Ambiguities will be discussed in 7.3.

[42]Note that in Sec. 3.3 we have taken the same form, $\theta + \chi \lrcorner L$, as our starting point for a bulk symplectic potential (denoting it $\theta_\chi$), and then proceeded to include boundary terms which render the full symplectic form independent of the choice of Cauchy slice through a subregion $m$. Likewise, the DFS choice in (192) is what we have called the $U$-covariant part of the symplectic potential $\theta_U^c = \theta + \hat{\chi} \cdot \theta$.

In Table 1, we have summarized the behavior under spacetime diffeomorphisms of the choices of symplectic structure resulting from (191), (192), and (194). We have also included a row for the choice

$$\tilde{\theta}_m = \theta + \chi \lrcorner L + \mathrm{d}q_\chi, \tag{197}$$

which is related to our choice of potential in Sec. 4 (see equations (78) and (82)) and will be discussed further throughout the rest of this work. The full potential utilized there also includes boundary terms enforcing conservation along a particular timelike boundary, but $\tilde{\theta}_m$ is the part that is independent of this choice of timelike boundary. We will discuss the relevance of these additional terms in Sec. 7.3, but for now note that they do not alter the results for spacetime diffeomorphisms from $\tilde{\theta}_m$. To interpret Table 1 recall that for each $\theta_{\mathrm{ext}}$ considered, the corresponding presymplectic structure is obtained as in equations (188) through (190). Particularly, the corresponding extended symplectic current is $\omega_{\mathrm{ext}} = \delta_\chi \theta_{\mathrm{ext}}$ as opposed to $\delta \theta_{\mathrm{ext}}$, and the relevant field-space Lie derivative is $\mathsf{L}^\chi_{\hat{\xi}}$, defined in (37), because (for example)

$$\mathsf{L}_{\hat{\xi}} U^\star[\theta_{\mathrm{ext}}] = [\delta, \hat{\xi}\cdot]_+ U^\star[\theta_{\mathrm{ext}}] = U^\star\big[[\delta_\chi, \hat{\xi}\cdot]_+ \theta_{\mathrm{ext}}\big] = U^\star[\mathsf{L}^\chi_{\hat{\xi}} \theta_{\mathrm{ext}}]. \tag{198}$$

Table 1: Various choices for extended symplectic potential currents and their properties with respect to local spacetime diffeomorphisms are shown. Results in the $\theta^{\mathrm{DF}}$ row are abbreviated in terms of the quantity $B(\xi) = C_\xi + \xi \lrcorner L$, though it is important to note that the original use of this form [10] was only in the context of vacuum general relativity with vanishing cosmological constant, where the term proportional to the Lagrangian vanishes on shell. The form $\tilde{\theta}_m$ is related to the choice utilized in the present work (see Sec. 4), though the latter also includes terms enforcing the conservation of the symplectic form along a particular timelike boundary. The form $\tilde{\theta}_m$ is the piece that is independent of this choice, though the inclusion of those terms does not alter the results in the table.

| | $\theta_{\mathrm{ext}}$ | $\hat{\xi} \cdot \theta_{\mathrm{ext}}$ | $\mathsf{L}^\chi_{\hat{\xi}} \theta_{\mathrm{ext}}$ | $-\hat{\xi} \cdot \omega_{\mathrm{ext}}$ | $\mathsf{L}^\chi_{\hat{\xi}} \omega_{\mathrm{ext}}$ |
|---|---|---|---|---|---|
| $\theta^{\mathrm{DF}}$ | $\theta + \mathrm{d}q_\chi$ | $B(\xi)$ | $B(\delta\xi)$ | $\delta_\chi B(\xi) - B(\delta\xi)$ | $\delta_\chi B(\delta\xi)$ |
| $\theta^{\mathrm{DFS}}$ | $\theta + \hat{\chi} \cdot \theta$ | $0$ | $0$ | $0$ | $0$ |
| $\theta^{\mathrm{CLPF}}$ | $\theta + \chi \lrcorner L$ | $J_\xi$ | $J_{\delta\xi}$ | $\delta_\chi J_\xi - J_{\delta\xi}$ | $\delta_\chi J_{\delta\xi}$ |
| $\tilde{\theta}_m$ | $\theta + \chi \lrcorner L + \mathrm{d}q_\chi$ | $C_\xi$ | $C_{\delta\xi}$ | $\delta_\chi C_\xi - C_{\delta\xi}$ | $\delta_\chi C_{\delta\xi}$ |

Note that all the entries in the last three columns can be expressed simply in terms of the result for $\hat{\xi} \cdot \theta_{\mathrm{ext}}$. For instance, using the fact that $[\hat{\xi}, \delta_\chi]_+ = \mathsf{L}_{\hat{\xi}} - \mathcal{L}_\xi = \Delta_\xi + \widehat{\delta\xi}\cdot$, the entries in the $-\hat{\xi} \cdot \omega_{\mathrm{ext}}$ column can all be expressed as

$$-\hat{\xi} \cdot \omega_{\mathrm{ext}} = \delta_\chi(\hat{\xi} \cdot \theta_{\mathrm{ext}}) - \widehat{\delta\xi} \cdot \theta_{\mathrm{ext}} \tag{199}$$

for *any* extended potential, so long as it is anomaly-free. This entails that field-independent $\xi$ are *always* associated with integrable charges given by $\int_\Sigma \hat{\xi} \cdot \theta_{\mathrm{ext}}$. Manipulations identical to those of (92) then show that such charges always satisfy a closed algebra anti-holomorphic in the spacetime vector fields, since the sole anomalous element in such charges is the vector field itself.

Considering now the $\mathsf{L}^\chi_{\hat{\xi}} \theta_{\mathrm{ext}}$ column, these entries illustrate the general fact that for *any* anomaly-free $\theta_{\mathrm{ext}}$ (or even the bare potential $\theta$), the resulting presymplectic form $\Theta_{\mathrm{ext}} = \int_\sigma U^\star[\theta_{\mathrm{ext}}]$ satisfies the gauge invariance condition $\mathsf{L}_{\hat{\xi}} \Theta_{\mathrm{ext}} = 0$ if the generators are

field-independent. This follows trivially from the pushforward to relational spacetime (which can be thought of as shifting the integration domain to negate the effect of the pullback), and is insensitive to the choice of $\chi$-dependent terms. Invariance under *field-dependent* generators ($\delta\xi \neq 0$) is a much stronger condition, and the various extended potentials behave distinctly in this regard. The choice $\theta^{\text{DF}}$ is on-shell invariant only if the Lagrangian vanishes. The choice $\theta^{\text{DFS}}$ is off-shell invariant, trivially, due to vanishing charges. The choice $\theta^{\text{CLPF}}$ is on-shell invariant only if the charge aspect vanishes. Lastly, the choice $\tilde{\theta}_m$ is always on-shell invariant.[43] Analogous statements hold for the on-shell gauge invariance of $\Omega_{\text{ext}}$ as seen from the last column. In particular, the only presymplectic structure that is non-trivially preserved under arbitrary field-dependent gauge diffeomorphisms, $L_{\hat{\xi}}\Omega_{\text{ext}} = 0$, for general Lagrangians is the one produced by $\delta_\chi \tilde{\theta}_m$ in the last row.

Although the various choices in Table 1 may be appropriate to different contexts, the point of view taken in this work is that the symplectic structure for a subregion, viewed as a subsystem of a larger diffeomorphism invariant theory (and irrespective of the asymptotic structure of that global theory), ought to retain the fact that spacetime diffeomorphisms acting locally at the subregion boundary are pure gauge transformations. It should also leave intact a nontrivial off-shell representation of the algebra of spacetime vector fields as the generators of these diffeomorphisms. For this purpose the appearance of the constraints $C_\xi$ in the $\tilde{\theta}_m$ column of Table 1 is crucial. In view of constructing a quantum theory for the subregion, it may be beneficial to have a proper constraint algebra at hand for which one can then attempt to find quantum representations. As pointed out at the end of Sec. 5.1, our construction thereby mirrors the achievements of Isham and Kuchar [72,73] in their work on canonical geometrodynamics, which similarly obtains a nontrivial off-shell representation of the full Lie algebra of spacetime diffeormorphisms through a phase space extension with embedding fields, though their focus was not on subregions.

With this proliferation of symplectic potentials, it is interesting to consider whether they arise by treating the fundamental fields $\Phi$ on spacetime, or the dressed fields $U^\star[\Phi]$ on relational spacetime, as primary. The variation of the dressed Lagrangian may be written in multiple ways suggesting different choices of extended symplectic form:

$$\delta U^\star[L] = U^\star[E]\delta U^\star[\Phi] + dU^\star[\theta + \hat{\chi} \cdot \theta]$$
$$\implies \theta_{\text{ext}} = \theta^{\text{DFS}} = \theta + \hat{\chi} \cdot \theta \,, \tag{200}$$

or

$$\delta U^\star[L] = U^\star[E\delta\Phi] + dU^\star[\theta + \chi \lrcorner L]$$
$$\implies \theta_{\text{ext}} = \theta^{\text{CLPF}} = \theta + \chi \lrcorner L \,. \tag{201}$$

Here we make explicit that the equations of motion term is a field-space one-form. Of course, the resulting equations of motion are unaffected under these two rewritings. From the perspective of this work, the status of $U$ as a dynamical reference implies that its dynamics is inherited solely through its dependence on the fundamental spacetime fields $\Phi$ (possibly including fields outside the subregion of interest). This could be taken to justify taking $\delta\Phi$ as the fundamental variation and utilizing $\theta^{\text{CLPF}}$. But the result of Sec. 4 is that, even taking $\theta^{\text{CLPF}}$ as a starting point, by considering an invariantly defined subregion $m = U(M)$ with boundary conditions on $\gamma \subset \partial m$, the conservation of symplectic structure demands a boundary contribution at least including $q_\chi$ (see equations (219), (221)). Taken alone, this term restores on-shell equivalence with $\Omega^{\text{DFS}}$. In a sense, the primacy of relational spacetime quantities is thus enforced. Note however that our choice $\Omega(\sigma)$ still differs from $\Omega^{\text{DFS}}$ off-shell through the constraint term $C_\xi$, as well as the possible additional boundary contributions which will be revisited in Sec. 7.3.

---

[43]Recall that this does not change with the inclusion of a $U$-covariant $\beta_U^c$ term.

The resulting symplectic form at least shares the common feature with $\Omega^{\text{DFS}}$ that all spacetime diffeomorphisms are returned to pure gauge. This is a marked contrast with the results that come by taking $\theta^{\text{CLPF}}$ as the final extended potential. In that case, the fact that all spacetime vector fields with nonvanishing $q_\xi$ are represented by integral charges is suggestive, but it is also mysterious, since these are certainly gauge directions of the global presymplectic potential. The interpretation we adopt here is that these should remain gauge, while any physical charges and symmetries associated with a true gravitational subsystem should arise through consideration of frame reorientations (transformations on relational spacetime) which we will consider next.

## 7.2 Relational spacetime symmetries

In addition to the standard spacetime diffeomorphisms, for each of the extended symplectic structures discussed above, we may consider the action of diffeomorphisms on relational spacetime $\mathfrak{m}$. These are "frame reorientations" acting on the map $U$ alone, and not on the spacetime fields. The resulting integrability conditions, charges, and algebra were discussed in detail in Sec. 5.4 for our choice of symplectic structure $\Omega(\sigma)$. Here we catalogue the corresponding charges and integrability conditions associated to alternative extended structures.

We start with a field-independent vector field $\rho \in \mathfrak{X}(\mathfrak{m})$ generating an infinitesimal diffeomorphism on $\mathfrak{m}$. In this section, we will denote the pullback of $\rho$ to $\mathcal{M}$ with an overbar:

$$\bar{\rho} := U_\star \rho \,. \tag{202}$$

Recall that $\breve{\rho}$ acts trivially on all fundamental fields $\Phi$, and its contraction with $\chi$ returns the corresponding vector on spacetime:

$$\begin{aligned} \breve{\rho} \cdot \delta\Phi &= 0 \,, \\ \breve{\rho} \cdot \chi &= \bar{\rho} \,. \end{aligned} \tag{203}$$

Results are collected in Table 2. For comparison, these are placed alongside the corresponding expressions for field-independent spacetime diffeomorphisms generated by $\xi \in \mathfrak{X}(\mathcal{M})$ (redundant with the second-to-last column of Table 1).

Table 2: Various extended presymplectic potentials and their properties are shown with respect to *relational spacetime* diffeomorphisms generated by field-independent $\rho \in \mathfrak{X}(\mathfrak{m})$, which act not on fundamental fields but only on the frame $U$. We denote $\bar{\rho} := U_\star \rho$. For comparison, the final column shows the behavior under spacetime diffeomorphisms generated by field-independent $\xi \in \mathfrak{X}(\mathcal{M})$. The results for $\tilde{\theta}_m$ are included here for completeness, though under relational spacetime diffeomorphisms they do *not* actually coincide with our full choice of symplectic structure $\Omega_m$ due to additional boundary terms included in the latter (see Sec. 7.3).

| | $\theta_{\text{ext}}$ | $-\breve{\rho} \cdot \omega_{\text{ext}}$ | $-\hat{\xi} \cdot \omega_{\text{ext}}$ |
|---|---|---|---|
| $\theta^{\text{DF}}$ | $\theta + \mathrm{d}q_\chi$ | $-\mathrm{d}\left(\bar{\rho} \lrcorner \theta_{\text{ext}}\right) + \bar{\rho} \lrcorner \left(E + \mathrm{d}\chi \lrcorner L\right) + \delta_\chi\left(\mathrm{d}q_{\bar{\rho}} - \bar{\rho} \lrcorner L\right)$ | $\delta_\chi\left(C_\xi + \xi \lrcorner L\right)$ |
| $\theta^{\text{DFS}}$ | $\theta + \hat{\chi} \cdot \theta$ | $-\mathrm{d}\left(\bar{\rho} \lrcorner \theta_{\text{ext}}\right) + \bar{\rho} \lrcorner \left(E + \hat{\chi} \cdot E\right) + \delta_\chi J_{\bar{\rho}}$ | $0$ |
| $\theta^{\text{CLPF}}$ | $\theta + \chi \lrcorner L$ | $-\mathrm{d}\left(\bar{\rho} \lrcorner \theta_{\text{ext}}\right) + \bar{\rho} \lrcorner E$ | $\delta_\chi\left(C_\xi + \mathrm{d}q_\xi\right)$ |
| $\tilde{\theta}_m$ | $\theta + \chi \lrcorner L + \mathrm{d}q_\chi$ | $-\mathrm{d}\left(\bar{\rho} \lrcorner \theta_{\text{ext}}\right) + \bar{\rho} \lrcorner E + \delta_\chi \mathrm{d}q_{\bar{\rho}}$ | $\delta_\chi C_\xi$ |

The analogue of expression (199) for relational spacetime diffeomorphisms is

$$-\check{\rho} \cdot \omega_{\text{ext}} = \delta_\chi (\check{\rho} \cdot \theta_{\text{ext}}) - \widetilde{\delta\rho} \cdot \theta_{\text{ext}} - \mathcal{L}_{\bar{\rho}} \theta_{\text{ext}}. \tag{204}$$

Rewritings of the last term for the various choices of $\theta_{\text{ext}}$ (as well as restriction to field-independent generators $\delta\rho = 0$) are responsible for the diversified expressions in table 2. Note that with the exception of the first row, the on-shell expressions for $-\check{\rho} \cdot \omega_{\text{ext}}$ all reduce to a boundary contribution (and even for the first row, if the Lagrangian vanishes), and they all including a term of the form $-\mathrm{d}(\bar{\rho} \lrcorner \theta_{\text{ext}})$ which controls the question of integrability. Of course, the actual expression for $\theta_{\text{ext}}$ differs in each row. This situation mirrors the case of the unextended symplectic structure under the action of spacetime diffeomorphisms (see equation (187) and surrounding discussion). As pointed out in [25] in the case of $\theta^{\text{DFS}}$, the similarity is no coincidence. For that choice, the new found gauge freedom under arbitrary spacetime diffeomorphisms can be used to make a gauge choice setting $\chi = 0$, which renders the comparison even more explicit by sending $\theta_{\text{ext}} \to \theta$. The case of $\theta^{\text{CLPF}}$ differs in that spacetime diffeomorphisms at the boundary can carry nonvanishing, integrable charges. The corresponding $\xi$ cannot be utilized in gauge fixing. This choice instead has additional gauge directions under relational transformations generated by $\rho$ which preserve the corner $\partial\sigma$ (noting that a charge aspect term always appears on one side or the other in table 2). We will revisit this point in section 7.3. The CLPF choice generically has no nontrivial charges on the relational spacetime side unless, working on-shell of some boundary condition, the term $\bar{\rho} \lrcorner \theta_{\text{ext}}$ is found to be field-space exact (see discussion under equation (108)).

Finally we note that the results for $\tilde{\theta}_m$ under relational spacetime diffeomorphisms do not actually coincide with those for our full choice of symplectic structure $\Omega(\sigma)$, due to additional boundary contributions in the latter. We include the row for $\tilde{\theta}_m$ here merely for completeness. Our full choice will be discussed in the following subsection.

## 7.3 Ambiguities and their resolution

Early implementations of the covariant phase space formalism due to Iyer and Wald are subject to certain ambiguities in the definitions of key objects [6, 8]. In particular, shifts of the form

$$\begin{aligned} L &\to L + \mathrm{d}a \,, \\ \theta &\to \theta + \delta a + \mathrm{d}b \,, \end{aligned} \tag{205}$$

do not affect the equations of motion or alter the key relation $\delta L = E + \mathrm{d}\theta$. However they can affect the expressions for Hamiltonian charges. As emphasized by Speranza [25], the extended potentials $\theta^{\text{DFS}}$ and $\theta^{\text{CLPF}}$ differ on shell by just such an exact term, $\mathrm{d}q_\chi$. Such ambiguities can be resolved [14, 19, 107–109] if the corner surface $\partial\Sigma$ is considered as a cut of a hypersurface $\Gamma$ bounding a spacetime region $M$, as in the present work. In the standard (unextended) case, the resolution proceeds by choosing a decomposition of the symplectic potential on $\Gamma$ as

$$\theta\big|_\Gamma = -\delta\bar{\ell} + \mathrm{d}\beta + \varepsilon \,, {}^{44} \tag{206}$$

where $\varepsilon$ is required to vanish when boundary conditions are imposed. The quantity $\bar{\ell}$ is then added to the Lagrangian as a boundary contribution for the subregion bounded by $\Gamma$, leading to

$$S = \int_M L + \int_\Gamma \bar{\ell} \,, \tag{207}$$

---

[44]We denote the quantity $\bar{\ell}$ with an overbar here merely to distinguish it from boundary terms defined on relational spacetime discussed in previous sections.

and the symplectic form is then modified to

$$\Omega = \int_\Sigma \omega - \delta \int_{\partial\Sigma} \beta \,. \tag{208}$$

The resulting symplectic form is conserved on shell (independent of the choice of $\Sigma$), and the corresponding charges are free from ambiguities resulting from shifts of the form (205). Given such a decomposition (206) for the bare symplectic potential $\theta|_\Gamma$, Speranza [25] identifies a related decomposition for the extended potential $\theta^{\mathrm{CLPF}}$ on the boundary $\gamma = U(\Gamma)$:

$$U^\star[\theta^{\mathrm{CLPF}}] \underset{\gamma}{=} U^\star[\theta + \chi_{\lrcorner}L] \tag{209}$$

$$\underset{\gamma}{=} U^\star[-\delta\bar{\ell} + \mathrm{d}\beta + \varepsilon + \chi_{\lrcorner}L] \tag{210}$$

$$\underset{\gamma}{=} \delta U^\star[-\bar{\ell}] + \mathrm{d}U^\star[\beta + \chi_{\lrcorner}\bar{\ell}] + U^\star\left[\varepsilon + \chi_{\lrcorner}(\mathrm{d}\bar{\ell} + L)\right] \tag{211}$$

and so identifying

$$\ell^{\mathrm{S}}_{\mathrm{ext}} := U^\star[\bar{\ell}] \tag{212}$$

$$\beta^{\mathrm{S}}_{\mathrm{ext}} := U^\star[\beta + \chi_{\lrcorner}\bar{\ell}] \tag{213}$$

$$\varepsilon^{\mathrm{S}}_{\mathrm{ext}} := U^\star\left[\varepsilon + \chi_{\lrcorner}(\mathrm{d}\bar{\ell} + L)\right] \tag{214}$$

with corresponding conserved symplectic structure

$$\Omega^{\mathrm{S}} := \Omega^{\mathrm{CLPF}}(\sigma) - \int_{\partial\sigma} \delta U^\star[\beta + \chi_{\lrcorner}\bar{\ell}] \,. \tag{215}$$

This symplectic structure is appropriate to the case that (214) vanishes under the boundary conditions, and particularly $\chi$ is set to zero on the boundary. The details of this symplectic structure are discussed in [25].

The procedure that we have outlined in this work, as an extension of the procedure in [1], may be considered as an alternative choice for this decomposition. We began by splitting the symplectic potential $\theta^{\mathrm{CLPF}}$ into $U$-covariant and non $U$-covariant parts:

$$\begin{aligned}
U^\star[\theta^{\mathrm{CLPF}}] &= U^\star[\theta + \chi_{\lrcorner}L] \\
&= U^\star[(\theta + \hat{\chi} \cdot \theta) + (-\hat{\chi} \cdot \theta + \chi_{\lrcorner}L)] \\
&= U^\star[\theta^{\mathrm{c}}_U - J_\chi] \,.
\end{aligned} \tag{216}$$

Then the spacetime $U$-covariant part $\theta^{\mathrm{c}}_U$ is decomposed on $\Gamma$ as

$$\theta^{\mathrm{c}}_U\big|_\Gamma = -\delta_\chi \ell^{\mathrm{c}}_U + \mathrm{d}\beta^{\mathrm{c}}_U + \varepsilon^{\mathrm{c}}_U \,. \tag{217}$$

This decomposition may directly mirror that taken for the bare symplectic potential in (206), but requiring the $U$-covariant part $\varepsilon^{\mathrm{c}}_U$ to vanish if boundary conditions are imposed, as opposed to $\varepsilon$ itself.[45] This is appropriate when boundary conditions are to be imposed only on gauge-invariant combinations of $\chi$ and fundamental fields on $\gamma = U(\Gamma)$, while $\chi$ does not have a boundary condition of its own. The gauge part of (216) then divides naturally between extended $\beta$ and $\varepsilon$ terms by writing $J_\chi = \mathrm{d}q_\chi + C_\chi$:

$$\ell_{\mathrm{ext}} := U^\star[\ell^{\mathrm{c}}_U] \,, \tag{218}$$

$$\beta_{\mathrm{ext}} := U^\star[\beta^{\mathrm{c}}_U - q_\chi] \,, \tag{219}$$

$$\varepsilon_{\mathrm{ext}} := U^\star[\varepsilon^{\mathrm{c}}_U - C_\chi] \,. \tag{220}$$

---

[45]Since $\ell^{\mathrm{c}}_U$ is a field space 0-form, the notation indicating its $U$-covariance does not entail any dependence on $\chi$. We keep the label as an explicit reminder that it arises from a choice of decomposition of $\theta^{\mathrm{c}}_U\big|_\Gamma$. Also note that at this stage of discussion, if $\ell^{\mathrm{c}}_U$ is added to the subregion boundary Lagrangian as in (207), it does not yet include a Lagrange multiplier component which we use in section 6 to impose the boundary conditions dynamically.

This leads to the modified conserved symplectic structure utilized in this work, written in equation (84) and denoted $\Omega_m$ when evaluated on shell. This may also be written as

$$\Omega(\sigma) = \Omega^{\text{CLPF}}(\sigma) + \int_{\partial\sigma} \delta U^\star \left[ q_\chi - \beta_U^{\text{c}} \right] \approx \Omega_m \,. \tag{221}$$

This symplectic structure is ambiguity-free in the same sense that (215) is; once a quantity $\varepsilon_U^{\text{c}}$ has been chosen to vanish at the boundary on the relevant leaf of solution space (under a particular type of boundary condition), the resulting symplectic form does not change if the Lagrangian is shifted by spacetime exact terms, or the bare potential by a field-space exact term. Furthermore, the form $\Omega(\sigma)$ is independent of the choice of boundary conditions in the following sense. Suppose a decompositon has been chosen of the form

$$\begin{aligned}
\theta_{\text{inv}}\big|_\gamma &= -\delta\ell_{\text{inv}} + \mathrm{d}\beta_{\text{inv}} + \varepsilon_{\text{inv}} \,, \\
\varepsilon_{\text{inv}} &= \pi_{\text{inv}}\delta\phi_{\text{inv}} \,,
\end{aligned} \tag{222}$$

where we now denote the pushforward of the $U$-covariant forms to $\mathfrak{m}$ as invariant: $\ell_{\text{inv}} := U^\star[\ell_U^{\text{c}}]$, $\beta_{\text{inv}} := U^\star[\beta_U^{\text{c}}]$, and $\varepsilon_{\text{inv}} := U^\star[\varepsilon_U^{\text{c}}]$. Suppose $\pi_{\text{inv}}$ and $\phi_{\text{inv}}$ are conjugate pairs of degrees of freedom, and fixing Dirichlet boundary conditions of the form $\phi_{\text{inv}} = \phi_{\text{inv}}^{(0)}$ defines a well-posed dynamical problem for the subregion $m$. Any alternate choice of boundary condition that amounts to a canonical transformation relative to the symplectic current $\delta\varepsilon_{\text{inv}}$ only involves shuffling terms between $\varepsilon_{\text{inv}}$ and $\ell_{\text{inv}}$, but does not alter $\beta_{\text{inv}}$ (or $\beta_U^{\text{c}}$), and therefore results in the same on-shell symplectic form $\Omega_m$.[46] Of course, the corresponding boundary action *does* shift under these alternate choices, by a term that can precisely be interpreted as a generating function for the said canonical transformation. Essentially, a choice of $\beta_{\text{inv}}$ designates conjugate pairs of degrees of freedom, and the symplectic form $\Omega_m$ appropriate to these pairs is fixed. The sense in which $\Omega_m$ can still depend on the choice of boundary condition is if an alternate decomposition

$$\theta_{\text{inv}}\big|_\gamma = -\delta\ell'_{\text{inv}} + \mathrm{d}\beta'_{\text{inv}} + \varepsilon'_{\text{inv}} \tag{223}$$

designates a different set of conjugate pairs of degrees of freedom which still define a well-posed dynamical problem on $m$.

We now compare the behavior of these "ambiguity-resolved" symplectic structures

$$\Omega^{\text{CLPF}}(\sigma) - \delta \int_{\partial\sigma} \beta_{\text{ext}}$$

under the alternative choices (213) and (219) for $\beta_{\text{ext}}$. We consider both spacetime diffeomorphisms and relational spacetime diffeomorphisms, here restricting to field-independent generators in each case. A more complete analysis of our choice $\Omega(\sigma) \approx \Omega_m$ (equivalent to using $\beta_{\text{ext}} = U^\star[\beta_U^{\text{c}} - q_\chi]$) is given in Sec. 5, and a more complete discussion of $\Omega^{\text{S}}$ (which uses $\beta_{\text{ext}} = U^\star[\beta + \chi \lrcorner \bar{\ell}]$) is given in [25]. The contributions from $\Omega^{\text{CLPF}}(\sigma)$ can already be ascertained from Tables 1 and 2:

$$\begin{aligned}
-\hat{\xi} \cdot \Omega^{\text{CLPF}}(\sigma) &= \delta \int_\sigma U^\star[C_\xi] + \delta \int_{\partial\sigma} U^\star[q_\xi] \,, \\
-\breve{\rho} \cdot \Omega^{\text{CLPF}}(\sigma) &= \int_\sigma \rho \lrcorner U^\star[E] - \int_{\partial\sigma} \rho \lrcorner U^\star[\theta + \chi \lrcorner L] \,.
\end{aligned} \tag{224}$$

---

[46]For example, a switch to Dirichlet boundary conditions requires

$$\ell_{\text{inv}} \to \ell_{\text{inv}} - \phi_{\text{inv}}\pi_{\text{inv}} \,, \quad \varepsilon_{\text{inv}} \to -\phi_{\text{inv}}\delta\pi_{\text{inv}} \,, \quad \beta_{\text{inv}} \to \beta_{\text{inv}} \,.$$

All that remains is to compute the modification due to the added boundary term $-\delta \int_{\partial\sigma} \beta_{\text{ext}}$ under contraction of field space vectors $-\hat{\xi}$ and $-\check{\rho}$. Using the fact that for any form $\alpha$

$$\hat{\xi} \cdot \delta U^\star[\alpha] = U^\star[\Delta_\xi \alpha] - \delta U^\star[\hat{\xi} \cdot \alpha]$$
$$\check{\rho} \cdot \delta U^\star[\alpha] = (\Delta_\rho + \mathcal{L}_\rho)U^\star[\alpha] - \delta U^\star[\check{\rho} \cdot \alpha],$$

(225)

we have

$$\hat{\xi} \cdot \delta U^\star[\beta + \chi \lrcorner \bar{\ell}] = U^\star[\Delta_\xi(\beta + \chi \lrcorner \bar{\ell}) - \delta_\chi(\hat{\xi} \cdot \beta - \xi \lrcorner \bar{\ell})]$$
$$\check{\rho} \cdot \delta U^\star[\beta + \chi \lrcorner \bar{\ell}] = \Delta_{\check{\rho}} U^\star[\beta + \chi \lrcorner \bar{\ell}] + \rho \lrcorner U^\star[d(\beta + \chi \lrcorner \bar{\ell}) - \delta_\chi \bar{\ell}]$$
$$\hat{\xi} \cdot \delta U^\star[\beta_U^c - q_\chi] = -\delta U^\star[q_\xi]$$
$$\check{\rho} \cdot \delta U^\star[\beta_U^c - q_\chi] = \Delta_{\check{\rho}} U^\star[\beta_U^c - q_\chi] + \rho \lrcorner dU^\star[\beta_U^c - q_\chi] + \delta U^\star[q_{\check{\rho}} - \check{\rho} \cdot \beta_U^c],$$

(226)

where we have dropped total derivative terms that will vanish upon integration over $\partial\sigma$.

When combined with the terms of (224), and using (211) to simplify, the former choice gives the following results for

$$\Omega^S = \Omega^{\text{CLPF}}(\sigma) - \int_{\partial\sigma} \delta U^\star[\beta + \chi \lrcorner \bar{\ell}] :$$

(227)

$$-\hat{\xi} \cdot \Omega^S \approx \delta \int_{\partial\sigma} U^\star[q_\xi + \xi \lrcorner \bar{\ell} - \hat{\xi} \cdot \beta] + \int_{\partial\sigma} U^\star[\Delta_\xi(\beta + \chi \lrcorner \bar{\ell})],$$

(228)

$$-\check{\rho} \cdot \Omega^S \approx -\int_{\partial\sigma} \rho \lrcorner U^\star[\theta + \chi \lrcorner L + \delta_\chi \bar{\ell} - d(\beta + \chi \lrcorner \bar{\ell})] + \int_{\partial\sigma} \Delta_{\check{\rho}} U^\star[\beta + \chi \lrcorner \bar{\ell}],$$

(229)

$$-\check{\rho}^\| \cdot \Omega^S \approx -\int_{\partial\sigma} \rho^\| \lrcorner U^\star[\varepsilon + \chi \lrcorner(L + d\bar{\ell})].$$

(230)

The third expression considers the case that $\rho$ is tangential to the timelike boundary $\gamma$. We have left possible anomalous flux terms for generality, allowing for agnosticism about how $\beta$ and $\bar{\ell}$ are extended off of $\Gamma$. However for $\rho^\|$ along the timelike boundary we assume vanishing anomaly. A full discussion of the utility of this symplectic structure is given in [25]. Here we note that on the relational spacetime side, there are generically no nondegenerate, integrable charges. Transformations preserving $\partial\sigma$ are all pure gauge, and on shell of boundary conditions there are additional gauge directions along $\rho$ preserving $\gamma$. (Recall that for this choice of $\beta_{\text{ext}}$, the quantity $\varepsilon^S_{\text{ext}} = U^\star[\varepsilon + \chi \lrcorner(L + d\bar{\ell})]$ vanishes if boundary conditions are imposed.)

Taking the choice $\beta_{\text{ext}} = U^\star[\beta_U^c - q_\chi]$ gives the form arrived at in section 4:

$$\Omega_m = \Omega^{\text{CLPF}}(\sigma) - \int_{\partial\sigma} \delta U^\star[\beta_U^c - q_\chi].$$

(231)

After a few manipulations, the contractions reduce to

$$-\hat{\xi} \cdot \Omega_m \approx 0,$$

(232)

$$-\check{\rho} \cdot \Omega_m \approx \int_{\partial\sigma} \rho \lrcorner U^\star[-\theta_U^c + d\beta_U^c] + \delta \int_{\partial\sigma} U^\star[q_{\check{\rho}} - \check{\rho} \cdot \beta_U^c] + \int_{\partial\sigma} \Delta_\rho U^\star[\beta_U^c],$$

(233)

$$-\check{\rho}^\| \cdot \Omega_m \approx -\int_{\partial\sigma} \rho^\| \lrcorner \varepsilon_{\text{inv}} + \delta \int_{\partial\sigma} \left( U^\star[q_{\check{\rho}^\|}] + \rho^\| \lrcorner \ell_{\text{inv}} - \check{\rho}^\| \cdot \beta_{\text{inv}} \right).$$

(234)

This coincides with the results of Sec. 5 (recall that $\bar{\rho} = U_\star[\rho]$). As expected, the spacetime diffeomorphisms are all gauge. For the relational spacetime diffeomorphisms, when $\rho$ preserves $\partial\sigma$ there is generically a nonvanishing charge. On shell of the boundary conditions (with $\varepsilon_{\text{inv}} = 0$) the symmetries are enhanced to include transformations tangential to the

timelike boundary $\gamma$. These constitute the symmetries discussed in Sec. 5.4. Note that the integrable charge density appearing in (234) takes the same form on relational spacetime as that in (228) for spacetime transformations. The form matches Harlow and Wu's [14] systematic treatment of covariant phase space with boundaries, but now in terms of dressed quantities on relational spacetime $\mathfrak{m}$, as should be expected given the shared criteria of conservation of subregion symplectic structure and well-posedness of the action variational problem.

As emphasized by Speranza [25], the fact that spacetime diffeomorphisms are pure gauge transformations for the choice (232) could be used to set $\chi = 0$ as a gauge choice. The action of a generic frame reorientation would be incompatible with this choice, as $\breve{\rho} \cdot \chi = \bar{\rho}$, but in conjunction with the rule $\hat{\breve{\xi}} \cdot \chi = -\xi$, the condition on $\chi$ can be held fixed through the combined action of $\breve{\rho} + \hat{\breve{\xi}}_\rho$ with $\xi_\rho = \bar{\rho}$. A similar statement can be made about the choice (227), in that gauge directions of both $\rho$ and $\xi$ preserving the corner can be used to set $\chi = 0$ everywhere off the corner, and under this choice the boundary condition itself sets $\chi = 0$ on the boundary. These statements seem to call into question the utility of the extended phase space. However we believe that the ability to canonically represent extended corner symmetries (first claimed in [23] and [24], even though we do not take this choice of symplectic structure here), the off-shell canonical representation of the constraint algebra of general diffeomorphisms (see equation (93)), and the enhanced flexibility in identifying bracket structures are all clear indications that the extended phase space represents an important step forward, though its ultimate role is yet to be fully understood.

# 8 Conclusion

The inclusion of dynamical embedding fields in the gravitational phase space has led to new insights and reinvigorated old questions on the road to quantum gravity: How is a phase space of a local subregion properly defined? What is the appropriate symplectic structure and associated canonical bracket which lends itself most naturally to the analysis of symmetries, charges, and ultimately, quantization? The original appearance of edge modes in this context [10] from variations of a dynamical embedding map indicates that the added structure is necessary to account for the relational gluing of a subregion and its complement in a diffeomorphism invariant theory [110].

In this work we have tried to put more "meat on the bone" in the interpretation of dynamical embedding maps (denoted $U$) as diffeomorphism-valued, relational reference frames. Spacetime tensors are rendered gauge-invariant by dressing with a reference frame $U$. At a mechanical level, a "dressed tensor" is invariant in the same sense that a spacetime tensor, *expressed in a particular coordinate system*, is trivially invariant under additional coordinate changes owing to the already-explicit reference to a fixed frame. But in Sec. 2 we have outlined some physically-motivated constructions of such reference frames $U$ that are intrinsically relational, relating dynamical degrees of freedom of a theory amongst themselves, hence also justifying the frames' status as dynamical. This leads to a conceptually powerful picture of the map $U : \mathcal{M} \to \mathfrak{m}$ as moving between spacetime $\mathcal{M}$ and what we call *relational spacetime* $\mathfrak{m}$.

In Sec. 3, we reviewed the variational principle in the framework of the covariant phase space formalism, both in the presence and absence of frame fields. We there provided some crucial formalism and terminology around the dressing of noncovariant objects on spacetime, generalizing the procedures outlined in [1] for other gauge theories. Generically, spacetime forms of nonzero field-space degree divide into a $U$-covariant part, and a non-$U$-covariant part on spacetime. When pushed forward via $U$ to relational spacetime, these become the *invariant* and *gauge* parts, respectively.

With this machinery in place, we proceeded in Sec. 4 to identify the symplectic structures appropriate to subregion theories through the process of splitting post-selection, again following the program outlined in [1]. One of the key guiding principles employed in this construction, which differentiates it from other proposals in the literature, is the consideration that the region of interest is a subsystem within a well-defined global theory. The bulk form of the subregion symplectic structure is directly inherited from that of the global theory, while any additional boundary contribution should, in a sense, justify itself as only playing the role that would have been played by the complement region in the full theory. This also entails that the symplectic structure of the complement region, under the same construction, includes equal-and-opposite boundary contributions such that together they are additive to the global theory; nothing has been fundamentally added which was not already present in the global theory. Similar considerations apply to the analysis of symmetries and gauge transformations. Gauge directions of the global presymplectic structure remain such upon reduction to the subregion. Those that act nontrivially anywhere in the subsystem, inluding at the boundary, are generated by first class constraints which close an algebra off-shell. Cleanly distinguished from these are the relational spacetime transformations, which may act either as the physical symmetries of the subregion theory or as open system transformations.

A closely tied guiding principle is that the subregion boundary itself is defined relationally, and any boundary conditions which might be imposed should only constrain gauge-invariant (relational) combinations of degrees of freedom. Upon choosing a suitable class of boundary conditions (deciding which quantities are to be held fixed on the boundary), we enforce that the symplectic form is conserved: we consider subregions of cylindrical topology with timelike boundary and choose that the symplectic form is independent, on-shell, of any choice of Cauchy surface transecting this cylinder. Under these conditions, we find that the resulting symplectic structure has a universal contribution that corresponds to using $U^\star[\tilde{\theta}_m + \ldots] = U^\star[\theta + \chi \lrcorner L + \mathrm{d}q_\chi + \ldots]$ as an extended symplectic potential current, where $\theta$ is the (any) "bare" potential current and $\chi$ is the Maurer-Cartan form. The ellipses indicate a non-universal part, with terms that may appear depending on a choice of boundary condition on the timelike boundary.

In Sec. 5 we demonstrated that for this choice of symplectic structure, all spacetime diffeomorphisms are gauge transformations regardless of how they behave near the subregion boundary. This is in line with our physical expectation that any gauge transformation of the global theory ought to remain such on a subsystem. There remains, however, an off-shell representation of the first-class constraint algebra which is anti-homomorphic to the Lie algebra of spacetime vector fields generating spacetime diffeomorphisms (see equation (93)). Having a proper representation of the Lie algebra of diffeomorphisms may prove advantageous in the quantum theory. We also investigated the action of relational spacetime diffeomorphisms, or frame reorientations acting on the frame $U$. Transformations along the timelike boundary are integrable, on shell of chosen boundary conditions, and those which preserve the boundary conditions constitute a subalgebra of conserved charges. Transformations preserving a chosen corner on the timelike boundary manifest a corner algebra, which universally includes $\mathrm{Diff}(s)$ and may, under alternate implementations, be extended (though in our example of Einstein-Hilbert gravity, the natural choice when considering Dirichlet or Neumann conditions on the boundary induced metric only includes $\mathrm{Diff}(s)$).

In section 6, we described the systematic generation of subregion actions from a global action via the process of post-selection. The method conceptually distinguishes two boundary contributions to the subregion action, one which ensures the validity of the variational problem and the conservation of subregion symplectic structure under a given *type* of boundary condition, and another which dynamically imposes a specific boundary condition and implements a specific gluing of the subregion theory into the complement. After illustrating this

procedure for a variety of boundary conditions in Einstein-Hilbert gravity, we concluded in section 7 with a relatively self-contained discussion of how our choice of symplectic structure relates to the many alternatives explored in recent literature.

At present, the full utility of the extended phase space is probably yet to be fully appreciated or understood. We believe that our choices for extended symplectic structure in this work are well-motivated by a strong adherence to the principle of general covariance. This includes the idea that subsystems of a diffeomorphism invariant theory ought to themselves be relationally defined, any boundary conditions imposed ought to restrict only gauge-invariant combinations of degrees of freedom, and any gauge transformations of the global theory ought to retain their status as gauge transformations when acting on such a subsystem. In view of the quantum theory, these gauge transformations should maintain their algebraic properties of the global theory, in this case a Poisson bracket representation of the Lie algebra of spacetime vector fields in terms of the constraints of the theory. Furthermore, since the dynamical frame fields introduced for these purposes are built out of the fundamental fields of the global theory as seen in section 2, the latter does not have to be extended by the embedding fields. As such, it is natural to also consider the resulting edge mode fields within the ensuing subregion theory and hence the extended phase space, even though this is equivalent to a gauge-fixed construction without edge modes. In particular, the nonlocal nature of the (immaterial) edge mode frame fields permits one to encode how the subregion of interest relates to its complement. These principles lead to a set of presymplectic structures that have not appeared elsewhere in the literature, and which are naturally related to choices of conjugate variables on which boundary conditions may be imposed. Aside from the examples in Einstein-Hilbert gravity in section 6.2, specific implementations of the formalism to various diffeomorphism invariant theories have been left to future work. Likewise, generalizations to include null boundaries, higher codimension corner terms, subregions of nontrivial topology, and applications to specific boundaries such as horizons all remain interesting future directions to investigate.

# Acknowledgements

We would like to thank Josh Kirklin, Fabio Mele, and Antony Speranza for some useful discussions.

**Funding information** In the course of this project SC was supported by a Radboud Excellence Fellowship from Radboud University in Nijmegen, the Netherlands. PH is grateful for support from the Foundational Questions Institute under grant number FQXi-RFP-1801A. This work was supported in part by funding from Okinawa Institute of Science and Technology Graduate University. In addition, this project/publication was made possible through the support of the ID# 62312 grant from the John Templeton Foundation, as part of the 'The Quantum Information Structure of Spacetime' Project (QISS). The opinions expressed in this project/publication are those of the author(s) and do not necessarily reflect the views of the John Templeton Foundation.

# A Intuition behind the Maurer–Cartan form $\chi$

The inclusion of dynamical embedding maps into phase space [10, 11] naturally leads to the emergence of the Maurer–Cartan form $\chi$ as an important object. When first encountered it may be somewhat counter-intuitive, so we here offer an informal explanation of some of its key properties.

For any tensor $\alpha$, dressed with dynamical reference frame (or embedding map) $U$, an arbitrary field space variation of the object $U^\star[\alpha]$ should allow for contributions from both the tensor variation $\alpha$ and the variation of the map $U$:

$$\delta U^\star[\alpha] = (\delta U)^\star[\alpha] + U^\star[\delta\alpha].$$

Intuitively, the first term expresses the difference between the pushforward of $\alpha$ under two infinitesimally nearby maps $U_1$ and $U_2$. It is always possible to relate two such maps by a spacetime diffeomorphism $\phi : M \to M$ such that $U_2 = U_1 \circ \phi$. By saying the two maps are "nearby" we mean we can take $\phi$ to be arbitrarily close to the identity diffeomorphism. For instance, if we had a one-parameter family of diffeomorphisms $\phi_\lambda$, these define a one-parameter family of maps $U_\lambda$ by

$$U_\lambda = U \circ \phi_\lambda. \tag{A.1}$$

where $U$ is a fixed map and $\phi_{\lambda=0}$ is the identity diffeomorphism. Then we could freely compute

$$\begin{aligned}
\frac{\mathrm{d}}{\mathrm{d}\lambda} U_\lambda^\star[\alpha]\Big|_{\lambda=0} &= \lim_{\lambda\to 0} \frac{1}{\lambda}\big[(U \circ \phi_\lambda)^\star[\alpha] - (U \circ \phi_0)^\star[\alpha]\big] \\
&= U^\star\left[\lim_{\lambda\to 0}\left(\frac{\phi_\lambda^\star(\alpha) - \phi_0^\star(\alpha)}{\lambda}\right)\right] \\
&= U^\star\big[-\mathcal{L}_\nu(\alpha)\big]
\end{aligned} \tag{A.2}$$

where $\nu$ is the spacetime vector field that generates the one-parameter family of diffeomorphisms:

$$\nu := \frac{\mathrm{d}\phi_\lambda}{\mathrm{d}\lambda}\bigg|_{\lambda=0},$$

which can also be extracted through the relation

$$-\nu = \frac{\mathrm{d}U_\lambda^{-1}}{\mathrm{d}\lambda} \circ U_\lambda\bigg|_{\lambda=0}. \tag{A.3}$$

In the absence of a specific one-parameter family of diffeomorphisms $\phi_\lambda$ or maps $U_\lambda$, a general field space variation of $U$ leads to an analogous object:

$$\chi := \delta U^{-1} \circ U \tag{A.4}$$

which is both a field space one-form and a spacetime vector field. It is defined such that

$$\hat{\xi} \cdot \chi = -\xi, \tag{A.5}$$

where $\hat{\xi}$ is the field space vector that generates diffeomorphism associated with the spacetime vector field $\xi$. This recovers (A.3) for diffeomorphisms. It also ensures that $U^\star[\alpha]$ is diffeomorphism invariant if $\alpha$ is a covariant spacetime tensor and field-space zero form. More generally, $\alpha$ may be of arbitrary field-space degree if it is what we have called a $U$-covariant object in the main text (see Sec. 3.3), such that $\hat{\xi} \cdot \delta\alpha = \mathcal{L}_\xi\alpha$. A general variation is then

$$\delta U^\star[\alpha] = U^\star[\delta\alpha + \mathcal{L}_\chi\alpha] \qquad \Longrightarrow \qquad \hat{\xi} \cdot \delta U^\star[\alpha] = 0. \tag{A.6}$$

We recommend section III and appendices of [24] for a concise but relatively complete exposition of properties of $\chi$ as field-space connection and its incorporation into the variational bicomplex [97].

# B Examples of $U$-covariant forms

## B.1 Covariant frame-dressing of Christoffel symbols

As an example of covariant dressing of a non-covariant field-space scalar, let us consider the Levi-Civita connection one-form $\Gamma[g]$ (i.e. the Christoffel symbols) associated to $(\mathcal{M}, g)$. The defining equation (50) simply means that $(\Gamma[g])_{\text{inv}} = \Gamma[U^\star g]$ is the connection one-form associated to $(\mathfrak{m}, U^\star g)$. Plugging-in the coordinate expressions of the Christoffel symbols, we obtain

$$(\Gamma[g]_{\text{inv}})^A_{BC} = \frac{\partial U^A}{\partial x^a} \frac{\partial x^b}{\partial U^B} \frac{\partial x^c}{\partial U^C} \Gamma[g]^a_{bc} - \frac{\partial^2 U^A}{\partial x^a \partial x^b} \frac{\partial x^a}{\partial U^B} \frac{\partial x^b}{\partial U^C}, \tag{B.1}$$

which can be equivalently interpreted as a (passive) change of coordinates form $x^a$ to $U^A$ on $\mathcal{M}$. The covariant and gauge components are finally obtained by pulling-back this expression to $\mathcal{M}$:

$$(\Gamma[g]^{\text{c}}_U)^a_{bc} = \Gamma[g]^a_{bc} - \frac{\partial x^a}{\partial U^A} \frac{\partial^2 U^A}{\partial x^b \partial x^c}, \qquad (\Gamma[g]^{\text{nc}}_U)^a_{bc} = \frac{\partial x^a}{\partial U^A} \frac{\partial^2 U^A}{\partial x^b \partial x^c}. \tag{B.2}$$

## B.2 Covariant dressing of a field-space two-form

Consider a field-space two-form $\alpha$. We want to prove that its $U$-covariant component is

$$\alpha^{\text{c}}_U = \alpha + \frac{3}{2} \hat{\chi} \cdot \alpha + \frac{1}{2} \hat{\chi} \cdot \hat{\chi} \cdot \alpha, \tag{B.3}$$

which coincides with the formula stated for $\omega$ in equation (65). By linearity, it is sufficient to prove this formula for $\alpha = \delta P \delta Q$, where $P$ and $Q$ are two field-space 0-forms. We this assumption, we find that

$$\alpha^{\text{c}}_U = \delta_\chi P \delta_\chi Q = \alpha + (\mathcal{L}_\chi P)\delta Q + \delta P(\mathcal{L}_\chi Q) + (\mathcal{L}_\chi P)(\mathcal{L}_\chi Q). \tag{B.4}$$

On the other hand, we can compute

$$\hat{\chi} \cdot \alpha = (\mathcal{L}_\chi P)\delta Q + \delta P(\mathcal{L}_\chi Q), \tag{B.5}$$

as well as (by invoking $\hat{\chi} \cdot \chi = -\chi$)

$$\hat{\chi} \cdot \hat{\chi} \cdot \alpha = -(\mathcal{L}_\chi P)\delta Q - \delta P(\mathcal{L}_\chi Q) + 2(\mathcal{L}_\chi P)(\mathcal{L}_\chi Q). \tag{B.6}$$

As a result, we can write the last equality as $(\mathcal{L}_\chi P)(\mathcal{L}_\chi Q) = \frac{1}{2}(\hat{\chi} \cdot \hat{\chi} \cdot \alpha + \hat{\chi} \cdot \alpha)$, which, together with (B.5), allows to identify (B.4) with the looked-for expression (B.3).

Finally, we can check that formula (B.3) is consistent with the general observation we made in equation (66):

$$\hat{\xi} \cdot \alpha^{\text{c}}_U = \hat{\xi} \cdot \left( \alpha + \frac{3}{2} \hat{\chi} \cdot \alpha + \frac{1}{2} \hat{\chi} \cdot \hat{\chi} \cdot \alpha \right) \tag{B.7}$$

$$= \hat{\xi} \cdot \alpha + \frac{3}{2} \left( \hat{\chi} \cdot \hat{\xi} \cdot \alpha - \hat{\xi} \cdot \alpha \right) + \frac{1}{2} \left( \hat{\chi} \cdot \hat{\xi} \cdot \hat{\chi} \cdot \alpha - \hat{\xi} \cdot \hat{\chi} \cdot \alpha \right) \tag{B.8}$$

$$= \hat{\xi} \cdot \alpha + \frac{3}{2} \left( \hat{\chi} \cdot \hat{\xi} \cdot \alpha - \hat{\xi} \cdot \alpha \right) + \frac{1}{2} \left( \hat{\chi} \cdot \hat{\chi} \cdot \hat{\xi} \cdot \alpha - \hat{\chi} \cdot \hat{\xi} \cdot \alpha - \hat{\xi} \cdot \hat{\chi} \cdot \alpha \right) \tag{B.9}$$

$$= \hat{\xi} \cdot \alpha + \frac{3}{2} \left( \hat{\chi} \cdot \hat{\xi} \cdot \alpha - \hat{\xi} \cdot \alpha \right) + \frac{1}{2} \left( \hat{\chi} \cdot \hat{\chi} \cdot \hat{\xi} \cdot \alpha - 2\hat{\chi} \cdot \hat{\xi} \cdot \alpha + \hat{\xi} \cdot \alpha \right) \tag{B.10}$$

$$= \hat{\xi} \cdot \alpha + \frac{3}{2} \left( \hat{\chi} \cdot \hat{\xi} \cdot \alpha - \hat{\xi} \cdot \alpha \right) + \frac{1}{2} \left( -3\hat{\chi} \cdot \hat{\xi} \cdot \alpha + \hat{\xi} \cdot \alpha \right) = 0, \tag{B.11}$$

where we have repeatedly used the relation $[\hat{\xi}\cdot, \hat{\chi}\cdot] = -\hat{\xi}\cdot$, and in the last line, we remarked that because $\hat{\xi} \cdot \alpha$ is a one-form, we must have $\hat{\chi} \cdot \hat{\chi} \cdot \hat{\xi} \cdot \alpha = \widehat{\hat{\chi} \cdot \chi}\,\hat{\xi} \cdot \alpha = -\hat{\chi} \cdot \hat{\xi} \cdot \alpha$.

## C  Anomalies from the corner term

In our post-selection of symplectic structure, the corner contribution to the symplectic form of a subregion included a $(d-2,1)$ form $\beta_U^c$ (see equation (83)). As with other $U$-covariant objects, we relate this to the invariant form $\beta_{\text{inv}}$ on $\partial\sigma$ as $\beta_{\text{inv}} := U^\star[\beta_U^c]$. In our analysis of the charge algebras, we allowed for an anomalous contribution from $\beta_{\text{inv}}$ under relational space-time diffeomorphisms, but no such contribution from $\beta_U^c$ under spacetime diffeomorphisms. The asymmetry arises from the fact that the relational Cauchy surface $\sigma = U(\Sigma)$ and timelike boundary $\gamma = U(\Gamma)$ are seen as fundamental in defining the subregion, and held fixed in the variational principle. We explain this below with reference to the timelike boundary $\gamma$, though similar statements apply to $\sigma$ and the corner $\partial\sigma$.

The pullback of a $U$-covariant form to the boundary $\Gamma$ implies a geometric dependence which, for timelike boundaries, can be entirely expressed in terms of the unit normal $n$ to $\Gamma$. Letting $\Gamma$ correspond to a level set of some smooth function $\bar{f}$ on spacetime, we can give an explicit coordinate expression to $n_\mu$ as

$$n_\mu = \frac{\nabla_\mu \bar{f}}{\sqrt{g^{\alpha\beta}\nabla_\alpha \bar{f}\nabla_\beta \bar{f}}}, {}^{47} \tag{C.1}$$

where $\bar{f}$ is given arbitrary smooth extension around $\Gamma$ so that this defines a normalized vector field in the vicinity of $\Gamma$. Because the surface $\gamma = U(\Gamma)$ is held fixed in the variational principle, the function $f = U^\star[\bar{f}]$ is considered a background function on relational spacetime, implying that $\delta_\chi \bar{f} = 0$ on spacetime. The result is that $U^\star[n]$ may be anomalous with respect to the relational spacetime diffeomorphisms which do not fix $\gamma$, whereas $n$ is not anomalous under any spacetime diffeomorphism:

$$\begin{aligned}
\Delta_\xi n_\mu &= \hat{\xi}\cdot\delta n_\mu - \mathcal{L}_\xi n_\mu \\
&= \hat{\xi}\cdot\left(\frac{\partial n_\mu}{\partial\nabla_\alpha\bar{f}}\nabla_\alpha\delta\bar{f} + \frac{\partial n_\mu}{\partial g_{\alpha\beta}}\delta g_{\alpha\beta}\right) - \mathcal{L}_\xi n_\mu \\
&= 0
\end{aligned} \tag{C.2}$$

whereas (denoting $U_\star[\rho]$ as $\bar{\rho}$)

$$\begin{aligned}
\Delta_\rho U^\star[n_\mu] &= \check{\rho}\cdot\delta U^\star[n_\mu] - \mathcal{L}_\rho U^\star[n_\mu] \\
&= \check{\rho}\cdot U^\star\left[\frac{\partial n_\mu}{\partial g_{\alpha\beta}}\delta_\chi g_{\alpha\beta}\right] - \mathcal{L}_\rho U^\star[n_\mu] \\
&= -U^\star\left[\frac{\partial n_\mu}{\partial\nabla_\alpha\bar{f}}\nabla_\alpha\mathcal{L}_{\bar{\rho}}\bar{f}\right].
\end{aligned} \tag{C.3}$$

In the latter we used the fact that $\delta_\chi\bar{f} = 0$. The last expression can be written explicitly as

$$\Delta_\rho U^\star[n_\mu] = -U^\star[n_\beta h_\mu{}^\alpha\nabla_\alpha\bar{\rho}^\beta + \bar{\rho}^\beta\nabla_\beta n_\mu], \tag{C.4}$$

from which it can be confirmed that for any $\bar{\rho}$ tangential to $\Gamma$, the anomaly vanishes. For $\bar{\rho}$ which is not tangential to $\gamma$, the anomaly may be nonvanishing. However it will depend on the

---

[47]In our example of general relativity in Sec. 6.2, we let $n$ denote the normal to $\gamma$ on relational spacetime, what is here denoted $U^\star[n]$. For the illustrations of this appendix we find it convenient to work primarily with the objects on spacetime. We do, however, use an overbar on $\bar{f}$ and $\bar{\rho}$, to strongly distinguish these from their relational spacetime counterparts $f = U^\star[\bar{f}]$ and $\rho = U^\star[\bar{\rho}]$, which are taken as background objects in field space variations.

arbitrary extension of $f$ away from $\gamma$ (equivalently, a choice of foliation by nearby level sets). There is a corresponding possibility of anomalous $\Delta_\rho \beta_{\text{inv}}$ for $\rho$ not preserving the boundary. We therefore allow for this contribution to the flux in equation (103), though it has no impact on the charges algebras discussed afterward, which focus on $\rho$ tangential to $\gamma$.

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
