# Peer review of "Edge modes as dynamical frames: charges from post-selection in generally covariant theories"

_SciPost Physics, doi:SciPost Phys. 17, 048 (2024)_

## Round 2 · Referee Report · Anonymous (Referee 1) · 2024-2-16

Strengths

1- Original and innovative work on the relation between dynamical reference frames and the diffeomorphism gauge invariance of general relativity in the presence of space-time boundaries

2- Mathematically clean treatment of the symplectic potential for general relativity in the covariant phase space formalism with boundaries, and of the dressing of observables with reference frames.

3- Self-contained work

4- Contains a thorough comparison with the several other work on the symplectic potential and symmetries of general relativity, with very clear and helpful technical details

Weaknesses

Weaknesses

1- Long, technical paper, insufficiently highlighting important or new features

2- Illustrations are not particularly helpful.

Report

This manuscript presents original and innovative work on the relation between dynamical reference frames and the diffeomorphism gauge invariance of general relativity in the presence of space-time boundaries. In more technical terms, it provides a mathematically clean treatment of the symplectic potential for general relativity in the covariant phase space formalism with boundaries, and of the appropriate dressing of observables with reference frames to make them gauge-invariant. The work is entirely self-contained, though long and technical as a result. It contains a thorough comparison with the several other works on the symplectic potential and symmetries of general relativity, with very clear and helpful technical details. It is clearly meant to become the reference on the subject. It is definitely a paper that deserves to be published and that meets the exceptionality standards of SciPost.

Nevertheless, the presentation of the work could be substantially improved by properly highlighting the important and new features of the formalism, as well as improving the present figures and adding new illustrations (e.g. for eqn (23), (57), (60), (156-157) and so on…) in order to enhance the readability of the manuscript, help the reader through the technicalities and clarifies the physical meaning of the various terms of the derived equations.
In particular, I would suggest to the authors to add to the Conclusion (Section 8) explanations on the appropriate chosen result and equation of the approach (the equation relating the map U and the symplectic potential or the derived charges or what the authors deem to be an essential result of their work), in order to highlight what should definitely be remembered beyond all the technical formalism. Moreover, it would be helpful to also describe what is meant mathematically and physically by “relational map” at the end of the 2nd paragraph in the conclusion.

---

## Round 2 · Referee Report · Anonymous (Referee 2) · 2024-7-16

Report

This paper is an interesting addition to the covariant phase space formalism in the presence of boundaries. The idea is to treat a subregion of spacetime as dynamical, and let the rest define a frame for it. Some parts are perhaps unnecessarily verbose, but the mathematical treatment appears useful.

Recommendation

Publish (easily meets expectations and criteria for this Journal; among top 50%)

---

## Editorial Decision

published